# Mitigating Premature Exploitation in Particle-based Monte Carlo for Inference-Time Scaling

Giorgio Giannone [1]   Guangxuan Xu [1]   Nikhil Nayak [1]

Rohan Awhad [1]   Shivchander Sudalairaj [1]   Kai Xu [1]   Akash Srivastava [1,2]

## Abstract

Inference-Time Scaling (ITS) improves language models by allocating more computation at generation time. Particle Filtering (PF) has emerged as a strong ITS method for complex mathematical reasoning tasks, but it is vulnerable when guided by process reward models, which often assign overconfident scores early in the reasoning process. This causes PF to suffer from premature exploitation: it myopically commits to locally promising trajectories, prunes potentially correct hypotheses, and converges to suboptimal solutions. This failure mode, known as particle impoverishment, is especially severe under constrained computational budgets. To address this, we analyze the problem and identify two root causes: a lack of diversity in the particle set due to overconfident resampling and consequent inability to assess the potential of a reasoning path. We introduce *Entropic Particle Filtering* (`ePF`), an algorithm that integrates two new techniques to solve these issues. The first technique, *Entropic Annealing* (`EA`), directly mitigates particle impoverishment by monitoring search diversity via entropy; when diversity drops, it intervenes by dynamically annealing the resampling distribution to preserve exploration. The second, an enhancement called *Look-ahead Modulation* (`LaM`), adds a predictive guide to evaluate a state's potential based on its successors. By effectively balancing exploration and exploitation, `ePF` significantly outperforms strong baselines on challenging math benchmarks, achieving up to a 50% relative improvement in task reward.

[1]AI Innovation, Red Hat [2]Core AI, IBM. Correspondence to: <ggiorgio@mit.edu>.

*Proceedings of the $43^{rd}$ International Conference on Machine Learning*, Seoul, South Korea. PMLR 306, 2026. Copyright 2026 by the author(s).

## 1. Introduction

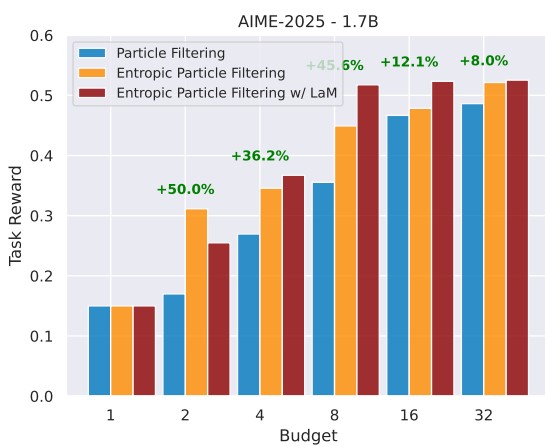

*Figure 1.* Task reward comparison on AIME-2025 using Qwen3-1.7B. Our Entropic Particle Filtering (`ePF`) and its Look-ahead variant (`ePF w/ LaM`) significantly improve performance over standard Particle Filtering (PF) across all particle budgets. This demonstrates that mitigating premature exploitation leads to significant performance gains.

Inference-Time Scaling (ITS) is a powerful paradigm for improving language model performance by allocating additional computation at generation time. Rather than decoding a single trajectory, ITS reframes generation as a guided search: multiple candidate solutions are explored in parallel, scored, and iteratively refined (Wei et al., 2022; Brown et al., 2024; Snell et al., 2024; Beeching et al., 2024). This approach has proven especially effective on reasoning tasks, where the search space is vast and correct answers are sparse.

Among ITS methods, *Particle Filtering (PF)* has emerged as a principled and efficient approach (Puri et al., 2025; Feng et al., 2024). PF maintains a set of candidate trajectories (*particles*), propagates them through the model, weights them using a process reward model (PRM), and resamples candidates with probabilities proportional to their weights. This propagate-weight-resample cycle adaptively focuses computation on promising regions of the search space while preserving some hypothesis diversity, often outperforming

well-established methods such as beam search or Best-of-$N$ sampling.

Despite these advantages, PF's effectiveness is often undermined by a critical vulnerability: when guided by imperfect PRMs, it is prone to premature exploitation. This issue is exacerbated by PF's inherent *myopia* - its inability to assess the long-term potential of a reasoning path beyond the immediate reward.

The early steps of a solution trajectory often carry little information about eventual correctness, yet uncalibrated PRMs frequently assign overconfident scores even at these stages (Park et al., 2025; Lightman et al., 2023). Ideally, rewards at these early stages should be conservative and relatively similar across trajectories, as the eventual solution quality cannot be confidently determined from initial steps alone, yielding a flatter resampling distribution that maintains diversity.

When PF resamples on the basis of these noisy, overconfident signals, it prematurely concentrates probability mass on a small set of trajectories, often trapping the search in a locally optimal, but globally incorrect, solution.

To illustrate the overconfidence phenomena in pretrained PRMs, we plot the histogram of rewards from PRM and estimated ground truth via Monte Carlo sampling in Fig. 2a. It can be seen that PRMs consistently assign higher rewards to partial solutions than is warranted by the final probability of correctness. A direct result of the overconfident rewards is high variance in the resampling distribution, which we show in Fig 2b. High variance diminishes particle diversity and results in a greedy-like search. This phenomenon, known as *particle impoverishment*, prematurely prunes viable hypotheses before they can reveal their value, causing the search to collapse into suboptimal solutions. As a final remark, we confirm that ESS, a measure of particle diversity that is inversely related to variance, is indeed correlated with the final performance of the algorithm (Fig. 2c). performance tracks closely with the effective sample size (ESS) - a direct measure of particle activation and variety - and the entropy of the resampling distribution.

Thus, a gap remains for a search method that is inherently robust against reward miscalibration and overconfidence. Our central hypothesis is that by dynamically preserving search diversity and incorporating forward-looking guidance, we can create a more resilient particle filtering algorithm.

To this end, we introduce *Entropic Particle Filtering (ePF)*, a robust extension of Particle Filtering designed to maintain exploration and prevent premature convergence. ePF integrates two complementary mechanisms: *(i) Entropic Annealing (EA)*, which dynamically adjusts resampling temperature based on particle diversity to avoid collapse, and *(ii) Look-ahead Modulation (LaM)*, which uses a one-step

look-ahead to bias sampling toward trajectories with high long-term potential.

**Contribution**  Our contributions are:

- We introduce Entropic Particle Filtering (ePF), which uses Entropic Annealing (EA) to dynamically modulate the resampling step based on particle diversity to prevent premature collapse.
- We propose Look-ahead Modulation (LaM), a one-step, forward-looking guidance mechanism that re-weights particles based on the predicted quality of their successors.
- We demonstrate a strong correlation between premature exploitation and poor performance, confirming that robust exploration is key to finding high-quality solutions.
- We show that ePF significantly outperforms strong baselines across several mathematical reasoning benchmarks, especially when operating under limited particle budgets.

By improving the exploration-exploitation balance, ePF makes the search more resilient to PRM miscalibration in long-horizon mathematical problems.

## 2. Background

**Sequential Importance Sampling**  For sequential models, where the distributions of interest evolve over time, Importance Sampling (IS; Kloek &Van Dijk, 1978; Robert et al., 1999) can be extended into a recursive framework known as Sequential Importance Sampling (SIS; Doucet et al., 2001a). This is the foundation of particle filters. The goal is to approximate the posterior distribution over a sequence of states $\mathbf{z}_{1:T}$ given a sequence of observations $\mathbf{o}_{1:T}$.

In the context of LLM inference, each state $\mathbf{z}_t$ is an intermediate sampling step, and the observation $\mathbf{o}_t$ is the scalar score provided by a PRM. In a standard state-space model with the Markov property, the un-normalized posterior can be factorized recursively. In particular given a posterior of the form $p(\mathbf{z}_{1:T}|\mathbf{o}_{1:T})$, leveraging a proposal $q(\mathbf{z}_{1:T})$ and using Bayes rule, we can write the importance weights $w_t$ for step $t$ as:

$$\begin{aligned}
\tilde{w}_t &= \frac{\tilde{p}(\mathbf{z}_{1:t}|\mathbf{o}_{1:t})}{q(\mathbf{z}_{1:t})} \\
&= \frac{p(\mathbf{o}_t|\mathbf{z}_t)p(\mathbf{o}_{1:t-1}|\mathbf{z}_{1:t-1})\, p(\mathbf{z}_t|\mathbf{z}_{t-1})p(\mathbf{z}_{1:t-1})}{q(\mathbf{z}_t|\mathbf{z}_{t-1})q(\mathbf{z}_{1:t-1})} \\
&= \tilde{w}_{t-1}\frac{p(\mathbf{o}_t|\mathbf{z}_t)p(\mathbf{z}_t|\mathbf{z}_{t-1})}{q(\mathbf{z}_t|\mathbf{z}_{t-1})},
\end{aligned} \quad (1)$$

where $\tilde{p}(\mathbf{z}_{1:t}|\mathbf{o}_{1:t})$ in Eq.1 represents the un-normalized posterior. A powerful computational simplification arises when we choose the proposal to be the model's dynamics, i.e.,

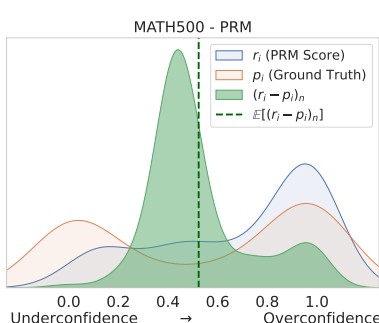

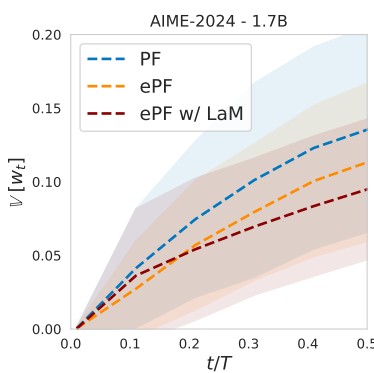

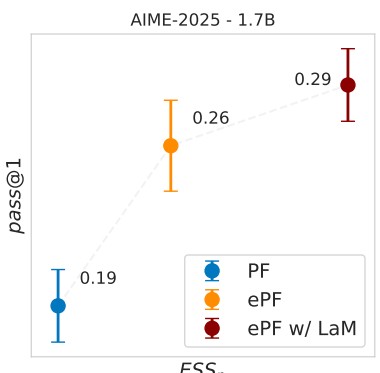

*(a)* Density of the PRM reward $r_i$ assigned to partial trajectories of 1000 tokens in length and the probability $p_i$ of the full response being correct for a 128 sample subset of MATH500.

*(b)* $\mathbb{V}[w_t]$ on AIME 2024. PF assigns overconfident scores early in the sampling process, generating high variance in the resampling distribution and poor state estimation.

*(c)* Expected pass@1 on AIME 2025 as a function of the normalized effective sample size. We aggregate pass@1 and $ESS_n$ over trajectories and particle budgets.

*Figure 2.* PRM overconfidence causes particle impoverishment and harms performance. **(a)** Uncalibrated PRMs contribute to low diversity by assigning overly optimistic scores to partial solutions (Fig. 17), causing Particle Filtering to converge prematurely (Fig. 9). **(b)** The variance of the resampling distribution increases with less particle diversity. **(c)** Task success is strongly correlated with high ESS.

$q(\mathbf{z}_t|\mathbf{z}_{t-1}) = p(\mathbf{z}_t|\mathbf{z}_{t-1})$. This choice gives rise to the (forward) Bootstrap Particle Filter (Gordon et al., 1993), and reduces the weight update to a simple multiplication by the observation likelihood:

$$\tilde{w}_t = \tilde{w}_{t-1}\, p(\mathbf{o}_t \mid \mathbf{z}_t). \tag{2}$$

This elegant result shows that the un-normalized importance weight at step $t$ is simply the previous weight at $t-1$ modulated by the likelihood of the current observation, efficiently propagating information through the sequence. More details in Appx A.

**Particle-based Monte Carlo**  Particle Filters (PF) are Sequential Monte Carlo (SMC) methods that use Sequential Importance Resampling (SIR; Liu et al., 2001) to approximate posterior distributions (Doucet et al., 2001a; Naesseth et al., 2019). They work by calculating sequential importance weights (Eq. 2) at each step $t$, which only requires an un-normalized posterior proportional to the likelihood $p(\mathbf{o}_t|\mathbf{z}_t)$ and the prior dynamics $p(\mathbf{z}_t|\mathbf{z}_{t-1})$. The algorithm iteratively applies three steps to a set of $N$ particles $\mathbf{z}^i$: *(i)* Propagate each particle using the model's dynamics $p(\mathbf{z}_t|\mathbf{z}_{t-1})$; *(ii)* Weight each resulting partial trajectory $\mathbf{z}_t^i$ using a reward function $g(\mathbf{z}_t^i)$ that outputs a score $r_t^i$; and *(iii)* Resample the particles with replacement from a normalized distribution, typically following a softmax distribution: $w^i \propto \exp(r_t^i)$.

## 3. Method

Our method builds on Particle Filtering for posterior estimation in LLMs (Puri et al., 2025; Feng et al., 2024).

Our goal is to address its tendency toward premature exploitation and overconfidence, particularly when guided by a PRM (Park et al., 2025). This issue is especially problematic for learning-free ITS methods that rely on frozen models and external feedback, as offline calibration is often infeasible or expensive.

We model sequential mathematical reasoning as a forward generative process over a sequence of latent states $\mathbf{z}_{1:T}$ conditioned on an input task $\mathbf{c}$. At each step $t$, a language model parameterized by $\theta$ defines the transition distribution:

$$\mathbf{z}_t \sim p_\theta(\mathbf{z}_t|\mathbf{z}_{1:t-1}, \mathbf{c}), \tag{3}$$

producing a trajectory of intermediate reasoning steps. A process reward model (PRM) provides feedback by assigning a scalar reward:

$$r(\mathbf{z}_{1:t}, \mathbf{c}) \approx p(\mathbf{o}_t = 1|\mathbf{z}_{1:t}, \mathbf{c}), \tag{4}$$

which we treat as an un-normalized log-likelihood of correctness for state $\mathbf{z}_t$. Our goal is to approximate the posterior over trajectories $p(\mathbf{z}_{1:T}|\mathbf{o}_{1:T}, \mathbf{c})$, proportional to:

$$p(\mathbf{z}_1|\mathbf{c})\prod_{t=2}^{T} p(\mathbf{o}_t|\mathbf{z}_{1:t}, \mathbf{c})\, p_\theta(\mathbf{z}_t|\mathbf{z}_{1:t-1}, \mathbf{c}) = p(\mathbf{o}_{1:T}, \mathbf{z}_{1:T}|\mathbf{c}), \tag{5}$$

where we set $p(\mathbf{o}_1|\mathbf{z}_1, \mathbf{c}) = 1$. We represent this posterior with $N$ weighted particles $\{(\mathbf{z}_t^i, w_t^i)\}_{i=1}^{N}$. Standard particle filtering alternates between: *(i)* forward propagation using $p_\theta$, *(ii)* weighting by $r(\mathbf{z}_{1:t}^i, \mathbf{c})$, and *(iii)* resampling particles proportionally to their normalized weights, $w_t(r_t)$.

The final decoded output is denoted by $\mathbf{x}$, which is extracted from the final state sequence $\mathbf{z}_{1:T}$. We now introduce the

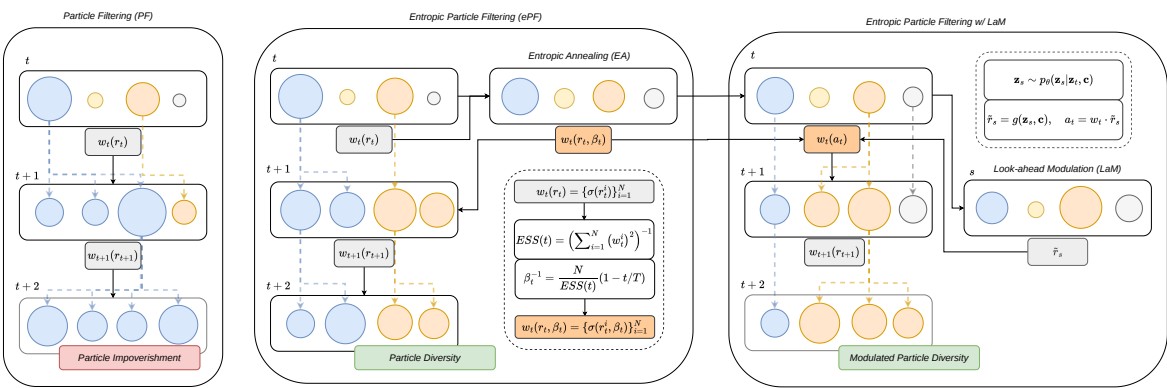

*Figure 3.* The Entropic Particle Filtering (ePF) pipeline and its core Mechanisms. Particle Filtering (left), Entropic Particle Filtering (center), and Entropic Particle Filtering w/ LaM (right). Each circle represents a particle at step $t$ and the size is proportional to the reward provided by the PRM. EA and LaM help the PF algorithm to mitigate early exploitation and myopic updates, greatly improving the diversity of the particles at step $t + 1$. Standard PF often suffers from *particle impoverishment*, where diversity is lost after resampling. Our ePF pipeline incorporates the EA step to maintain particle diversity. ePF with LaM, further adds the LaM step to guide the search more effectively. Pipeline details in Appx H.

two main building blocks of our method: *Entropic Annealing* and *Look-ahead Modulation*.

## 3.1. Why Particle Filtering Collapses

Ideally, resampling concentrates computation on promising particles while retaining diversity. In practice, however, PF often *collapses* early: a few particles acquire nearly all the weight mass, leading to *particle impoverishment* (Fig. 3). This effect for particle diversity can be quantified using the normalized entropy $H_n(t)$ and the normalized effective sample size $ESS_n(t)$:

$$
H_n(t) = -\frac{\sum_{i=1}^{N} w_t^i \log w_t^i}{\log N},
$$
$$
ESS_n(t) = \frac{\left(\sum_{i=1}^{N} (w_t^i)^2\right)^{-1}}{N}. \tag{6}
$$

Low $H_n(t)$ or $ESS_n(t)$ indicates that only a small subset of particles are being explored. In multi-step reasoning, this collapse often occurs prematurely because PRM scores at early steps are overconfident despite being weakly informative (Figure 2a, Figure 9).

As a result, PF commits to trajectories prematurely, under-exploring paths that may lead to correct, high-reward solutions and thus reducing its overall success probability. This behavior can be quantified by the variance of the resampling weights (Liu, 1996; Kong, 1992), $\mathbb{V}[w_t]$, which becomes excessively high early in the sampling process (Fig. 2b). High variance indicates that the weight distribution is concentrated on only a few particles, leading to a low $ESS(t)$, a low-entropy state, and ultimately a poor approximation of the posterior in Eq. 5. Details and derivation in Appx B.

## 3.2. Entropic Particle Filtering (ePF)

To mitigate premature collapse, we introduce *Entropic Annealing (EA)*, which adaptively modulates the resampling temperature $\beta_t^{-1}$ based on particle diversity:

$$
\beta_t^{-1} = \frac{N}{ESS(t)}(1 - t/T),
$$
$$
w_t^i(r_t, \beta_t) = \frac{\exp(r_t^i \cdot \beta_t)}{\sum_{j=1}^{N} \exp(r_t^j \cdot \beta_t)}, \tag{7}
$$

where $ESS(t) = \left(\sum_{i=1}^{N} (w_t^i)^2\right)^{-1}$. When particle diversity is low, the temperature increases, producing a flatter resampling distribution $w_t^i$ that maintains exploration (Fig. 4). As $t \to T$, $\beta_t$ gradually anneals to 1, shifting the algorithm from exploration toward exploitation.

By dynamically increasing the temperature when particle diversity is low, EA effectively reduces the variance of the resampling weights, preventing the particle set from collapsing prematurely. We also explored additional schedules: linear and entropy-based and conclude that ESS-based schedule provides a more direct response to particle impoverishment (Appx G.10).

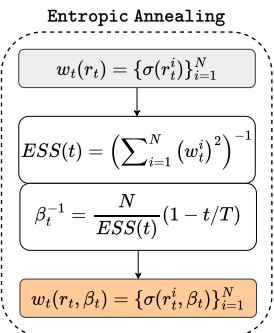

*Figure 4.* Adapting the resampling distribution temperature using Entropic Annealing.

We employ *systematic resampling* (Kitagawa, 1996), a low-

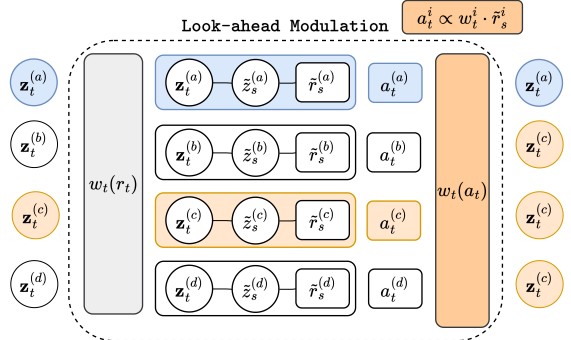

*Figure 5.* 1-step forward-looking resampling distribution update using Look-ahead Modulation. We use the dynamics model at step $t$ to predict an intermediate next state $s$.

variance sampling technique that uses a single stratified draw to mitigate early estimation errors common in multinomial resampling. This choice better preserves the distribution's structure and reduces random fluctuations, ensuring a more faithful representation for robust exploration. Details in Appx C.

### 3.3. Look-ahead Modulation (LaM)

While EA preserves exploration, PF remains fundamentally myopic: resampling decisions depend only on current rewards. To address this, we introduce *Look-ahead Modulation (LaM)*, which adjusts resampling weights using predicted successor quality before resampling. LaM is a novel, computationally efficient adaptation of the principles of the APF framework (Pitt &Shephard, 1999a) for language modeling inference. For each particle $i$ at step $t$, we sample a one-step look-ahead $\mathbf{z}_s^i \sim p_\theta(\mathbf{z}_s^i|\mathbf{z}_t^i, \mathbf{c})$, score it with the PRM to obtain $\tilde{r}_s$, and compute modulated weights:

$$a_t^i = w_t^i(r_t) \cdot \tilde{r}_s^i, \quad w_t^i(a_t) = \frac{a_t^i}{\sum_{j=1}^N a_t^j}. \quad (8)$$

These forward-looking weights bias resampling toward particles that are likely to produce high-reward successors, making the update less myopic. Crucially, look-ahead states $\mathbf{z}_s$ are discarded after modulation, so the forward model remains faithful to the propagation dynamics. While LaM introduces a computational overhead of one extra forward pass per particle before resampling, the cost is often offset by significant performance gains, achieving comparable accuracy with smaller particle budgets (see Fig. 21).

Algorithm 2 and 3 summarizes the combined procedure. Together, EA and LaM transform PF into a guided search that *(i)* maintains exploration until sufficient information has been gathered, and *(ii)* directs computation toward promising trajectories, achieving a better exploration–exploitation balance. We provide an analysis of the computation overhead for LaM in Appx D.

## 4. Experiments

**Benchmarks and Models**  We conduct our evaluation on six math benchmarks of increasing difficulty: GSM8K (Lightman et al., 2023), MATH500 (Hendrycks et al., 2024), DEEPMATH (He et al., 2025), OMNI-MATH (Gao et al., 2024), and the challenging AIME-2024 and AIME-2025 datasets. The problem of early exploitation becomes more relevant and detrimental for hard benchmarks (AIME), given that the sampling trajectories tend to have an order of magnitude more steps than for easier benchmarks (MATH500). The four primary models used are generalist models from the Qwen family (Yang et al., 2024; 2025), including Qwen2.5-1.5B-Instruct, Qwen2.5-7B-Instruct, Qwen3-0.6B and Qwen3-1.7B. All generation processes are guided by the same Process Reward Model (PRM), Qwen2.5-Math-PRM-7B (Puri et al., 2025; Park et al., 2025).

**Baselines and Setup**  We compare our algorithms against a strong set of baselines, including Self-Consistency with majority voting, weighted and unweighted Best-of-N, Beam Search, and standard Particle Filtering. For all guided methods, the generation is managed by a vLLM sampler with predefined budgets of $N \in \{2, 4, 8, 16, 32\}$, a maximum of 300 generation steps, and a limit of 512 tokens per step. We use ePF for the first 50 % of the steps in the sampling trajectory; we activate ePF (w/o and w/ LaM) with a threshold of $ESS_n(t) \leq 0.5$.

**Evaluation and Verification**  We assess performance primarily using Top-1 accuracy and pass@1 as our main evaluation metrics, which allow direct comparison with state-of-the-art methods in iterative test-time search. For output verification and parsing, we use the `math_verify` library[1], a deterministic and restrictive verifier chosen to provide a conservative estimate of performance. The final solution from each run is selected using the same PRM that provides intermediate guidance.

### 4.1. General Results on Mathematical Benchmarks

We start by evaluating `ePF` against leading inference-time scaling algorithms on a suite of four math benchmarks with increasing difficulty (GSM8K, MATH500, DEEPMATH, and OMNIMATH). Our goal was to determine if preventing premature exploitation translates to better reasoning performance. As shown in Table 1, `ePF` consistently performs well, achieving the highest accuracy across most configurations for both Qwen2.5-1.5B-Instruct and Qwen2.5-7B-Instruct models. This preliminary experiment validates our core hypothesis: by intelligently mitigating early exploitation, `ePF` consistently boosts a language model's ability to

---

[1]`https://github.com/huggingface/Math-Verify`

*Table 1.* Top-1 accuracy comparison of inference-time scaling algorithms on mathematical reasoning benchmarks with increasing complexity. Our proposed method, ePF, demonstrates superior performance over established baselines across multiple datasets of increasing difficulty for Qwen2.5-1.5B-Instruct and Qwen-2.5-7B-Instruct models. Best results are in **bold**. We use random subsets of 128 samples for each dataset. ORM: Output Reward Model; PRM: Process Reward Model; MV: Majority Voting.

| Algorithm | Selection | Scoring | Qwen2.5-1.5B-Instruct | | | | Qwen2.5-7B-Instruct | | | |
|---|---|---|---|---|---|---|---|---|---|---|
| | | | GSM8K | MATH500 | DEEPMATH | OMNIMATH | GSM8K | MATH500 | DEEPMATH | OMNIMATH |
| Base Sampling | - | - | 67.19 | 45.31 | 10.15 | 5.46 | 93.40 | 60.93 | 23.43 | 8.59 |
| Self-Consistency | MV | - | 82.03 | 53.90 | 13.28 | 7.03 | 94.80 | 65.62 | 30.46 | 9.37 |
| Best-of-N | Argmax | ORM | 92.97 | 57.81 | 20.31 | **10.15** | 96.00 | 67.96 | 32.03 | 9.37 |
| Beam-Search | Argmax | PRM | 91.40 | 62.50 | 21.09 | 9.37 | **96.20** | 66.40 | 32.03 | **10.93** |
| PF | Argmax | PRM | **93.75** | 60.15 | 22.65 | 8.59 | **96.20** | 70.31 | 34.37 | 10.15 |
| ePF (ours) | Argmax | PRM | **93.75** | **66.42** | **25.00** | **10.15** | 95.80 | **71.09** | **35.93** | **10.93** |

solve mathematical problems.

Fig. 20 details how performance scales with the particle budget $N \in \{2, 4, 8, 16, 32\}$. Our goal was to assess if ePF's advantage over standard PF grows with task complexity. While ePF consistently matches or outperforms PF, its superiority becomes most apparent on more challenging problems. Specifically, the performance gap widens significantly as dataset complexity increases (MATH500 → DEEPMATH) and model size grows (1.5B → 7B). ePF's exploration becomes more useful for more challenging problems, confirming its value in navigating vast and complex solution spaces.

### 4.2. Exploration for Hard Problems and Small Budgets

We now shift our focus on the AIME benchmarks. Our goal is to understand how different particle budgets impact exploration for long multi-step mathematical reasoning trajectories. In Table 2 and Fig. 6, ePF consistently outperforms established baselines and standard PF across most budgets. Its effectiveness is especially clear on the AIME 2025 benchmark with the Qwen2.5-7B model, where it achieves up to a 28.8% Top-1 rate across budgets. Crucially, its top performance under the inversely weighted (iw) metric highlights its superior sample efficiency at small budgets. ePF's enhanced exploration strategy provides a significant advantage to solve difficult mathematical problems.

### 4.3. Inference-Time Scaling with Small Reasoning Models

The rise of models trained on reasoning traces (Jaech et al., 2024; Guo et al., 2025), which learn to perform internalized search and backtracking, raises a key question: is parallel inference-time scaling still necessary? While training for reasoning is powerful, our results in Table 3 show that ITS is a highly effective complementary approach.

Applying the ePF algorithm, a Qwen3-1.7B model without thinking can elevate its performance to rival or even surpass specialized reasoning models. For instance, with a 12k sequence budget, ePF boosts the standard Qwen3-1.7B

model to a score of 38.9, outperforming dedicated models like R1-distilled-Qwen-1.5B and Nemotron-R-1.5B. This highlights ePF's strength: it effectively utilizes a parallel budget to explore longer solution paths, unlike standard PF which converges prematurely. This confirms that parallel search remains a crucial technique for maximizing a model's reasoning potential.

### 4.4. Guidance with Look-ahead Modulation

The previous experiments established that ePF is an effective mechanism for enhancing base models. We now study LaM and focus our attention on the Qwen3 series without thinking mode. To overcome the inherent myopia of the particle filtering algorithm, we introduce Look-ahead Modulation (LaM). LaM incorporates a predictive estimate of the next state's value into the current resampling step. This guidance provides a significant performance boost to our ePF framework, particularly on complex reasoning tasks, as shown in our results in Table 4 and Fig. 7. ePF w/ LaM outperforms all the baselines over datasets, model size, and budget re-weighting, providing strong evidence that a non-myopic resampling, leveraging a relatively cheap single step look-ahead, is an effective mechanism to improve performance. In Fig. 1 and 9 we provide intrinsic metrics to characterize ePF w/ LaM. By re-weighting particles based on this forward-looking signal, LaM steers the search toward trajectories with higher long-term potential, rather than those that are only locally optimal, proving crucial for success on complex reasoning tasks. In Appx E, where we provide an iso-computational analysis, showing that ePF w/ LaM achieves comparable performance with ePF on AIME 2024 with 1/4 and 1/2 of the particle budget.

### 4.5. Characterizing Guided-Search with Intrinsic Metrics

Thus far, our evaluation has focused on benchmarking our proposed method against the state-of-the-art in ITS, primarily using Top-1 and pass@1 accuracy. While these metrics are key indicators of model performance, they offer little insight into the solution-finding process or the diversity of

*Table 2.* Entropic Particle Filtering (`ePF`) outperforms baselines on AIME math benchmarks. The table shows aggregate Top-1 accuracies (%) across budgets ($N \in \{2, 4, 8, 16, 32\}$), reweighted to favor large (proportional weighting `w`), uniform (`u`), or small (inverse weighting `iw`) budgets. Averaged over 5 runs; higher is better.

| | AIME 2024 | | | | | | AIME 2025 | | | | | |
| | Qwen2.5-1.5B-In | | | Qwen2.5-7B-In | | | Qwen2.5-1.5B-In | | | Qwen2.5-7B-In | | |
| | w | u | iw | w | u | iw | w | u | iw | w | u | iw |
|---|---|---|---|---|---|---|---|---|---|---|---|---|
| Base Sampling (Yang et al., 2025) | - | 3.33 | - | - | 10.00 | - | - | 3.33 | - | - | 6.66 | - |
| Best-of-N | 9.90 | 7.20 | 4.00 | 23.09 | 20.20 | 17.48 | 5.13 | 3.60 | 2.52 | 17.41 | 15.80 | 14.90 |
| Beam-Search | 10.29 | 8.40 | 5.55 | 17.93 | 14.00 | 11.26 | 9.45 | **7.40** | **4.32** | 14.19 | 16.20 | 16.81 |
| PF (Puri et al., 2025) | 11.16 | 9.00 | **6.99** | **26.06** | **21.60** | **18.13** | 7.32 | 4.50 | 2.87 | 21.61 | 19.80 | 17.61 |
| `ePF` (ours) | **17.06** | **11.20** | 5.55 | **26.23** | 21.00 | 16.06 | **10.82** | 7.28 | 3.42 | **28.83** | **25.10** | **21.96** |

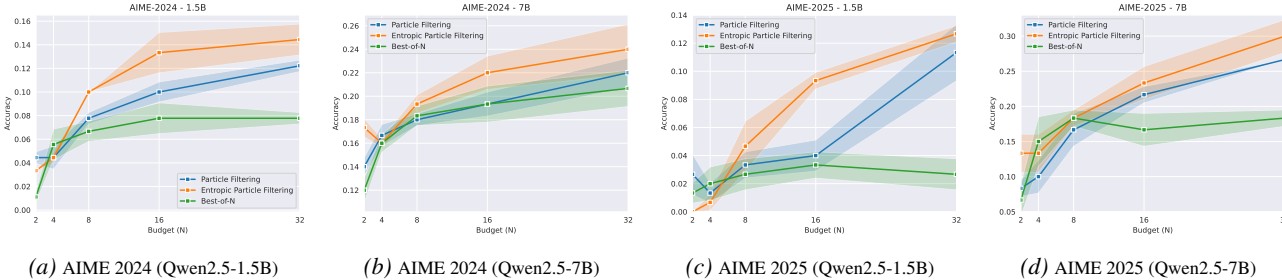

| *(a)* AIME 2024 (Qwen2.5-1.5B) | *(b)* AIME 2024 (Qwen2.5-7B) | *(c)* AIME 2025 (Qwen2.5-1.5B) | *(d)* AIME 2025 (Qwen2.5-7B) |

*Figure 6.* Top-1 accuracy as a function of inference budget ($N$) for Qwen2.5-1.5B-Instruct and Qwen-2.5-7B-Instruct models. The curves illustrate that `ePF` (orange) not only reaches a higher peak performance but also shows a steeper initial climb, indicating superior efficiency compared to PF (blue) and BoN (green), especially at smaller computation budgets. See Fig. 24 for more results and longer sequence length.

*Table 3.* Reasoning and ITS on AIME 2025. `ePF` effectively leverages an increased sequence length budget (4k → 12k) to dramatically boost the Qwen3-1.7B's performance (without thinking), while baselines methods and PF show limited gains. This demonstrates that an efficient search algorithm can elevate a non-thinking model to match specialized reasoning models, highlighting the power of ITS. Higher scores are better.

| Model | Budget/Seq | Algorithm | Thinking | AIME 2025 |
|---|---|---|---|---|
| Qwen3-1.7B | 32k | Reasoning | w/ | 35.5 |
| R1-Qwen-1.5B | 32k | Reasoning | w/ | 23.1 |
| Nemotron-R-1.5B | 32k | Reasoning | w/ | 33.6 |
| Qwen3-1.7B | 32k | Reasoning + BoN | w/ | 41.1 |
| e3-1.7B | 32k | Reasoning + Tuning | w/ | **43.8** |
| Qwen3-1.7B | 4k | CoT | w/o | 6.66 |
| Qwen3-1.7B | 4k | BoN | w/o | 13.3 |
| Qwen3-1.7B | 4k | PF | w/o | **20.0** |
| Qwen3-1.7B (ours) | 4k | `ePF` | w/o | 18.9 |
| Qwen3-1.7B | 12k | CoT | w/o | 16.6 |
| Qwen3-1.7B | 12k | BoN | w/o | 28.8 |
| Qwen3-1.7B | 12k | PF | w/o | 26.6 |
| Qwen3-1.7B (ours) | 12k | `ePF` | w/o | **38.9** |

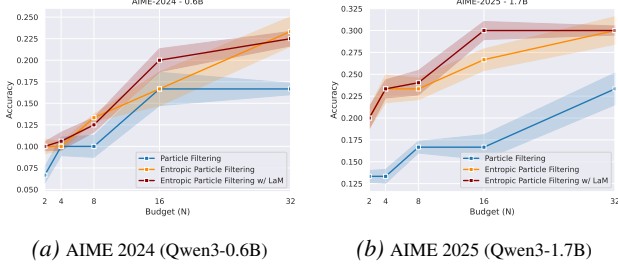

| *(a)* AIME 2024 (Qwen3-0.6B) | *(b)* AIME 2025 (Qwen3-1.7B) |

*Figure 7.* AIME 2024 and AIME 2025 results with Qwen3-0.6B and Qwen3-1.7B w/o thinking mode for `ePF` and `ePF` w/ LaM. Look-ahead Modulation boosts `ePF` and outperforms PF.

observation likelihood.

Our approach reframes guided decoding as posterior inference (Eq. 5), motivated by the core hypothesis that *the path to a solution is an essential component of the solution itself.* Under this view, analyzing the sampling trajectory through these intrinsic metrics is key to understanding how `ePF` balances guided search and exploration.

The aggregate measures, shown in Fig. 1, 8, and 9, provide strong evidence for our central hypothesis. For instance, Fig. 8 reveals a striking difference in behavior: PF converges on a narrow distribution of shorter answers, whereas `ePF` produces a bimodal distribution, indicating it explores both simple solutions and significantly longer, more complex

the solutions generated.

To better understand the algorithmic behavior of ePF, we now examine several intrinsic metrics: the final output reward, step-by-step trajectory reward, step-wise variance and effective sample size of the resampling distribution, and sequence length. Leveraging the probabilistic interpretation of particle filtering algorithms allows us to study these metrics that characterize the entire generation process and

*Table 4.* Look-ahead Modulation boosts performance on the AIME math benchmarks. We compare entropic particle filter (ePF) with and without Look-ahead Modulation (ePF w/ LaM) against baselines using Qwen3 models (no thinking). The table shows aggregate Top-1 scores (%) across computational budgets ($N \in \{2, 4, 8, 16, 32\}$), reweighted to favor large (proportional weighting w), uniform (u), or small (inverse weighting iw) budgets. Averaged over 5 runs; higher is better.

| | AIME 2024 | | | | | | AIME 2025 | | | | | |
| | Qwen3-0.6B | | | Qwen3-1.7B | | | Qwen3-0.6B | | | Qwen3-1.7B | | |
| | w | u | iw | w | u | iw | w | u | iw | w | u | iw |
|---|---|---|---|---|---|---|---|---|---|---|---|---|
| Base Sampling (Yang et al., 2025) | - | 3.40 | - | - | 13.40 | - | - | 2.60 | - | - | 9.80 | - |
| Best-of-N | 7.90 | 6.00 | 4.32 | 20.51 | 18.80 | 17.12 | 17.83 | 15.80 | 11.54 | 19.22 | 18.40 | 18.45 |
| PF (Puri et al., 2025) | 14.46 | 12.20 | 9.38 | 20.45 | 20.40 | 20.45 | 20.25 | 16.40 | 12.61 | 19.10 | 18.60 | **19.09** |
| ePF (ours) | 14.59 | 11.40 | 8.79 | 28.83 | 25.00 | 21.96 | 22.06 | 18.40 | 13.35 | 23.13 | 20.80 | 18.38 |
| ePF w/ LaM (ours) | **17.66** | **13.50** | **9.67** | **29.13** | **25.60** | **23.13** | **25.16** | **19.60** | **13.55** | **26.97** | **22.99** | 19.03 |

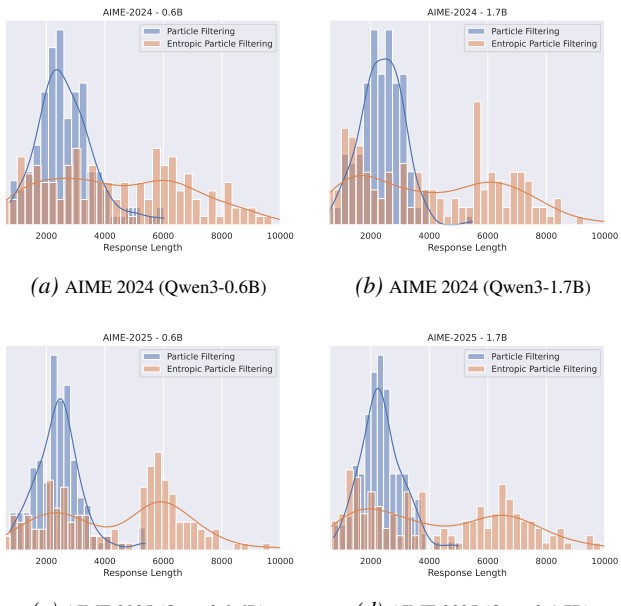

*(a)* AIME 2024 (Qwen3-0.6B)    *(b)* AIME 2024 (Qwen3-1.7B)

*(c)* AIME 2025 (Qwen3-0.6B)    *(d)* AIME 2025 (Qwen3-1.7B)

*Figure 8.* Distribution Response Length using Particle Filtering and Entropic Particle Filtering with Qwen3-0.6B Qwen3-1.7B w/o thinking mode on AIME 2024 and AIME 2025. Both algorithms can use up to 12k tokens for each response. Notice how PF tends to generate shorter answers and converge to local solutions, where ePF explore the search space more, and converges to better solutions.

reasoning paths that would otherwise be pruned.

This commitment to exploration is further evidenced by the step-reward curves in Fig. 9. Notably, ePF's average step-reward initially dips below that of PF as entropic annealing actively forces the search away from greedy, high-reward initial steps (third column Fig. 9). This early investment in exploration allows the algorithm to discover superior, high-reward trajectories later on, ultimately leading to higher overall task rewards and better final solutions (first and second columns Fig. 9). The variance of the resampling distribution can also be used to quantify PF exploitation versus ePF exploration (Fig. 11). See Appx G.12 for a

solution diversity analysis.

**Iso-Computational Cost of LaM**  Fig. 15 evaluates if LaM's performance justifies its overhead. While worst-case bounds suggest standard ePF could afford $1.5\times$ to $2\times$ more particles for the same cost, LaM's actual overhead is minimal since it only triggers in 10–12% of steps. Empirically, LaM is highly compute-efficient: on AIME 2024, an 8-particle ePF with LaM matches a 32-particle standard ePF (Qwen2.5-1.5b). This proves predictive guidance is significantly more effective than simply spending equivalent compute to scale standard ePF particles (details in Appendix E).

**Ablations**  We conducted extensive ablation studies to validate our proposed method in Appx G. As shown in Table 15 and Table 16, ePF outperforms state-of-the-art baselines, surpassing TSMC (Feng et al., 2024) on the MATH500 and GSM8K benchmarks, and IAS-C (Park et al., 2025) on the combined AIME-24-25 dataset without offline calibration and PRM tuning. Further analyses isolate the contributions of key components, assess generation diversity (Fig. 30), and test various configurations, including alternative LLaMA backbones (Fig. 26), and different temperature and resampling schedules (Fig. 28).

**Qualitative**  Furthermore, the qualitative example in Appx J offers a concrete and tangible illustration of this process, contrasting a failed 3200 token attempt by PF with a successful 5400 token solution found by ePF, making the abstract concept of mitigating early exploitation concrete.

## 5. Related Work

We highlight key related work below, with an extended discussion in Appendix A.

**Inference-Time Scaling**  Existing search methods struggle to balance diversity and compute. Beam search variants (Vijayakumar et al., 2016; Snell et al., 2024) often yield low-diversity solutions, while exploration methods

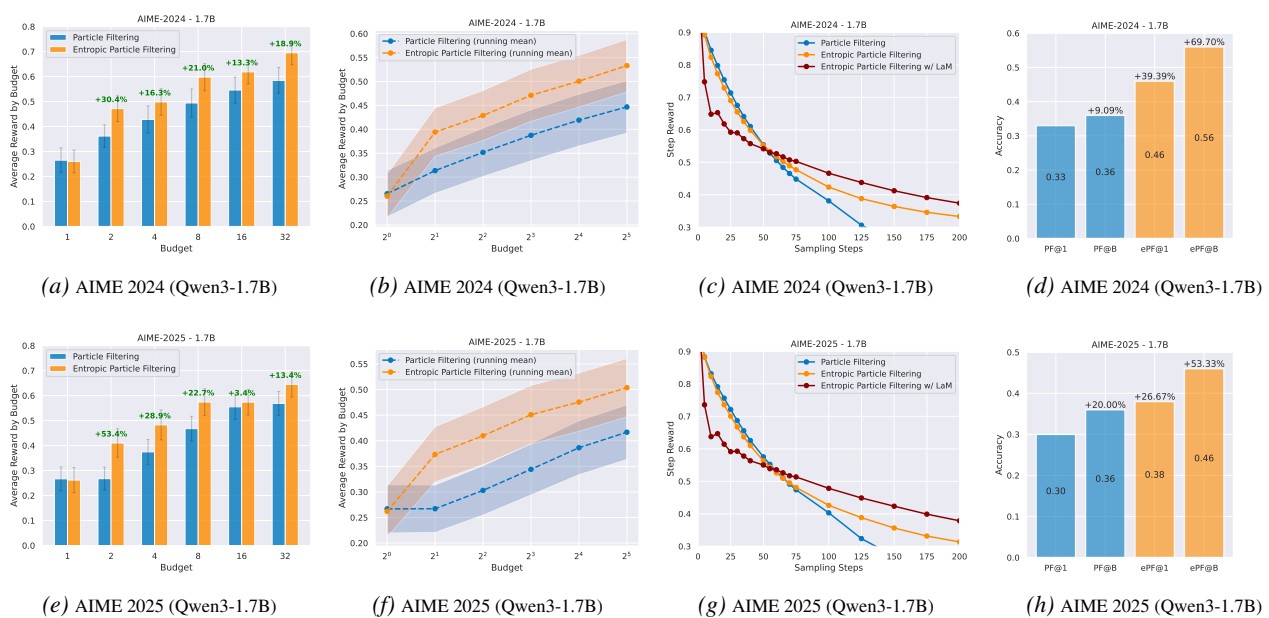

*Figure 9.* Intrinsic Metrics Analysis of `ePF` on AIME 2024 and 2025 benchmarks with Qwen3-1.7B. The plots illustrate how `ePF` balances exploration and exploitation in the sampling process. *(Third Column)*: Step-wise rewards initially dip (first 50/60 steps) as entropic annealing forces exploration, but this allows the algorithm to discover superior, high-reward regions later on. While not directly reward-seeking, this novelty-driven search leads to higher overall task rewards *(First and Second Column)* and a more diverse set of correct solutions across budgets *(Fourth Column)*, effectively mitigating premature convergence and early exploitation.

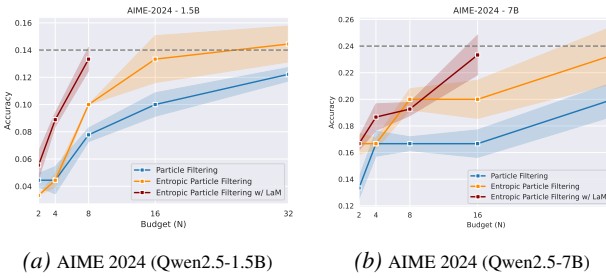

*Figure 10.* AIME-24 performance for ePF with and without LaM. ePF w/ LaM reaches comparable performance to standard ePF ($N = 32$) using significantly fewer particles (8 particles for the 1.5B model, and 16 for the 7B model), demonstrating superior computational efficiency.

## 6. Limitations and Conclusion

This paper introduces *Entropic Particle Filtering* (ePF), a guided search algorithm that effectively mitigates premature convergence in language models by balancing exploration and exploitation. Through *Entropic Annealing* (EA) and *Look-ahead Modulation* (LaM), our method preserves hypothesis diversity and incorporates less-myopic guidance, leading to significant performance gains on complex reasoning tasks under tight computational budgets.

**Limitations**    *ePF* advantage is most significant with small computational budgets and diminishes as the number of particles increases. Second, Look-ahead Modulation introduces computational overhead due to its extra forward pass per resampling step. Finally, the method's effectiveness is constrained by the quality of the reward model; ePF can mitigate overconfidence but not a consistently inaccurate signal. In conclusion, ePF offers a principled and effective strategy for improving inference-time search. By promoting robust exploration and forward-looking guidance, our methods enable the discovery of higher-quality solutions in complex domains.

like DVTS (Beeching et al., 2024) under-perform on small budgets. Monte Carlo Tree Search (MCTS) (Silver et al., 2016) is computationally heavy and requires reliable value functions, making it challenging for long-horizon math. Finally, standard Particle Filtering (PF) and Sequential Monte Carlo (SMC) (Feng et al., 2024; Zhao et al., 2024) require careful per-task tuning and remain vulnerable to early over-commitment.

**Particle-based Monte Carlo**    Auxiliary Particle Filters (APF) (Pitt &Shephard, 1999a;b) tackle the myopic resampling of standard PFs by introducing a look-ahead mechanism. This biases the resampling distribution toward particles predicted to reach more promising future states.

## Impact Statement

This work advances the field of Inference-Time Scaling (ITS) by improving the reliability and efficiency of language models in complex reasoning tasks. By mitigating particle impoverishment, our method allows models to reach higher-quality solutions under constrained computational budgets. This has positive implications for the deployment of accessible AI systems in resource-limited environments and scientific applications requiring high precision. However, as with all advancements in ITS and reasoning capabilities, there is a potential risk that these methods could be adapted to enhance the effectiveness of malicious agents in generating complex, harmful plans or code. We believe the benefits of robust, verified reasoning in benign applications outweigh these risks, but continuous monitoring of ITS deployment is advisable.

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

## Reproducibility statement

Particle Filtering (PF), Entropic Particle Filtering (ePF), and ePF with Look-ahead Modulation (LaM), are detailed with pseudo code in Appendix I. The experimental setup, including the models, datasets, baselines, evaluation metrics, and key hyperparameters, is described in the Method section and Appendix K. Comprehensive results and extensive ablation studies supporting our claims are presented throughout the main paper and are further detailed in Appendix G. The baselines, algorithms, and experiments are implemented using `its_hub`[2] and `reward_hub`[3]. We use `Math-Verify`[4] as math verifier.

---

[2]`https://github.com/Red-Hat-AI-Innovation-Team/its_hub`
[3]`https://github.com/Red-Hat-AI-Innovation-Team/reward_hub`
[4]`https://github.com/huggingface/Math-Verify`

# Contents

# A. Extended Related Work

**Inference-Time Scaling in Language Modeling**   Inference-Time Scaling (ITS) is a class of techniques that improve generative models performance by dedicating more computation at test time (Wang et al., 2022; Wei et al., 2022; Brown et al., 2024; Snell et al., 2024; Beeching et al., 2024; Kang et al., 2025). These methods reframe generation as a guided search over a vast solution space, using either sequential self-refinement and linearized backtracking (Shinn et al., 2024; Yao et al., 2022; Jaech et al., 2024; Guo et al., 2025), or generating and evaluating multiple candidate solutions in parallel (Yao et al., 2023; Brown et al., 2024; Snell et al., 2024), performing a form of parallel search. This search includes many classic ITS algorithms, such as Self-Consistency (Wang et al., 2022; Chen et al., 2023), Best-of-N sampling (Brown et al., 2024; Kang et al., 2025; Amini et al., 2024), and more sophisticated procedures like Beam Search (Snell et al., 2024; Vijayakumar et al., 2016) and Sequential Monte Carlo methods (Lew et al., 2023; Zhao et al., 2024; Feng et al., 2024; Puri et al., 2025; Loula et al., 2025). These algorithms typically leverage either an external outcome verifier to score complete solutions (Lightman et al., 2023; Cobbe et al., 2021) or use process rewards to guide the step-by-step construction of a solution (Puri et al., 2025; Geuter et al., 2025). Ultimately, all ITS methods can be conceptualized as forms of guided search over the sampling space, effectively trading increased computational budget for higher-quality outputs (Wang et al., 2025a; Wu et al., 2025).

However, existing ITS methods have limitations and offer only partial solutions. Beam search and its diverse variants (Vijayakumar et al., 2016; Snell et al., 2024) can improve coverage but remain prone to low-diversity solutions and premature exploitation, while exploration-promoting methods like DVTS (Beeching et al., 2024) tend to under-perform for small budgets. Standard PF and Sequential Monte Carlo (SMC) approaches (Feng et al., 2024; Zhao et al., 2024) require careful per-task tuning of a twist function and remain vulnerable to early overcommitment under miscalibration. Monte Carlo Tree Search (MCTS (Coulom, 2006)) and tree-based search methods (Yao et al., 2023; Bi et al., 2024; Wang et al., 2025b; Inoue et al., 2025), while powerful in domains where exact simulation is cheap, like in game benchmarks (Silver et al., 2016), and step-level feedback is available or easy to construct, like in coding benchmarks (Wang et al., 2025b; Inoue et al., 2025), remain challenging to apply to long-horizon reasoning and mathematical tasks due to their computational complexity, the difficulty of training a reliable value function approximator (Park et al., 2025), and the sequential structure that is hard to parallelize on modern accelerators (Ding et al., 2025).

**Importance Sampling**   Importance Sampling (IS) is a fundamental Monte Carlo technique used across Statistics (Robert et al., 1999; Casella &Berger, 2024), Physics (Metropolis &Ulam, 1949; Metropolis et al., 1953; Hastings, 1970), and Engineering (Kalman, 1960; Kolmogorov, 1941) to address the common challenge of sampling from a complex target distribution $\pi(\mathbf{z})$ (Robert et al., 1999). This problem is especially prevalent in Bayesian inference, where the goal is to characterize the posterior distribution $p(\mathbf{z}|\mathbf{o})$ of a random variable $\mathbf{z}$ given an observation $\mathbf{o}$. Direct sampling from this posterior is often intractable due to high dimensionality or an unknown normalizing constant. However, we can typically evaluate the posterior up to this constant, as Bayes' theorem states $p(\mathbf{z}|\mathbf{o}) \propto p(\mathbf{o}|\mathbf{z})p(\mathbf{z}) = \tilde{p}(\mathbf{z}|\mathbf{o})$, where the evidence $Z_p = \int p(\mathbf{o}|\mathbf{z})p(\mathbf{z})d\mathbf{z}$ is the intractable component. IS circumvents this by drawing samples from a simpler proposal distribution $q(\mathbf{z})$, which is easy to sample from and evaluate, and whose support covers that of $p(\mathbf{z}|\mathbf{o})$. The expectation of a function $f(\mathbf{z})$ under the target can then be estimated by re-weighting samples from the proposal:

$$\mathbb{E}_{p(\mathbf{z}|\mathbf{o})}\left[f(\mathbf{z})\right] = \int p(\mathbf{z}|\mathbf{o})f(\mathbf{z})d\mathbf{z} = \int q(\mathbf{z})\frac{p(\mathbf{z}|\mathbf{o})}{q(\mathbf{z})}f(\mathbf{z})d\mathbf{z} = \mathbb{E}_{q(\mathbf{z})}\left[w(\mathbf{z})f(\mathbf{z})\right]. \tag{9}$$

The terms $w(\mathbf{z}) = p(\mathbf{z}|\mathbf{o})/q(\mathbf{z})$ are the importance weights that correct for the discrepancy between the proposal and target distributions. Since we can only evaluate $\tilde{p}(\mathbf{z}|\mathbf{o})$, we compute un-normalized weights $\tilde{w}(\mathbf{z}) = \tilde{p}(\mathbf{z}|\mathbf{o})/q(\mathbf{z})$. For a set of $N$ samples $\{\mathbf{z}_i\}_{i=1}^{N}$ drawn from $q(\mathbf{z})$, these are normalized to yield the estimator $w_i = \tilde{w}(\mathbf{z}_i)/\sum_{j=1}^{N}\tilde{w}(\mathbf{z}_j)$. While this estimator introduces a bias for finite $N$, it has lower variance and the estimator is consistent. The choice of proposal is critical; ideally, $q(\mathbf{z})$ should be close to the target, e.g., $q(\mathbf{z}|\mathbf{o})$. Notice that in principle we can set the proposal equal to the prior, $q(\mathbf{z}) = p(\mathbf{z})$. This simplifies the un-normalized weights to be the likelihood itself: $\tilde{w}(\mathbf{z}) = p(\mathbf{o}|\mathbf{z})p(\mathbf{z})/p(\mathbf{z}) = p(\mathbf{o}|\mathbf{z})$.

IS is a cornerstone of modern computational methods, forming the basis for MCMC sampling algorithms (Metropolis &Ulam, 1949), Variational Inference (Jordan et al., 1999; Kingma &Welling, 2019), Reinforcement Learning (Peters &Schaal, 2007; Roux et al., 2025; Schulman et al., 2017), and Monte Carlo Gradient Estimation (Mohamed et al., 2020; Schulman et al., 2015; Foerster et al., 2018).

**Extending Particle-based Monte Carlo**   Standard Sequential Monte Carlo (SMC(Doucet et al., 2001a;b; Chopin et al., 2020; Naesseth et al., 2019)) and Particle Filtering (PF) methods are inherently myopic, as they estimate the current state

distribution relying solely on present and past observations without anticipating future evidence. Consequently, these algorithms often suffer from sample impoverishment, as particles are propagated and resampled without the benefit of forward-looking information that could guide them toward high-likelihood regions emerging in later time steps.

Auxiliary Particle Filters (APF) introduce a look-ahead mechanism to mitigate the inefficiencies of standard myopic resampling (Pitt &Shephard, 1999a;b). APF algorithms bias the resampling process toward particles predicted to lead to more promising future states, thereby introducing a notion of forward-looking update for the resampling distribution.

Beyond single-step look-ahead for filtering, SMC methods have been adapted in the Reinforcement Learning (RL) literature to tackle long-horizon planning (Piché et al., 2018; Macfarlane et al., 2024). In this context, planning is often cast as probabilistic inference (Levine, 2018) where the goal is to sample trajectories proportional to their expected return. (Piché et al., 2018) leverage this control as inference perspective, proposing a SMC planner that iteratively re-weights and resamples action sequences based on future rewards, effectively performing a multi-step look-ahead search in continuous control tasks. (Macfarlane et al., 2024) introduced Sequential Monte Carlo Policy Optimization (SPO), which utilizes SMC as a highly parallelizable policy improvement operator within an Expectation-Maximization framework. Unlike tree-based search methods based on MCTS, SPO uses particle-based sampling to explore the policy space, generating high-reward trajectories that serve as targets for training a neural policy, thus offering a scalable alternative for planning in complex environments.

Recent research addresses particle impoverishment in SMC-based RL, a phenomenon where resampling reduces trajectory diversity and hinders exploration (Lioutas et al., 2022; de Vries et al., 2025). To mitigate this, (Lioutas et al., 2022) introduced critic-guided resampling, using value estimates to bias distributions toward high-likelihood regions while maintaining diversity. (de Vries et al., 2025) synthesizes ideas from MCTS and SMC, applying trust-region constraints to ensure that particle updates remain within plausible posterior regions, counteracting particle degeneracy and state collapse. These methods ground resampling in bayesian inference, improving state estimation for algorithms like actor-critic and bridging the gap between probabilistic inference and deep RL.

However, adapting these planning mechanisms for LLMs and extended reasoning traces proves challenging. Learning an accurate, calibrated value function to guide sampling remains difficult, while standard resampling steps often trigger particle collapse, causing candidate diversity to vanish before the planning horizon is reached. Integrating LLMs with RL is an active research area, and techniques like SMC and PF offer powerful ways to enhance these algorithms. See (Murphy, 2024) for an overview of probabilistic methods for language models and planning.

**Maximum Entropy Algorithms**   Maximum Entropy Algorithms (Jaynes, 1957; Ziebart, 2010; Neumann, 2011; Levine, 2018), seek to maximize a given score while maintaining the highest possible entropy, thus balancing exploitation with exploration. This concept is central to methods like Soft Actor-Critic (SAC (Haarnoja et al., 2018)) and is related to quality-diversity algorithms (Pugh et al., 2016; Mouret &Clune, 2015). For a given sampling distribution $q(\mathbf{z})$, these algorithms typically optimize the following objective:

$$\mathcal{F}(\mathbf{z}) = \mathbb{S}(\mathbf{z}) + \alpha \, \mathbb{H}(\mathbf{z}) \propto \mathbb{E}_{q(\mathbf{z})}\left[f(\mathbf{z})\right] - \mathbb{E}_{q(\mathbf{z})}\left[\log q(\mathbf{z})\right]. \tag{10}$$

Here, the scoring function $f(\mathbf{z})$ can represent rewards in Reinforcement Learning, where $f(\mathbf{z}) = r(\mathbf{z})$ (Williams &Peng, 1991; Mnih et al., 2016), or the log-likelihood in Variational Inference, where $f(\mathbf{z}) = \log p(\mathbf{z})$ (Jordan et al., 1999; Mohamed &Lakshminarayanan, 2016).

`ePF` is inspired by this framework. It aims to find a problem's solution by following a trajectory of high entropy, constrained by a reward signal defined by a likelihood model over observations, $p(\mathbf{o}|r(\mathbf{z}))$. In essence, we encourage a policy that acts as randomly as possible while still achieving high rewards. This approach enhances exploration and effectively mitigates the risk of premature convergence to suboptimal solutions.

**Premature Exploitation**   Premature convergence is a critical failure mode in evolutionary, reinforce-based, and control algorithms where the process becomes trapped in a suboptimal solution, fundamentally stemming from an imbalance in the exploration-exploitation trade-off (Kaelbling et al., 1996; Pandey et al., 2014; Lewis et al., 2012). This occurs when an algorithm prioritizes exploiting local good partial solutions over exploring uncharted regions of the search space, causing it to miss the global optimum or higher reward regions. In Particle Filtering, we can frame this issue as a form of poor state estimation in posterior inference, where premature convergence represents a mode collapse - the algorithm's belief about the optimal solution becomes too narrow and overconfident, stifling the discovery of novel, potentially superior solutions.

Mixing mechanisms, hybrid algorithms, and multiple chains have been explored in the MCMC literature to counteract such issues (Doucet et al., 2001b; Binder et al., 1992; Hastings, 1970).

A core challenge in complex multi-step problem solving is the tension between early exploration and exploitation. In engineering, over-optimization can lead to designs that are theoretically optimal on a single metric but fail on crucial, unmodeled objectives like manufacturability or robustness, limiting the diversity and quality of final designs (Chen et al., 2016; Ahmed et al., 2016).

Quality-Diversity (QD (Mouret &Clune, 2015; Chatzilygeroudis et al., 2021; Pugh et al., 2016)) algorithms shift the goal from finding a single best solution to finding a wide array of diverse, high-performing ones. Similar ideas are captured by entropy methods, particularly in Bayesian optimization (Wang &Jegelka, 2017; Hernández-Lobato et al., 2016; 2015), which use entropy as a measure of uncertainty to actively guide the search. By maximizing information gain - seeking out areas where the outcome is most uncertain - these methods ensure the algorithm maintains full coverage of the search space, effectively preventing over-optimization and ensuring a more robust and comprehensive search for the true best solution. Recently, language modeling and quality-diversity have been scaled to solve algorithmic code-based problems (Novikov et al., 2025; Lange et al., 2024; Lehman et al., 2023).

# B. Variance of the Resampling Distribution

This section explains how Entropic Particle Filtering (ePF) reduces the variance of the resampling distribution. By doing so, ePF increases the effective sample size, leading to a more thorough exploration of the sampling space and a more accurate estimation of the true state.

## B.1. Importance Weights

In importance sampling (Casella &Berger, 2024), we draw samples from a tractable proposal distribution, $q(\mathbf{z})$, to estimate properties of an intractable target distribution, $p(\mathbf{z})$. This is achieved by weighting each sample. The importance weight, $w(\mathbf{z})$, for a sample $\mathbf{z}$ is the ratio of the target to proposal densities:

$$w(\mathbf{z}) = \frac{p(\mathbf{z})}{q(\mathbf{z})}, \quad \text{where} \quad \mathbb{E}_{q(\mathbf{z})}[w(\mathbf{z})] = 1 \quad \text{and} \quad w(\mathbf{z}) \geq 0. \tag{11}$$

These weights allow us to estimate the expectation of any function $f(\mathbf{z})$ under the target distribution using $N$ samples $\{\mathbf{z}_i\}_{i=1}^N$ drawn from the proposal: $\hat{I} = \frac{1}{N}\sum_{i=1}^N w(\mathbf{z}_i)f(\mathbf{z}_i)$.

In a sequential context like particle filtering, these weights can be updated recursively. The unnormalized weight $\tilde{w}_t$ for a particle at time $t$ is calculated based on its weight at the previous step, the likelihood of the new observation $\mathbf{o}_t$, and the transition dynamics:

$$\tilde{w}_t = \tilde{w}_{t-1} \frac{p(\mathbf{o}_t|\mathbf{z}_t)\, p(\mathbf{z}_t|\mathbf{z}_{t-1})}{q(\mathbf{z}_t|\mathbf{z}_{t-1})}. \tag{12}$$

These quantities are then normalized to produce the final weights $w_t = \tilde{w}_t / \sum_{j=1}^N \tilde{w}_t^j$, which form the resampling distribution.

Entropic Particle Filtering operates on the principle that *robust exploration in the early stages of a guided search yields higher-quality solutions later on*. Since the importance weights guide the resampling process, their distribution is critical. The ePF hypothesis implies that, particularly early in the search, the variance of the importance weights should be kept low. A distribution closer to uniform encourages broader exploration of the search space, increases the effective sample size, and makes better use of the available particles.

## B.2. Particle Collapse: Degeneracy and Impoverishment

Particle collapse occurs when the particle filter loses its ability to approximate the posterior distribution. This failure can manifest in two primary ways: degeneracy and impoverishment.

*Particle degeneracy* occurs when most particles acquire negligible weights, often due to noisy observations or model mismatch. In this state, only a handful of particles contribute to the posterior estimate, wasting computational resources. Degeneracy is typically addressed by resampling, where particles are drawn with replacement from the current set, proportional to their weights.

However, the solution to degeneracy can lead to *particle impoverishment*. If the observation model is overly confident or uncalibrated, resampling may repeatedly select only a few high-weight particles. As these particles are cloned, the diversity of the particle set is drastically reduced. The entire population converges to a small, localized portion of the search space, potentially missing the true state entirely, generating a poor estimate of the posterior.

For instance, consider a system with just $N = 4$ particles and normalized weights $w = [0.96, 0.01, 0.02, 0.01]$. During resampling, the first particle is overwhelmingly likely to be selected multiple times. Propagating a state composed almost entirely of clones of this single particle severely restricts the diversity and quality of future solutions.

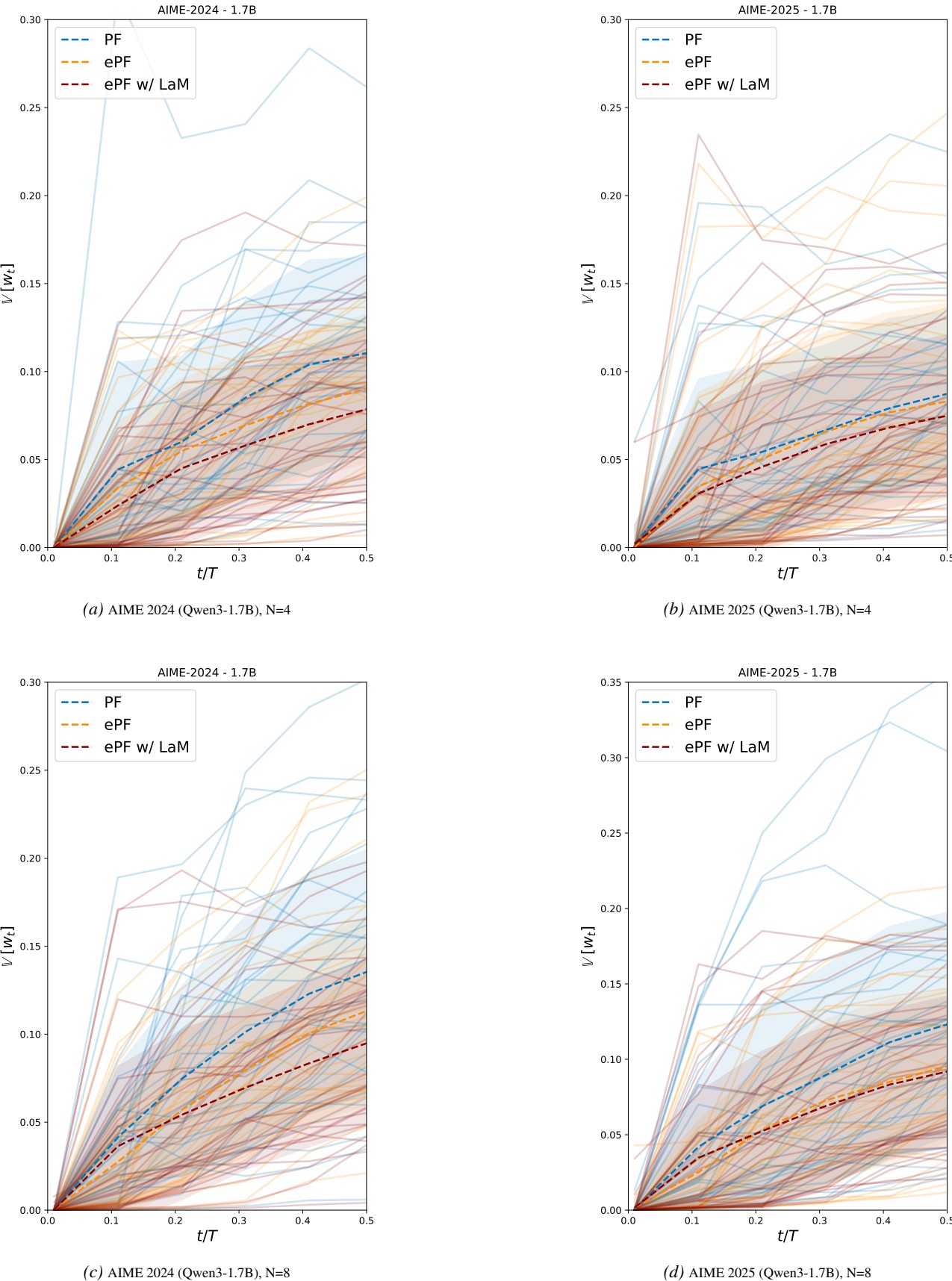

*Figure 11.* Running Variance of the resampling distribution weights over the first 50 steps for PF, ePF, and ePF with LaM, using $N \in \{4, 8\}$ particles on the AIME 2024 and 2025 dataset and Qwen3-1.7B as the sampler. Each line represents a single run. Standard PF (blue) exhibits a rapid increase in variance as a few particles quickly dominate the weight distribution. In contrast, ePF methods maintain lower variance, promoting greater particle diversity.

### B.3. The Variance of the Weights Distribution

The health of a particle filter can be quantified by the *Effective Sample Size (ESS)*, which measures the diversity of the particles. Following (Liu, 1996; Kong, 1992), the ESS at step $t$ for a particle filter with $N$ particles is inversely related to the variance of the normalized importance weights, $\mathbb{V}[w_t]$:

$$ESS(t) = \frac{N}{1 + \mathbb{V}[w_t]} \tag{13}$$

where the weights $w_t^i$ are typically computed via a softmax over log-rewards $r_t^i$ as $w_t^i = \exp(r_t^i)/\sum_{j=1}^{N} \exp(r_t^j)$. The sample variance of these weights can be estimated as:

$$\hat{\mathbb{V}}[w_t] \approx \frac{1}{N-1} \sum_{i=1}^{N} \left(w_t^i - \frac{1}{N}\right)^2 \tag{14}$$

This relationship makes it clear that a high variance in the weight distribution leads to a low effective sample size. As variance increases, one or more weights become large while others shrink, indicating that fewer particles are effectively contributing to the state representation. A more direct and widely used definition of ESS is (Doucet et al., 2001b; Martino et al., 2017):

$$ESS(t) = \frac{1}{\sum_{i=1}^{N} (w_t^i)^2} \tag{15}$$

This formulation is intuitive:

- Maximum ESS: If all weights are uniform ($w_t^i = 1/N$), then $ESS = 1/\sum(1/N)^2 = 1/(N/N^2) = N$. The effective size is the total number of particles.

- Minimum ESS: In the most degenerate case, one weight is 1 and all others are 0. Then $ESS = 1/(1^2) = 1$. The effective size has collapsed to a single particle.

Crucially, low variance in the resampling distribution at step $t$ corresponds to high diversity among the particles selected for propagation to step $t+1$.

### B.4. Entropic Annealing as a Variance Reduction Technique

High variance in the importance weights is a direct cause of particle impoverishment in Sequential Importance Resampling (SIR) methods. *Entropic Annealing (EA) directly mitigates this by functioning as a dynamic variance reduction technique* integrated into the resampling step.

The core principle is that the variance of a Monte Carlo estimator for the resampling distribution is minimized when the importance weights are uniform. EA operationalizes this by modulating the resampling distribution's "temperature" based on the ESS. When the ESS is low - indicating high weight variance and potential particle collapse - EA increases the entropy of the weight distribution. It does this by applying a temperature parameter, $\beta_t \leq 1$, to the log-rewards before the softmax calculation:

$$w_t^i = \frac{\exp(r_t^i \cdot \beta_t)}{\sum_{j=1}^{N} \exp(r_t^j \cdot \beta_t)} \tag{16}$$

Lowering $\beta_t$ (i.e., "heating up" the distribution) flattens the output of the softmax, pushing the weights closer to a uniform distribution. This action actively reduces the variance of the weights at each step in the early phase of sampling (Fig. 11). By preventing the particle set from collapsing onto a few high-reward hypotheses, this proactive management of weight variance ensures a more stable and robust approximation of the posterior distribution throughout the sequential filtering process.

High-variance in the resampling distribution will create a poor estimate for the state and the posterior. In the limit, high-variance over the importance weights will generated a deterministic state. In summary, the `ePF` algorithm is a form of variance reduction: limiting extreme overconfidence (see the two blue lines with large variance in Fig. 11) over few particles. Notice that the ESS-based schedule for a given step is closely related to the inverse of the variance, in particular:

$$\frac{\beta_t^{-1}}{t/T} - 1 = \frac{N}{ESS(t)} - 1 = \hat{\mathbb{V}}[w_t], \tag{17}$$

clearly showing that, for a given $t$, higher the variance, higher the temperature $\beta_t^{-1}$, flattening the resampling distribution and better exploring the state.

Notice that the variance at step $t$ for the importance sampling distribution can also be written as $\mathbb{V}[w_t] = \mathbb{E}[w_t^2] - \mathbb{E}[w_t]^2 \propto \mathbb{E}[w_t^2]$, showing that the quantity $\sum_{i=1}^{N}(w_t^i)^2$ is intrinsically related to the variance of the importance weight estimator.

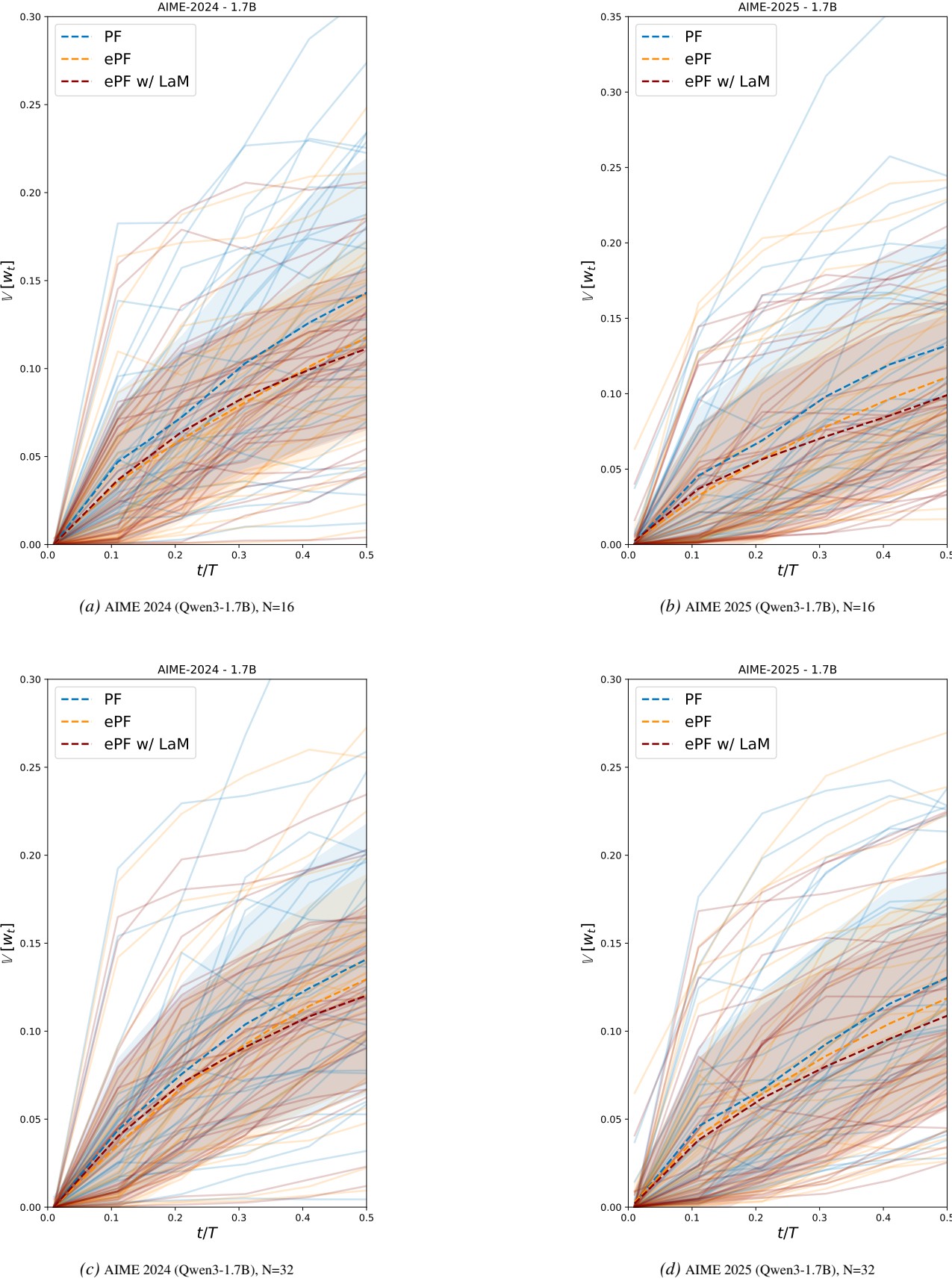

*(a)* AIME 2024 (Qwen3-1.7B), N=16

*(b)* AIME 2025 (Qwen3-1.7B), N=16

*(c)* AIME 2024 (Qwen3-1.7B), N=32

*(d)* AIME 2025 (Qwen3-1.7B), N=32

*Figure 12.* Running Variance of the resampling distribution weights over the first 50 steps for PF, ePF, and ePF with LaM, using $N \in \{16, 32\}$ particles on the AIME 2024 and 2025 dataset and Qwen3-1.7B as the sampler. Each line represents a single run. Standard PF (blue) exhibits a rapid increase in variance as a few particles quickly dominate the weight distribution. In contrast, ePF methods maintain lower variance, promoting greater particle diversity.

## C. Multinomial and Systematic Resampling

In particle filtering, both *multinomial* and *systematic resampling* aim to generate a new population of $N$ particles from an existing set based on their normalized weights $\{w_k\}_{k=1}^N$, thereby combating particle degeneracy. The two methods are conceptually similar but differ critically in their sampling strategy and resulting statistical properties.

Multinomial resampling (Casella & Berger, 2024) operates by making $N$ independent draws from a categorical distribution defined by the particle weights. For each of the $N$ new particle slots, a random number $u_i \sim U[0, 1)$ is independently drawn, and the particle index $j_i$ is selected such that $c_{j_i - 1} \le u_i < c_{j_i}$, where $c_k$ is the cumulative sum of weights. The primary advantage of this method is its simplicity and the statistical independence of each selection. However, its significant drawback is high variance; the number of offspring for any given particle is binomially distributed, meaning a particle with a reasonably high weight can be lost by chance, introducing unnecessary Monte Carlo error and reducing particle diversity.

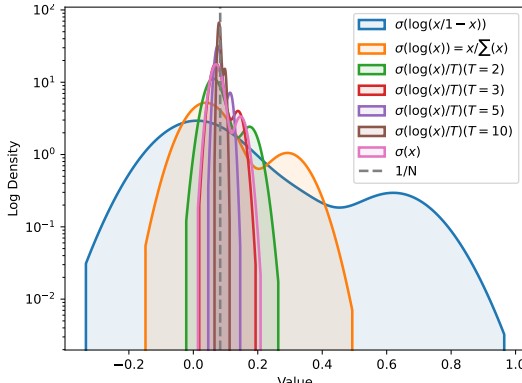

*Figure 13.* Different ways to compute the resampling distribution using the PRM output. $N = 12$, $ESS/N \in [0.18, 0.43]$, $1/N$ is the uniform. Our goal is to find a balance between early exploration and late exploitation in guided sampling.

Systematic resampling (Kitagawa, 1996), conversely, is a lower-variance technique that uses a single random draw to select the entire population. It generates one random number $u \sim U[0, 1)$ to create a deterministic, evenly-spaced set of $N$ pointers, $u_i = (i - 1 + u)/N$ for $i \in \{1, \ldots, N\}$. This stratified sequence is then used to select the new particles from the same cumulative weight distribution. The main advantage of this approach is its efficiency and reduced sampling variance. It guarantees that a particle with weight $w_k$ will be selected approximately $N \cdot w_k$ times, ensuring the resampled population is a much more faithful representation of the target distribution, at the cost of introducing a small correlation among the selected indices, as they all depend on the initial draw $u$. Due to its ability to better preserve the distribution's structure and reduce random fluctuations, systematic resampling is often the preferred method in most engineering applications for tracking and robotics.

# D. Computational Cost

The overall computational cost is determined by the forward passes through two models: the main generator (sampler) and the Process Reward Model (PRM). The total complexity scales linearly with the particle budget ($N$), the number of generation steps ($T$), and the maximum number of tokens generated per step ($C$).

- *PF and ePF:* In these particle filtering methods, each of the $N$ particles requires $C$ forward pass through the sampler for propagation (generation) and one forward pass through the PRM for scoring at each step.

- *ePF w/ LaM*: The Look-ahead Mechanism (LaM) introduces an additional propagation and scoring pass for each particle during its look-ahead phase. In the worst-case scenario, where look-ahead is used for all the steps in the first half of the trajectory $T$, the computational cost increases by approximately 50% compared to standard ePF.

*Table 5.* Comparison of computational complexity per sampling trajectory.

| Algorithm | Sampler Forwards | PRM Forwards | Worst-Case Total Ops |
|---|---|---|---|
| ePF | $N \times T \times C$ | $N \times T$ | $(N \times T \times C) + (N \times T)$ |
| ePF w/ LaM | $N \times (T + T/2) \times C$ | $N \times (T + T/2)$ | $(N \times (T + T/2) \times C) + (N \times (T + T/2))$ |

Although the PRM can be a larger model than the sampler, its contribution to the total compute is significantly smaller. This is because the sampler's cost scales with the number of generated tokens per step ($C$), as it operates auto-regressively, while the PRM requires only a single forward pass to score the entire output. Given that $C$ is typically in the range of $10^2$ to $10^3$, the generation cost overwhelmingly dominates the verification cost.

In practice, the overhead of LaM is much lower than its worst-case estimate. The look-ahead mechanism is only activated when entropic resampling is required (low effective sample size), which, as shown in Figure 14, occurs in only about 10-12% of the steps, not the theoretical maximum of 50%. This makes the actual added cost quite manageable.

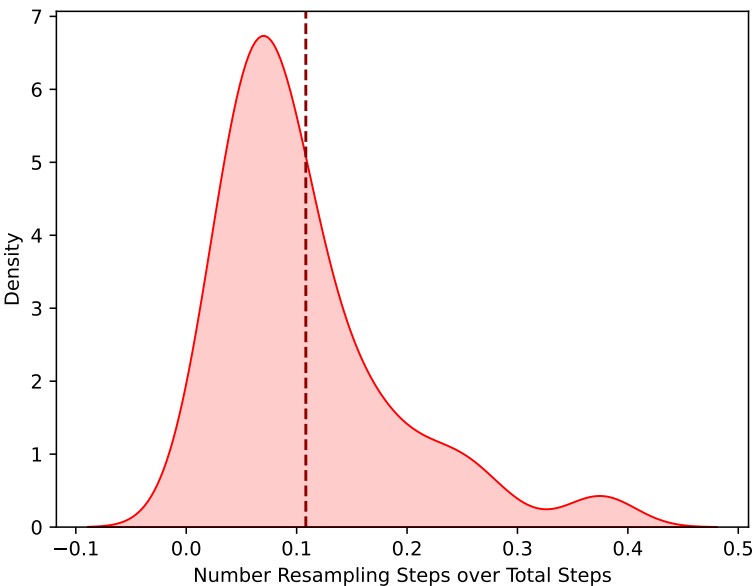

*Figure 14.* Density of resampling steps with LaM as a fraction of the total steps in the sampling trajectories on the AIME 2025 dataset, aggregated across various budgets. The data shows that resampling with look-ahead typically occurs in only 10-12% of the generation steps.

## D.1. Wall-Clock Analysis

*Table 6.* Wall-Clock by Budget over the 30 tasks in AIME 2024 using Qwen2.5-1.5B. Benchmark run on a single A100 using `vllm` for sampling.

| Algorithm | Budget | Mean | Median | Max |
|---|---|---|---|---|
| Best-of-N | 8 | 30.252 | 22.457 | 110.173 |
| | 16 | 57.920 | 43.473 | 138.366 |
| | 32 | 117.069 | 95.988 | 341.408 |
| Beam Search | 8 | 37.294 | 35.075 | 117.227 |
| | 16 | 85.077 | 72.572 | 228.574 |
| | 32 | 184.827 | 175.451 | 425.077 |
| Particle Filtering | 8 | 37.435 | 35.451 | 91.908 |
| | 16 | 96.992 | 85.207 | 275.590 |
| | 32 | 198.243 | 177.322 | 414.395 |
| Entropic Particle Filtering | 8 | 50.163 | 47.149 | 129.590 |
| | 16 | 125.121 | 116.733 | 377.558 |
| | 32 | 271.590 | 244.704 | 559.433 |
| Entropic Particle Filtering w/ LaM | 8 | 53.674 | 49.506 | 155.508 |
| | 16 | 138.884 | 126.072 | 468.172 |
| | 32 | 304.181 | 266.727 | 676.914 |

# E. Iso-Computational Cost of LaM

In Fig. 15 we evaluate if the performance gains from Look-ahead Modulation (LaM) justify its computational overhead (an extra forward pass/step) by comparing ePF w/ LaM at $N$ particles to standard ePF with a cost-equivalent budget.

A simple worst-case computational cost is established (Appx D): standard ePF complexity is $C_{\text{ePF}} \propto N_{\text{ePF}} \times T \times (C+1)$. Assuming LaM is active for its maximum $50\%$ of steps ($T/2$), its worst-case cost is $C_{\text{LaM}} \propto N_{\text{LaM}} \times (T + T/2) \times (C+1)$. Equating these costs ($C_{\text{ePF}} = C_{\text{LaM}}$) implies standard ePF can use $50\%$ more particles to reach the same cost as ePF w/ LaM. In practice, this means that from an iso-computational perspective, we can upper-bound the equivalent budget $N_{\text{ePF}}$ doubling the budget provided to LaM, i.e. $N_{\text{ePF}} = 1.5 \times N_{\text{LaM}} \leq 2 \times N_{\text{LaM}}$.

This analysis suggests comparing ePF w/ LaM @ 8 particles to ePF @ 12/16, and ePF w/ LaM @ 16 to ePF @ 24/32. However, this $1.5\times$ theoretical overhead is a worst-case; empirically (Fig. 14), LaM is triggered in only 10/12% of steps, making the actual overhead much lower. More importantly, empirical results demonstrate efficiency far exceeding this model: on AIME 2024, ePF w/ LaM with only 8 particles achieves performance comparable to standard ePF with 32 particles using Qwen2.5-1.5b. This shows LaM's predictive guidance is significantly more effective than re-allocating its cost to add more particles in ePF.

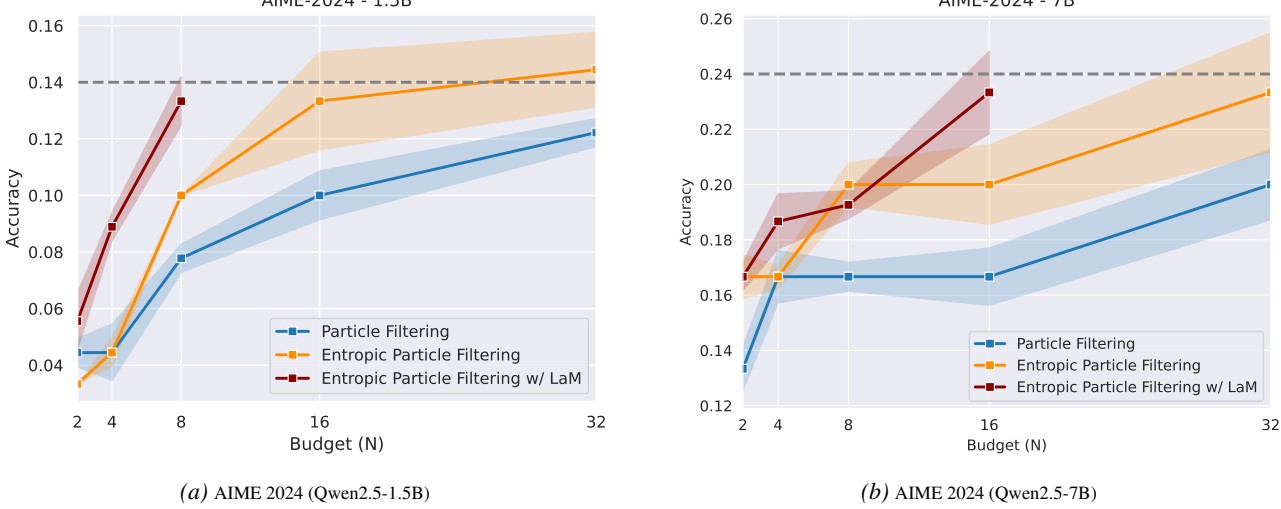

*(a)* AIME 2024 (Qwen2.5-1.5B)  *(b)* AIME 2024 (Qwen2.5-7B)

*Figure 15.* AIME-24 results with Qwen-2.5-1.5b-Instruct and Qwen-2.5-7b-Instruct for Entropic Particle Filtering and Look-Ahead Modulation. We run ePF w/ LaM until it reaches performance within 5% of the ePF with $N = 32$. We can see that ePF w/ LaM and 8 particles reaches a performance comparable to ePF with 32 particles using a Qwen2.5-1.5B-Instruct model. And ePF w/ LaM and 16 particles is competitive with ePF with 32 particles using a Qwen2.5-7B-Instruct model.

# F. Additional Baselines

## F.1. DORA Ablation

Our approach complements recent advances in Inference-Time Scaling (ITS) that seek to optimize compute allocation and avoid premature exploitation. DORA (Wang et al., 2025a), for instance, identifies that many search strategies suffer from a "solution-level bias" and proposes an optimal resource allocation policy at the direction-level to mitigate this. ePF can be viewed as a direct, algorithmic mitigation for the effects of this bias within a Sequential Monte Carlo framework; by monitoring population diversity via ESS , our Entropic Annealing mechanism explicitly prevents the particle set from collapsing onto a single, overconfident direction prematurely. In Figure 16a and 16b we compare PF, ePF, and DORA performance on AIME 2024 and AIME 2025 using Qwen2.5-1.5B-Instruct. We can see that ePF is competitive or better than DORA for most budgets and configurations.

Other methods, such as the tree-search algorithm REBASE (Wu et al., 2025), manage compute by using a node-quality reward to guide tree expansion, avoiding expensive rollouts. In contrast to these node-based tree structures, ePF remains a population-based method that manages a fixed-size set of full trajectories in parallel, offering a highly parallelizable alternative that balances exploration and less-myopic exploitation. In Figure 16c and 16d we compare REBASE and ePF.

## F.2. MCTS, abMCTS, FoT Ablations

Monte Carlo Tree Search (MCTS (Coulom, 2006)) and our Entropic Particle Filtering (ePF) share the core objective of balancing the exploration-exploitation trade-off to mitigate premature convergence. Both are guided algorithms that leverage reward signals to find high-quality solutions in complex, vast search spaces. However, their fundamental search strategies differ. MCTS is a pure search algorithm, where ePF is a bayesian inference algorithm designed to estimate a posterior probability distribution.

We use ePF to find density regions with high reward, where we can select promising solutions. MCTS is a node-based method that iteratively builds a single, asymmetric search tree, typically using a local selection policy (Upper Confidence Bounds, UCB (Kocsis & Szepesvári, 2006)) to balance exploring uncertain nodes with exploiting high-reward nodes.

In contrast, ePF is a population-based Sequential Monte Carlo method that maintains a fixed-size population of $N$ complete, parallel trajectories (particles). Consequently, their exploration mechanisms differ: MCTS applies exploration pressure locally at each decision node, whereas ePF's Entropic Annealing acts globally, intervening to preserve diversity across the entire particle population when its global diversity metric (for example ESS) drops.

Finally, while MCTS learns node values via full rollouts and backpropagation, our Look-ahead Modulation serves as a computationally cheap reweighting modulation, one-step forward-looking guide to make ePF's population-wide resampling step less myopic. In Table 7 and 8 we compare ePF with strong tree-based baselines, including MCTS, adaptive branching MCTS (abMCTS), and Forest-of-Thoughs (FoT) on AIME 2024 using Qwen2.5-1.5B-Instruct and Qwen2.5-7B-Instruct. ePF is competitive with the tree-based baselines, requiring smaller budgets and less compute.

*Table 7.* Baseline performance on AIME 2024 math benchmarks using Qwen2.5-1.5B-Instruct and Qwen2.5-7B-Instruct. The table shows Top-1 scores for particle budgets ($N \in \{16, 32\}$) for ePF, and call budget ($N \in \{16, 32, 64\}$) for MCTS and abMCTS.

| | AIME 2024 | | | |
| | Qwen2.5-1.5B-Instruct | | Qwen2.5-7B-Instruct | |
| | Budget | Correct | Budget | Correct |
|---|---|---|---|---|
| MCTS (Inoue et al., 2025) | 16 | 1/30 | 16 | 3/30 |
| abMCTS (Inoue et al., 2025) | 16 | 1/30 | 16 | 6/30 |
| MCTS (Inoue et al., 2025) | 32 | 4/30 | 32 | 3/30 |
| abMCTS (Inoue et al., 2025) | 32 | 3/30 | 32 | 6/30 |
| MCTS (Inoue et al., 2025) | 64 | 2/30 | 64 | 4/30 |
| abMCTS (Inoue et al., 2025) | 64 | 2/30 | 64 | 5/30 |
| ePF | 16 | 6/30 | 16 | 8/30 |
| ePF | 32 | 6/30 | 32 | 10/30 |

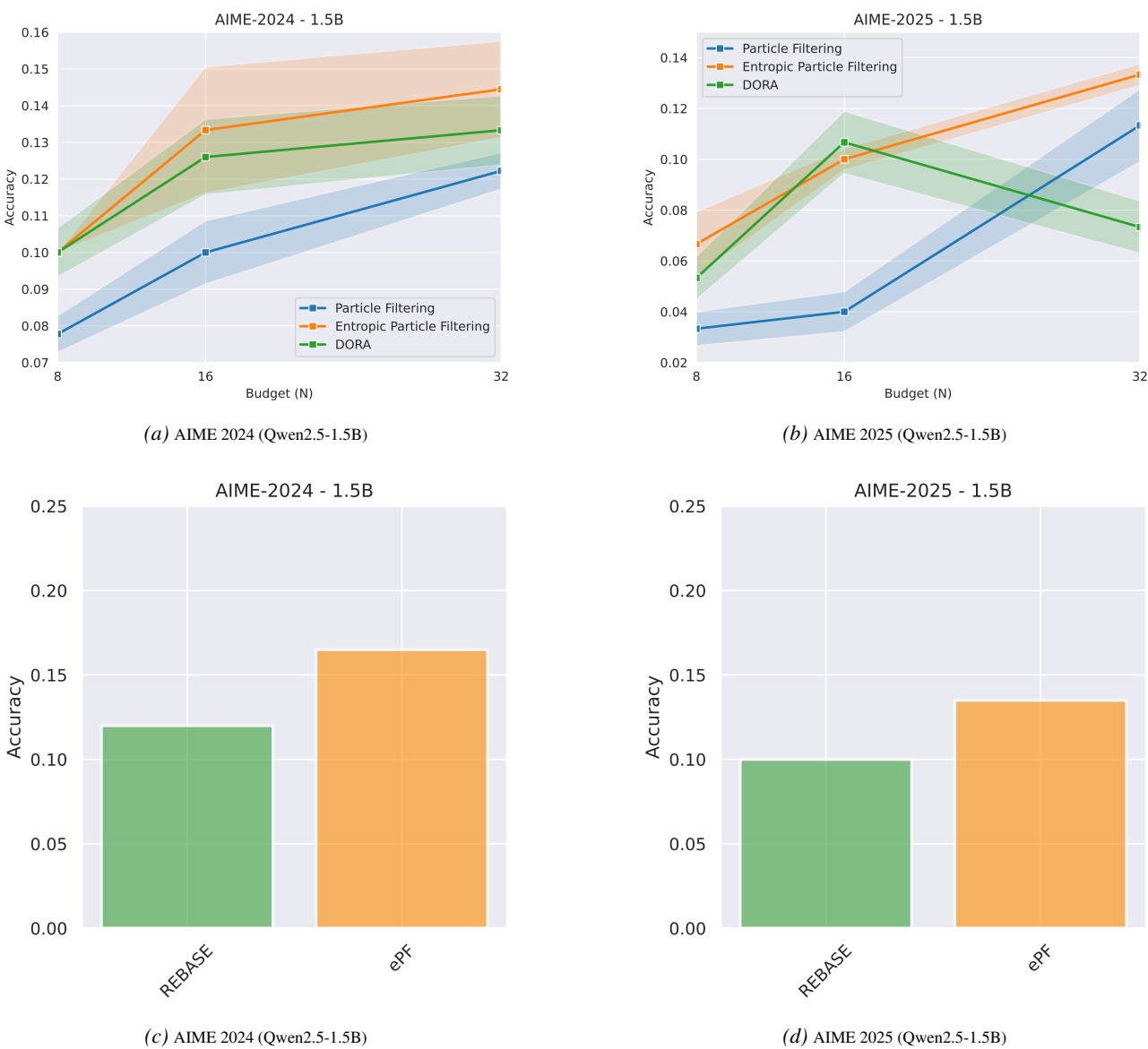

*Figure 16.* AIME 2024 and AIME 2025 results with Qwen-2.5-1.5b-Instruct for Particle Filtering, Entropic Particle Filtering, DORA, and REBASE. ePF is competitive or better than DORA (Wang et al., 2025a) and REBASE (Wu et al., 2025) for most budget and configurations.

*Table 8.* Performance comparison of Qwen2.5-1.5B-Instruct on AIME24 using ePF and FoT with various tree configurations. While FoT (Bi et al., 2024) achieves results comparable to ePF, it incurs significantly higher computational costs, necessitating parallelization across 8 A100 GPUs to complete within an hour. In contrast, ePF ($N = 32$) requires less than a quarter of that time.

*(a)* Performance comparison using ePF and FoT with various tree configurations.

| Configuration | GPUs | Duration | Accuracy |
|---|---|---|---|
| FoT (2 trees) | 8 | 1h 16 mins | 7/30 |
| FoT (4 trees) | 8 | 1h 30 mins | 4/30 |
| FoT (8 trees) | 8 | 1h 32 mins | 5/30 |
| ePF (32 particles) | 8 | 16 mins | 6/30 |

*(b)* Average memory required to solve an AIME 2024 question.

| Method | Memory Usage |
|---|---|
| MCTS | 5.9 GB |
| FoT (2 trees) | 12.3 GB |
| ePF (32 particles) | 18.4 GB |
| FoT (8 trees) | 21.1 GB |

# G. Additional Experiments

## G.1. Process Reward Models Overconfidence

In this work we investigate the reliability of Process Reward Models (PRMs), identifying a tendency toward overconfidence that drives premature exploitation in complex reasoning tasks. Figure 17 illustrates this calibration issue, visualizing the density of reward scores against ground truth on the MATH500 benchmark for two different PRMs: a stronger Qwen2.5-Math-PRM (Zhang et al., 2025) and a weaker Llama3.1-8B-PRM (Wang et al., 2024). In both cases, the PRMs are overconfidence, with the situation worsening weaker the PRM.

Despite these noisy signals, Entropic Particle Filtering (ePF) demonstrates significant robustness. As shown in Figure 18, when paired with a weaker, uncalibrated PRM, ePF effectively counteracts early exploitation and consistently outperforms standard Particle Filtering.

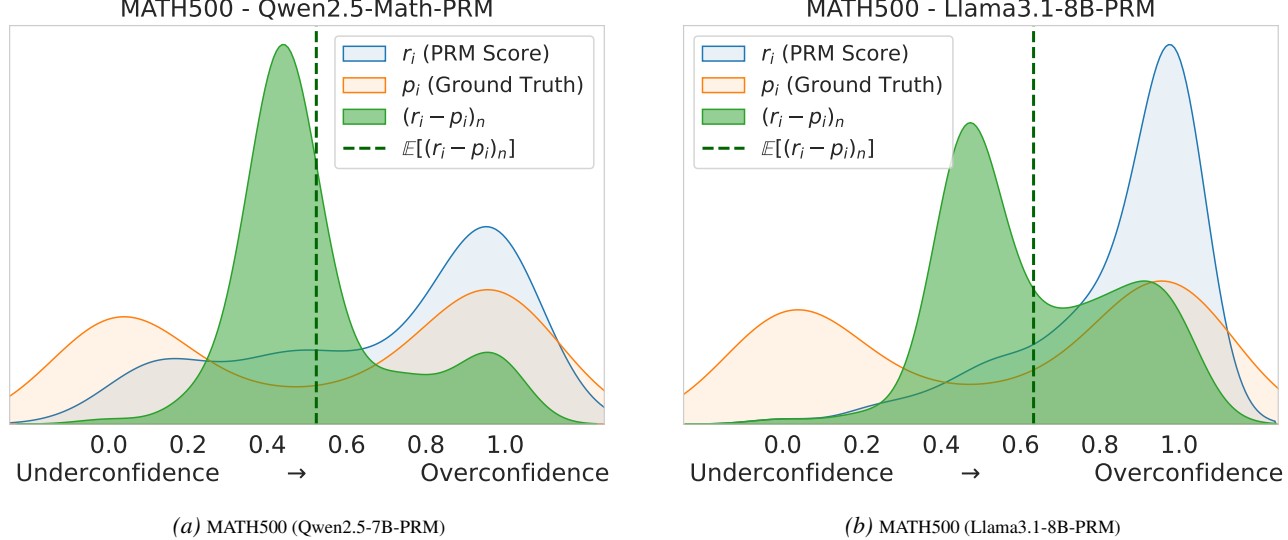

*(a)* MATH500 (Qwen2.5-7B-PRM)            *(b)* MATH500 (Llama3.1-8B-PRM)

*Figure 17.* PRM Overconfidence over MATH500 using different reward models. Early exploitation is a general problem when using step-level reward models for complex and long sequences.

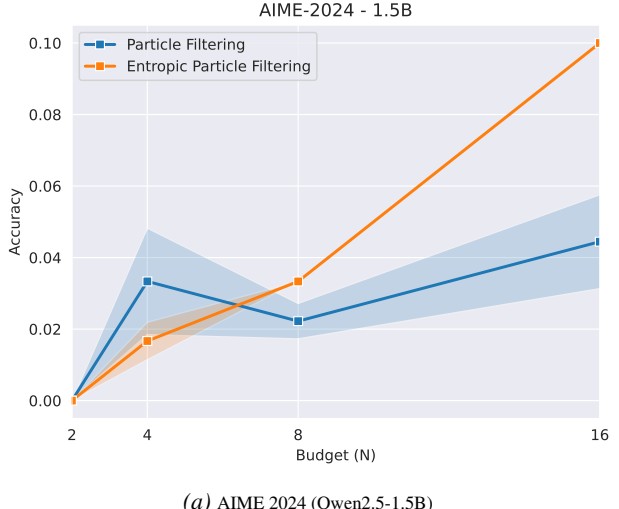

*(a)* AIME 2024 (Qwen2.5-1.5B)

*Figure 18.* AIME 2024 results with Qwen-2.5-1.5b-Instruct for PF and ePF for $N \in \{2, 4, 8, 16, 32\}$ using a weaker PRM (based on Llama3.1-8B-PRM, Fig. 17b). ePF is effective at reducing early exploitation using weak and heavily uncalibrated PRM, outperforming standard PF.

## G.2. Domain Ablation

While our primary research focuses on mathematical reasoning, Entropic Particle Filtering is a versatile technique with broader applications. To demonstrate its generalizability, we conducted a domain ablation study by applying the same methods to problems in finance and chemistry. Figure 19 illustrates the performance of Particle Filtering and Entropic Particle Filtering on subsets of the FinanceBench (Islam et al., 2023) and NumGLUE (Mishra et al., 2022) benchmarks, respectively, using the Qwen-2.5-7B-Instruct model. This analysis helps to validate the robustness and adaptability of our approach beyond its original mathematical context.

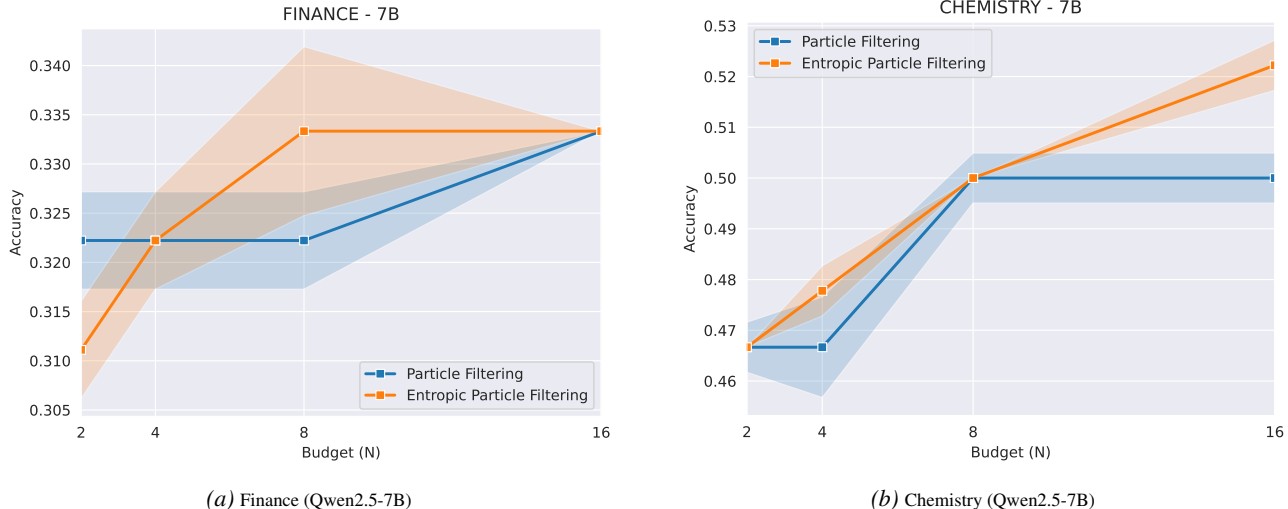

*(a)* Finance (Qwen2.5-7B)      *(b)* Chemistry (Qwen2.5-7B)

*Figure 19.* Finance and Chemistry results with Qwen-2.5-7b-Instruct for Particle Filtering, Entropic Particle Filtering for $N \in \{2, 4, 8, 16\}$. We consider a subset of 128 samples from FinanceBench (Islam et al., 2023) for financial problems and NumGLUE (Mishra et al., 2022) chemistry problems. We use the same PRM used for mathematical reasoning (Zhang et al., 2025).

### G.3. General Results on Mathematical Benchmarks

*Table 9.* Top-1 performance comparison of inference-time scaling algorithms on mathematical reasoning benchmarks with increasing complexity. Our proposed method, ePF, demonstrates superior performance over established baselines across multiple datasets of increasing difficulty for Qwen2.5-1.5B-Instruct and Qwen-2.5-7B-Instruct models. Best results are in **bold**. We use random subsets of 128 samples for each dataset and average over 3 runs. ORM: Output Reward Model; PRM: Process Reward Model; MV: Majority Voting.

| Algorithm | Selection | Scoring | Qwen2.5-1.5B-Instruct | | | | Qwen2.5-7B-Instruct | | | |
|---|---|---|---|---|---|---|---|---|---|---|
| | | | GSM8K | MATH500 | DEEPMATH | OMNIMATH | GSM8K | MATH500 | DEEPMATH | OMNIMATH |
| Base Sampling | - | - | $67.38_{\pm1.48}$ | $45.12_{\pm1.57}$ | $10.45_{\pm0.97}$ | $5.33_{\pm0.71}$ | $93.15_{\pm0.80}$ | $60.84_{\pm1.54}$ | $23.56_{\pm1.34}$ | $8.42_{\pm0.88}$ |
| Self-Consistency | MV | - | $82.19_{\pm1.21}$ | $53.62_{\pm1.58}$ | $13.11_{\pm1.07}$ | $7.24_{\pm0.82}$ | $94.65_{\pm0.71}$ | $65.43_{\pm1.50}$ | $30.22_{\pm1.45}$ | $9.56_{\pm0.93}$ |
| Best-of-N | Argmax | ORM | $92.84_{\pm0.82}$ | $57.91_{\pm1.56}$ | $20.15_{\pm1.27}$ | $\mathbf{10.35}_{\pm0.96}$ | $96.12_{\pm0.61}$ | $67.82_{\pm1.48}$ | $32.18_{\pm1.48}$ | $9.25_{\pm0.92}$ |
| Beam-Search | Argmax | PRM | $91.48_{\pm0.88}$ | $62.34_{\pm1.53}$ | $21.25_{\pm1.29}$ | $9.45_{\pm0.92}$ | $\mathbf{96.31}_{\pm0.60}$ | $66.21_{\pm1.49}$ | $32.15_{\pm1.48}$ | $\mathbf{10.84}_{\pm0.98}$ |
| PF | Argmax | PRM | $\mathbf{93.62}_{\pm0.77}$ | $60.28_{\pm1.55}$ | $22.54_{\pm1.32}$ | $8.51_{\pm0.88}$ | $\mathbf{96.19}_{\pm0.60}$ | $70.45_{\pm1.44}$ | $34.22_{\pm1.50}$ | $10.25_{\pm0.96}$ |
| ePF (ours) | Argmax | PRM | $\mathbf{93.85}_{\pm0.76}$ | $\mathbf{66.31}_{\pm1.49}$ | $\mathbf{25.12}_{\pm1.37}$ | $\mathbf{10.29}_{\pm0.96}$ | $95.74_{\pm0.64}$ | $\mathbf{71.28}_{\pm1.43}$ | $\mathbf{35.87}_{\pm1.52}$ | $\mathbf{10.88}_{\pm0.98}$ |

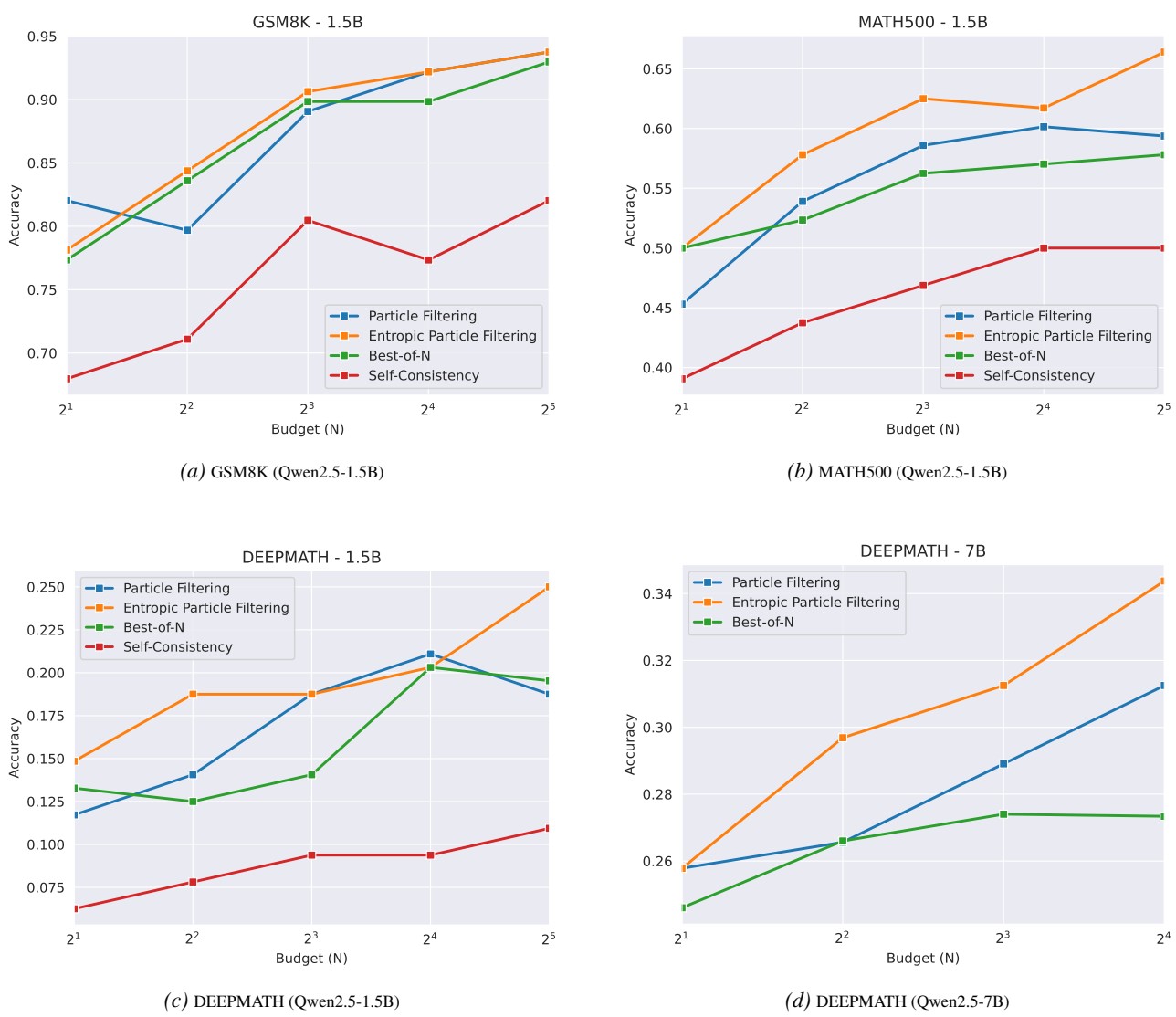

*Figure 20.* Math Benchmarks in order of complexity. We run Qwen2.5-1.5b-Instruct (1.5B) and Qwen2.5-7b-Instruct (7B) over a subset of GSM8K, MATH500, and DEEPMATH. ePF is competitive or better than strong baselines on all benchmarks. Table 1 for more results.

## G.4. Exploration for Hard Problems and Small Budgets

*Table 10.* Entropic Particle Filtering (`ePF`) outperforms baselines on AIME math benchmarks. The table shows aggregate Top-1 scores (%) across budgets ($N \in \{2, 4, 8, 16, 32\}$), reweighted to favor large (proportional weighting `w`), uniform (`u`), or small (inverse weighting `iw`) budgets. Higher is better. ePF provides a large gains for most models and budgets compared to PF.

| | AIME 2024 | | | | | | AIME 2025 | | | | | |
| | Qwen2.5-1.5B-In | | | Qwen2.5-7B-In | | | Qwen2.5-1.5B-In | | | Qwen2.5-7B-In | | |
| | w | u | iw | w | u | iw | w | u | iw | w | u | iw |
|---|---|---|---|---|---|---|---|---|---|---|---|---|
| Base Sampling (Yang et al., 2025) | - | 3.33 | - | - | 10.00 | - | - | 3.33 | - | - | 6.66 | - |
| Best-of-N | 9.90 | 7.20 | 4.00 | 23.09 | 20.20 | 17.48 | 5.13 | 3.60 | 2.52 | 17.41 | 15.80 | 14.90 |
| Beam-Search | 10.29 | 8.40 | 5.55 | 17.93 | 14.00 | 11.26 | 9.45 | **7.40** | **4.32** | 14.19 | 16.20 | 16.81 |
| PF (Puri et al., 2025) | 11.16 | 9.00 | **6.99** | **26.06** | **21.60** | **18.13** | 7.32 | 4.50 | 2.87 | 21.61 | 19.80 | 17.61 |
| `ePF` (ours) | **17.06** | **11.20** | 5.55 | 26.23 | 21.00 | 16.06 | **10.82** | 7.28 | 3.42 | **28.83** | **25.10** | **21.96** |
| $\Delta$(`ePF`, PF) | +24.53 % | | | -3.84 % | | | +53.85 % | | | +28.58 % | | |

*Table 11.* Baseline performance on AIME 2024 math benchmarks using Qwen2.5-1.5B-Instruct and Qwen2.5-7B-Instruct. The table shows Top-1 scores for budgets ($N \in \{2, 4, 8, 16, 32\}$).

| | AIME 2024 | | | | | | | | | |
| | Qwen2.5-1.5B-Instruct | | | | | Qwen2.5-7B-Instruct | | | | |
| | 2 | 4 | 8 | 16 | 32 | 2 | 4 | 8 | 16 | 32 |
|---|---|---|---|---|---|---|---|---|---|---|
| Best-of-N | 0.03 | 0.03 | 0.10 | 0.13 | 0.10 | 0.16 | 0.16 | 0.23 | 0.20 | 0.26 |
| Beam-Search | 0.03 | 0.06 | 0.10 | 0.13 | 0.10 | 0.10 | 0.11 | 0.13 | 0.13 | 0.23 |
| PF | 0.06 | 0.06 | 0.10 | 0.10 | 0.13 | 0.16 | 0.20 | 0.16 | 0.26 | 0.30 |
| `ePF` | 0.03 | 0.03 | 0.10 | 0.20 | 0.20 | 0.13 | 0.16 | 0.20 | 0.26 | 0.30 |

*Table 12.* Baseline performance on AIME 2025 math benchmarks using Qwen2.5-1.5B-Instruct and Qwen2.5-7B-Instruct. The table shows Top-1 scores for budgets ($N \in \{2, 4, 8, 16, 32\}$).

| | AIME 2025 | | | | | | | | | |
| | Qwen2.5-1.5B-Instruct | | | | | Qwen2.5-7B-Instruct | | | | |
| | 2 | 4 | 8 | 16 | 32 | 2 | 4 | 8 | 16 | 32 |
|---|---|---|---|---|---|---|---|---|---|---|
| Best-of-N | 0.03 | 0.01 | 0.03 | 0.06 | 0.06 | 0.16 | 0.10 | 0.20 | 0.13 | 0.20 |
| Beam-Search | 0.01 | 0.06 | 0.10 | 0.10 | 0.10 | 0.16 | 0.16 | 0.26 | 0.10 | 0.13 |
| PF | 0.03 | 0.01 | 0.03 | 0.04 | 0.11 | 0.16 | 0.20 | 0.16 | 0.23 | 0.20 |
| `ePF` | 0.01 | 0.02 | 0.10 | 0.10 | 0.13 | 0.20 | 0.23 | 0.23 | 0.26 | 0.33 |

*Table 13.* Baseline performance on AIME 2024 math benchmarks using Qwen3-0.6B and Qwen3-1.7B. The table shows Top-1 scores for budgets ($N \in \{2, 4, 8, 16, 32\}$).

| | AIME 2024 | | | | | | | | | |
| | Qwen3-0.6B | | | | | Qwen3-1.7B | | | | |
| | 2 | 4 | 8 | 16 | 32 | 2 | 4 | 8 | 16 | 32 |
|---|---|---|---|---|---|---|---|---|---|---|
| Best-of-N | 0.03 | 0.05 | 0.06 | 0.06 | 0.10 | 0.16 | 0.16 | 0.23 | 0.16 | 0.23 |
| PF | 0.07 | 0.07 | 0.17 | 0.12 | 0.16 | 0.20 | 0.23 | 0.16 | 0.23 | 0.20 |
| ePF | 0.08 | 0.08 | 0.14 | 0.10 | 0.18 | 0.20 | 0.23 | 0.23 | 0.26 | 0.33 |
| ePF w/ LaM | 0.08 | 0.10 | 0.10 | 0.20 | 0.20 | 0.23 | 0.20 | 0.26 | 0.26 | 0.33 |

*Table 14.* Baseline performance on AIME 2025 math benchmarks using Qwen3-0.6B and Qwen3-1.7B. The table shows Top-1 scores for budgets ($N \in \{2, 4, 8, 16, 32\}$).

| | AIME 2025 | | | | | | | | | |
| | Qwen3-0.6B | | | | | Qwen3-1.7B | | | | |
| | 2 | 4 | 8 | 16 | 32 | 2 | 4 | 8 | 16 | 32 |
|---|---|---|---|---|---|---|---|---|---|---|
| Best-of-N | 0.07 | 0.13 | 0.20 | 0.23 | 0.16 | 0.20 | 0.16 | 0.16 | 0.20 | 0.20 |
| PF | 0.10 | 0.13 | 0.16 | 0.20 | 0.23 | 0.20 | 0.20 | 0.13 | 0.20 | 0.20 |
| ePF | 0.10 | 0.13 | 0.16 | 0.30 | 0.23 | 0.15 | 0.23 | 0.20 | 0.20 | 0.26 |
| ePF w/ LaM | 0.10 | 0.13 | 0.16 | 0.33 | 0.26 | 0.16 | 0.20 | 0.23 | 0.26 | 0.30 |

## G.5. Look-ahead Modulation Efficiency

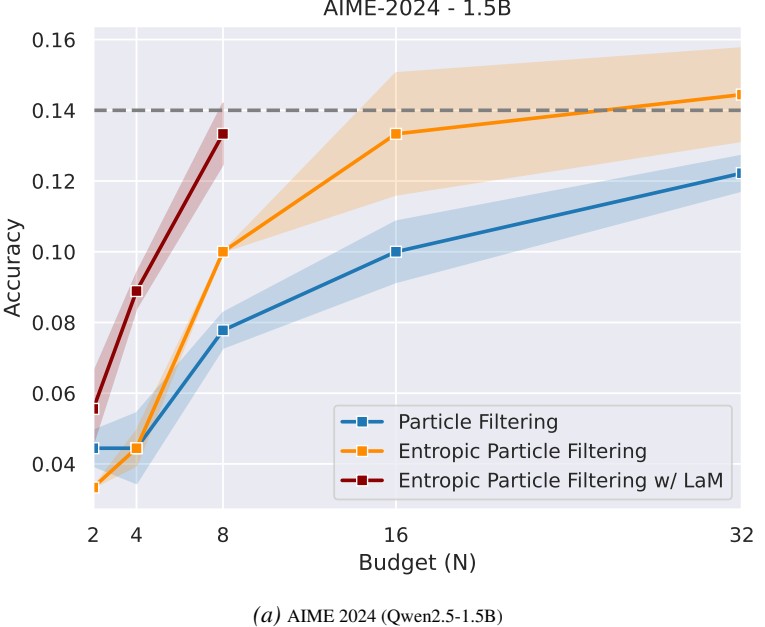

*(a)* AIME 2024 (Qwen2.5-1.5B)

*(b)* AIME 2024 (Qwen2.5-7B)

*Figure 21.* AIME-24 results with Qwen-2.5-1.5b-Instruct and Qwen-2.5-7b-Instruct for Entropic Particle Filtering and Look-Ahead Modulation. We run ePF w/ LaM until it reaches performance within 5% of the ePF with $N = 32$. We can see that ePF w/ LaM and 8 particles reaches a performance comparable to ePF with 32 particles using a Qwen2.5-1.5B-Instruct model. And ePF w/ LaM and 16 particles is competitive with ePF with 32 particles using a Qwen2.5-7B-Instruct model.

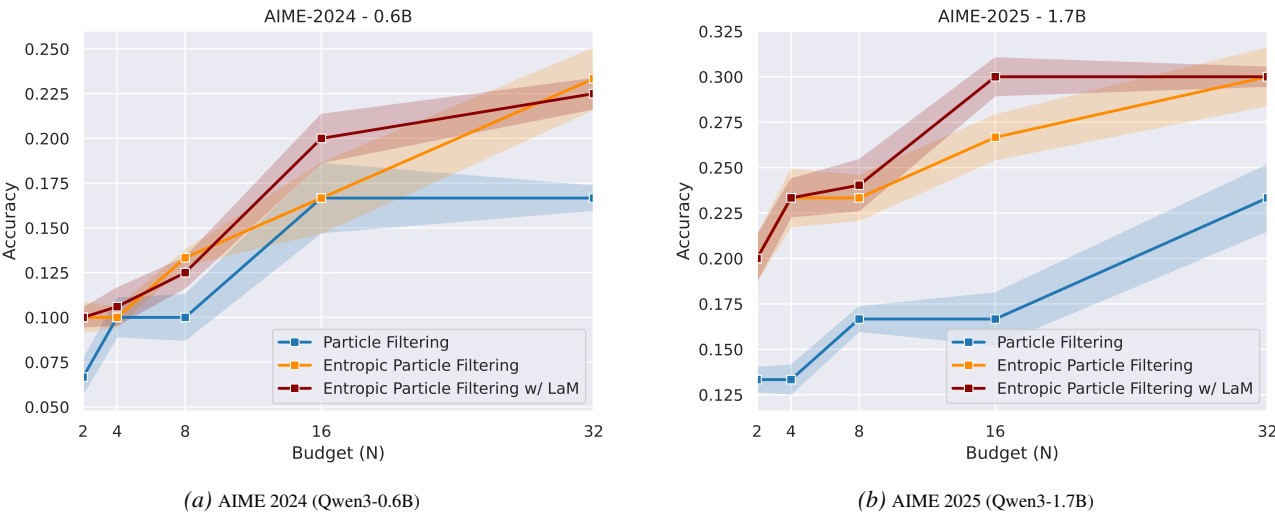

*(a)* AIME 2024 (Qwen3-0.6B)

*(b)* AIME 2025 (Qwen3-1.7B)

*Figure 22.* AIME-24 and AIME-25 results with Qwen3-0.6B and Qwen3-1.7B w/o thinking mode for ePF and ePF w/ LaM.

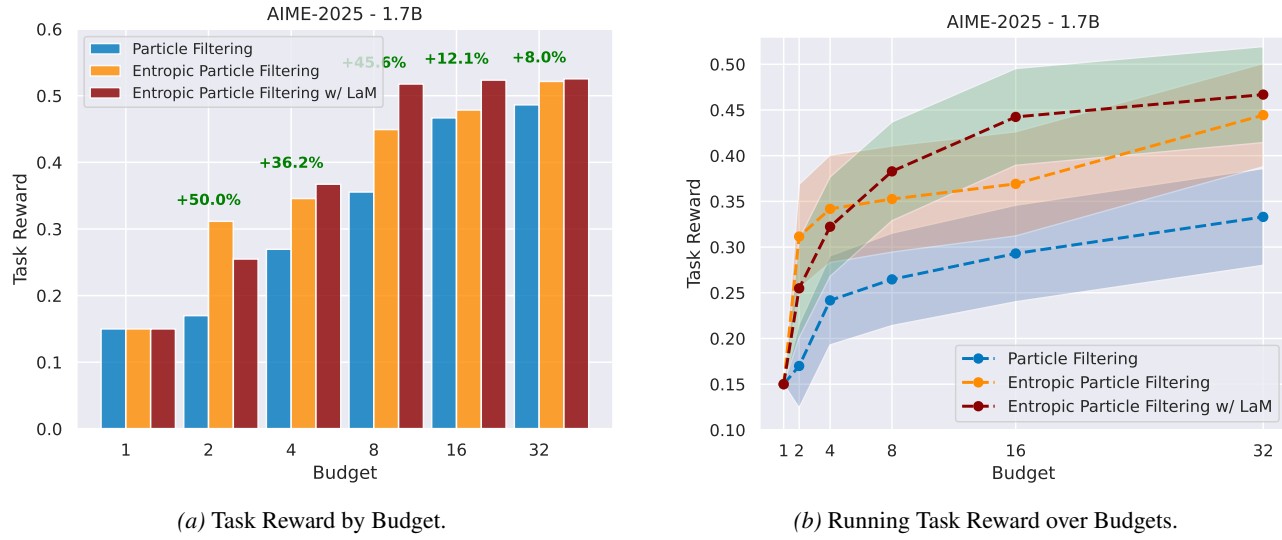

*(a)* Task Reward by Budget.

*(b)* Running Task Reward over Budgets.

*Figure 23.* Task reward comparison on AIME-2025 using Qwen3-1.7B. Our Entropic Particle Filtering (`ePF`) and its Look-ahead variant (`ePF w/ LaM`) significantly improve performance over standard Particle Filtering (PF) across all particle budgets. This demonstrates that mitigating premature exploitation leads to significant performance gains.

## G.6. PF and ePF Max Performance

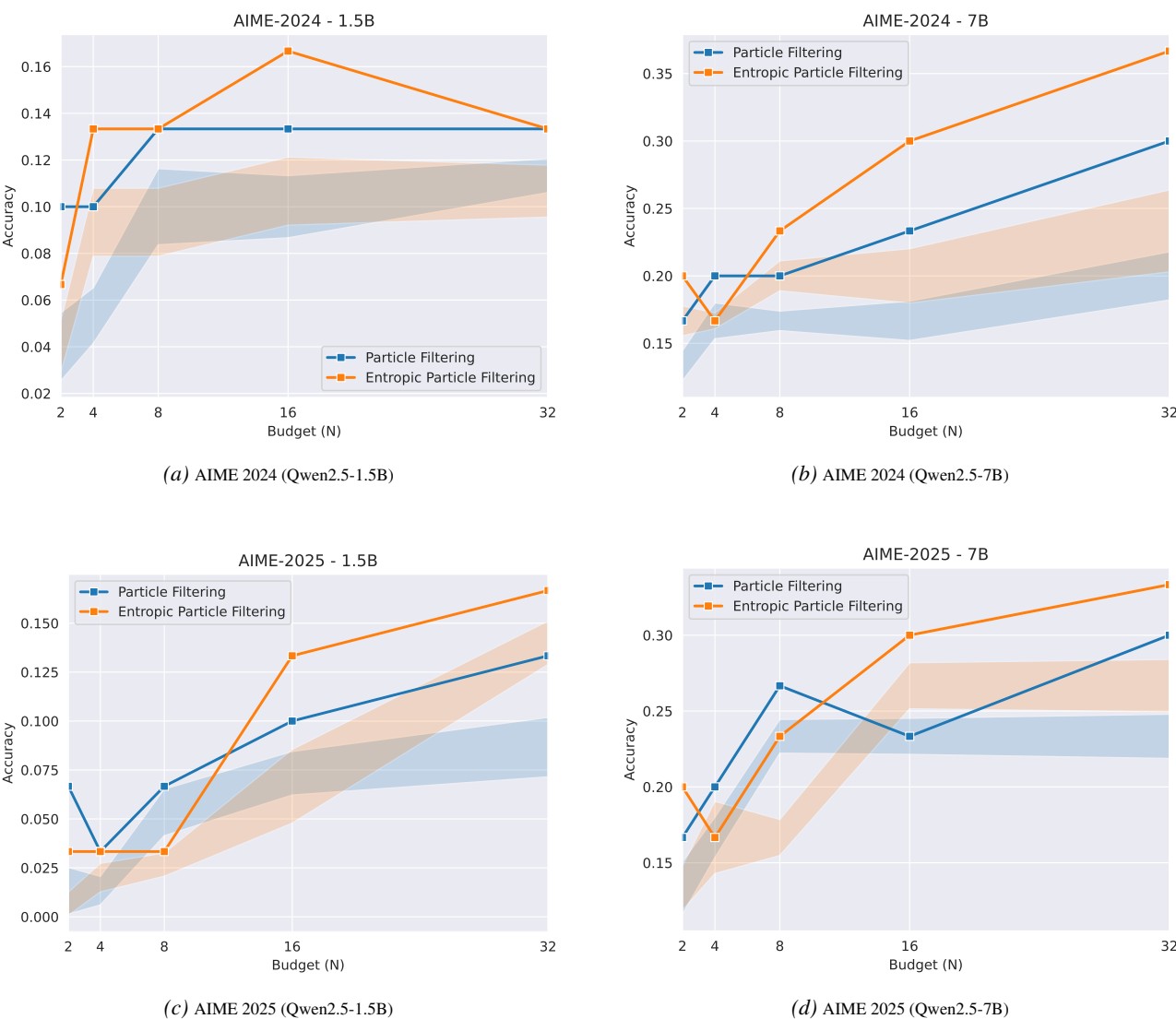

*Figure 24.* Best Top-1 performance as a function of inference budget ($N$) across 5 runs. ePF clearly outperforms PF on AIME-2024 and AIME-2025 using models of different sizes (Qwen2.5-1.5B-Instruct and Qwen2.5-7B-Instruct). Max Sequence length of 12288.

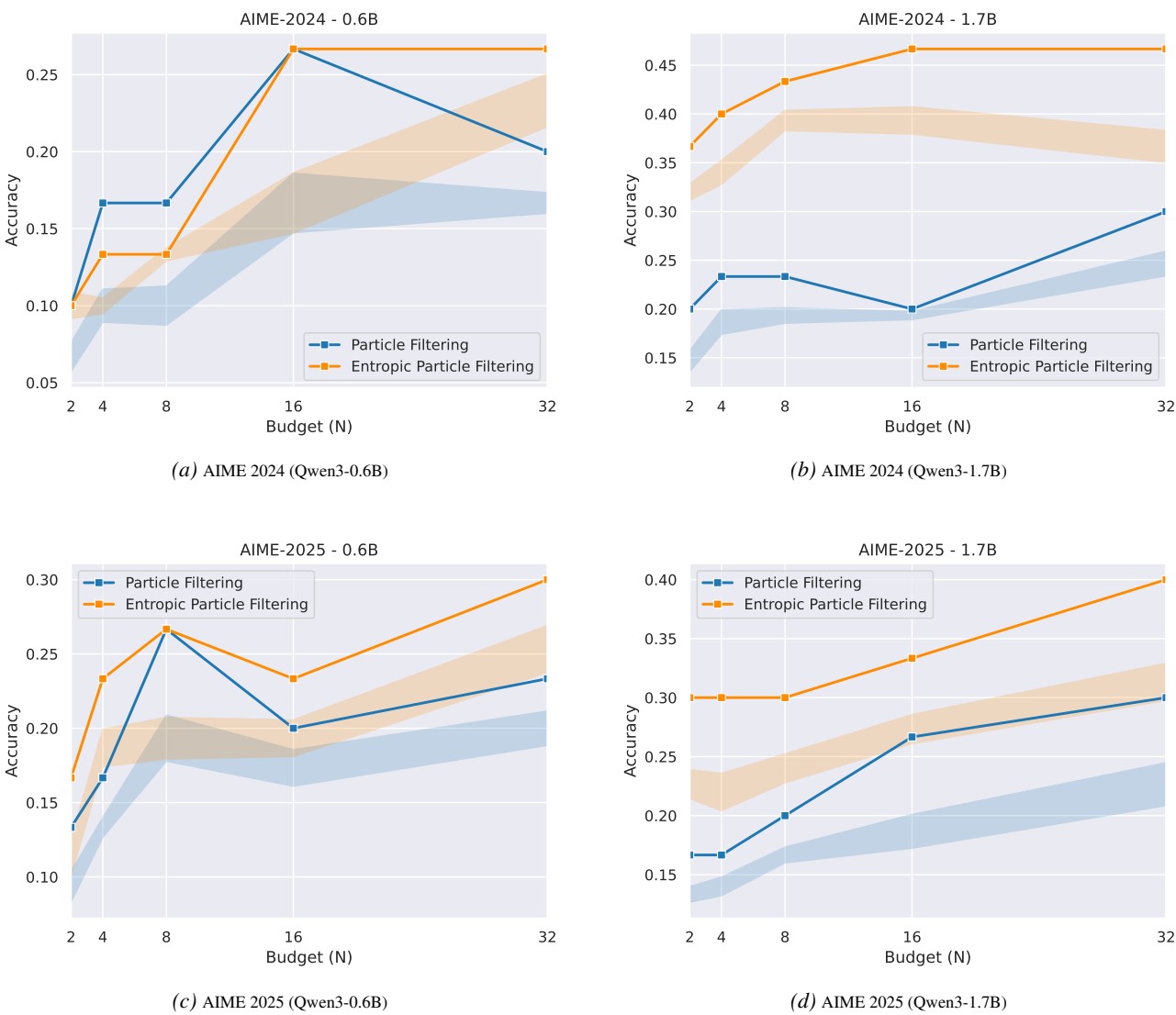

*Figure 25.* Best Top-1 performance as a function of inference budget ($N$) across 5 runs. ePF clearly outperforms PF on AIME-2024 and AIME-2025 using models of different sizes (Qwen3-0.6B and Qwen3-0.6B). Max Sequence length of 12288.

### G.7. Specialist and Generalist Models

*Table 15.* Evaluating Specialized and Generalist Language Models on AIME-2024. We report the best Top-1 results among 5 runs. Math-Specialists results from (Puri et al., 2025).

| Method | AIME 2024 |
|---|---|
| *Math-Specialists* (Qwen2.5-Math-7B-Instruct) | |
| Greedy | 5/30 |
| Self Consistency | 4/30 |
| Best-of-N (W) | 5/30 |
| Beam-Search | 7/30 |
| DVTS (Beeching et al., 2024) | 6/30 |
| PF (Puri et al., 2025) | 10/30 |
| *Generalist* (Qwen2.5-7B-Instruct) | |
| Greedy | 3/10 |
| PF | 9/30 |
| `ePF` (ours) | 11/30 |
| `ePF w/ LaM` (ours) | 10/30 |
| *Generalist* (Qwen3-0.6B, w/o thinking) | |
| Greedy | 3/30 |
| PF | 8/30 |
| `ePF` (ours) | 10/30 |
| `ePF w/ LaM` (ours) | 10/30 |
| *Generalist* (Qwen3-1.7B, w/o thinking) | |
| Greedy | 4/30 |
| PF | 10/30 |
| `ePF` (ours) | 14/30 |
| `ePF w/ LaM` (ours) | 14/30 |

## G.8. Guided-Search Ablation

*Table 16.* Comparison of our method, ePF, against state-of-the-art baselines on the MATH500, GSM8K, and AIME-24-25 mathematical reasoning benchmarks. Our approach consistently achieves higher scores, demonstrating its effectiveness.

*(a)* MATH500

| Method | Score |
| --- | --- |
| TSMC (Feng et al., 2024) | 60.8 |
| ePF (ours) | 65.1 |

*(b)* GSM8K

| Method | Score |
| --- | --- |
| TSMC (Feng et al., 2024) | 91.7 |
| ePF (ours) | 94.3 |

*(c)* AIME-24-25

| Method | Score |
| --- | --- |
| IAS-C (Park et al., 2025) | 18.3 |
| ePF (ours) | 26.4 |

## G.9. Backbone Ablation

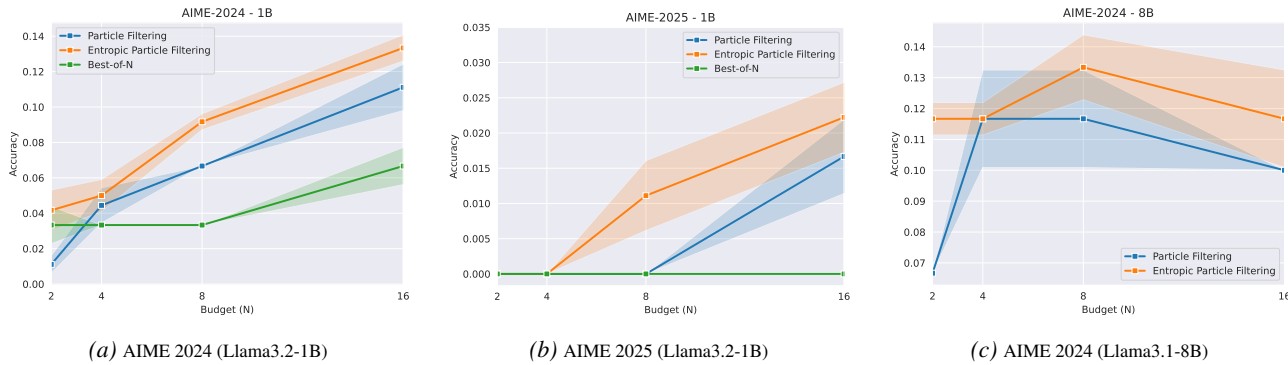

*(a)* AIME 2024 (Llama3.2-1B)  *(b)* AIME 2025 (Llama3.2-1B)  *(c)* AIME 2024 (Llama3.1-8B)

*Figure 26.* Llama3.2-1B and Llama3.1-8B ablation for AIME-24 and AIME-25. Mitigating exploitation using ePF improves performance on hard tasks even when using weak models.

## G.10. Temperature Schedule Ablation

While particle filtering ideally explores the search space by resampling from a diverse set of particles, it often suffers from premature collapse. This issue arises when the search process oversamples a few promising particles too early, often due to misspecified rewards or overconfidence. This effectively turns the algorithm into a greedy search, which is particularly detrimental for complex problems as it stifles thorough exploration. To counteract this, we introduce an annealing strategy that modulates the temperature of the resampling distribution to preserve particle diversity and mitigate early exploitation.

To control this process, we introduce a dynamically adjusted temperature, $1/\beta_t$, which follows an annealing schedules at each time step $t$ (Fig. 27). Two of these schedules adaptively respond to particle diversity, which we quantify using the normalized entropy,

$$H_n(t) = -\frac{\sum_{i=1}^{N} w_t^i \log w_t^i}{\log N},$$

and the effective sample size,

$$ESS(t) = \frac{1}{\sum_{j=1}^{N} (w_t^j)^2}.$$

ESS and Entropy are closely related [5].

The proposed schedules are (Figure 27):

- *Linear*: a simple decay schedule, $\beta_t^{-1} = k - t/T$.

- *ESS-based*: a temperature schedule defined as $\beta_t^{-1} = (N/ESS(t)) \cdot (1 - t/T)$.

- *Entropy-based*: a convex combination given by $\beta_t = H_n(t) + (1 - H_n(t)) \cdot t/T$.

At each step, the resulting inverse temperature $\beta_t$ modulates the particle rewards $r_t^i$ to produce the resampling weights via a softmax function (Fig. 4):

$$w_t^i = \frac{\exp(r_t^i \cdot \beta_t)}{\sum_{j=1}^{N} \exp(r_t^j \cdot \beta_t)}.$$

See Algorithm 2 for implementation details and Figure 28 for a temperature schedule ablation over AIME-2024 and AIME-2025. If not otherwise specified, we use the *ESS-based* schedule for the core experiments.

The ESS-based schedule is designed to react aggressively to particle collapse. When the ESS is low (indicating low diversity), the temperature sharply increases to encourage exploration. The $(1 - t/T)$ term ensures this exploratory pressure is strongest in the early stages and gradually anneals or cools over time, allowing the search to shift from exploration to exploitation as it progresses (Fig. 27).

---

[5]One way to define $ESS(t)$ is $\exp(H(t))$ (Martino & Elvira, 2025).

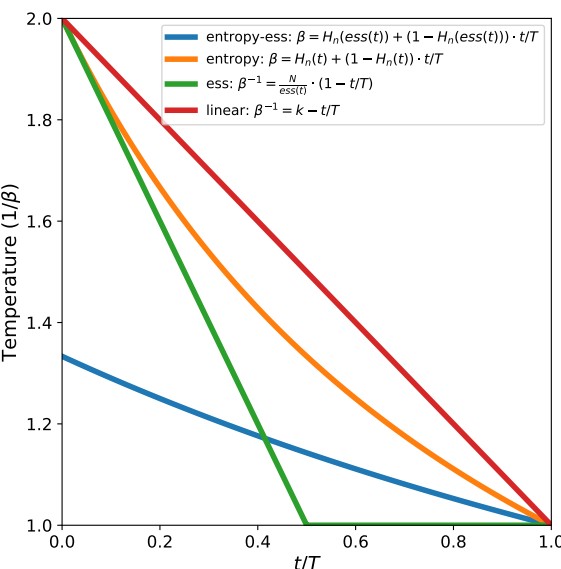

*Figure 27.* Different Schedules for the resampling distribution temperature annealing increasing the number of sampling steps. Here we set $k = 2$, $H_n = 0.5$, $N = 16$, and $ess(t) = 8$.

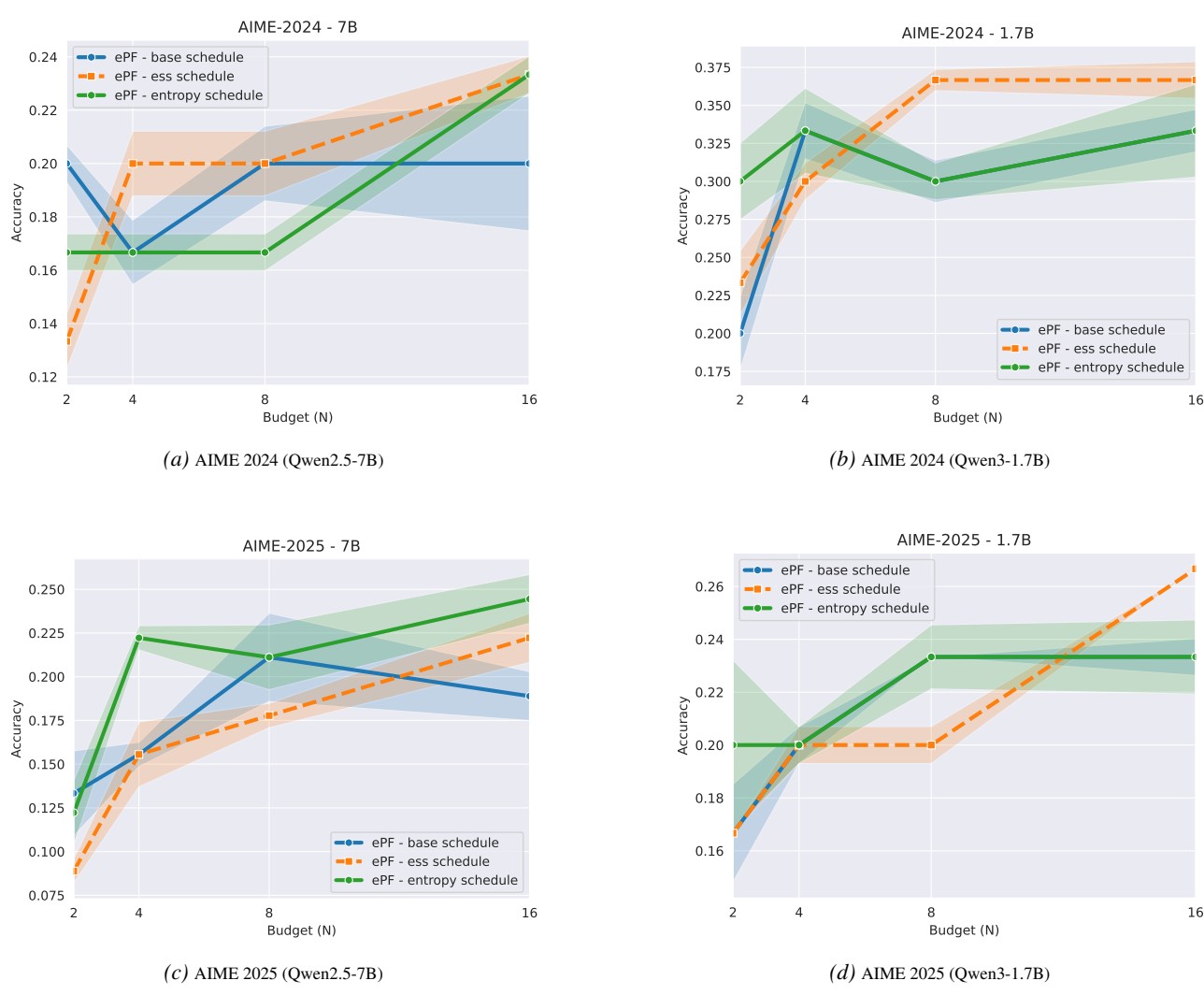

*Figure 28.* Temperature Schedule ablation for AIME-24 and AIME-25 using Qwen2.5-7B-Instruct and Qwen3-1.7B. Among models and datasets, ESS-based temperature annealing is the most consistent.

## G.11. Effective Sample Size Ablation

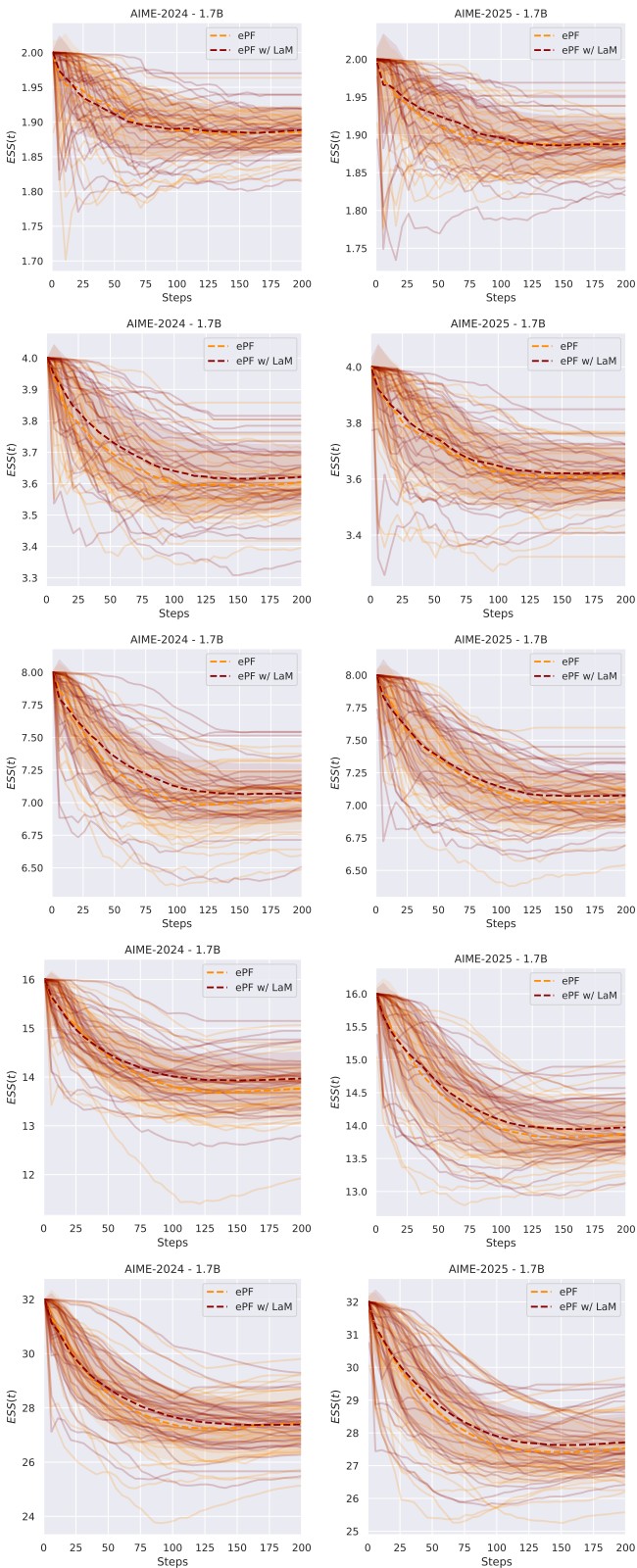

*Figure 29.* Running Effective Sample Size (post-resampling) for AIME 2024 and AIME 2025 using Qwen3-1.7B over increasing budgets $N \in \{2, 4, 8, 16, 32\}$.

## G.12. Coverage and Diversity

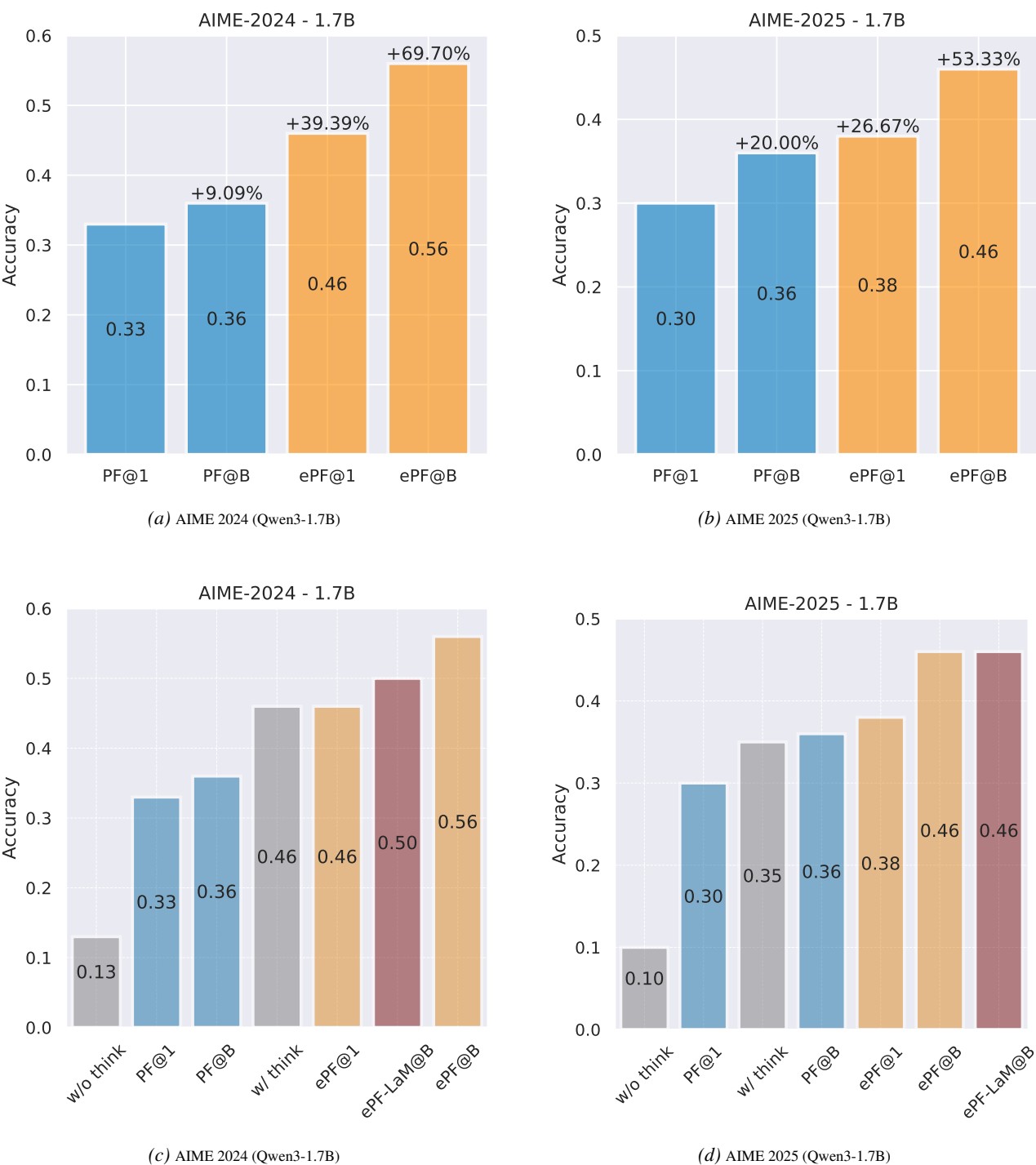

*Figure 30.* Diversity of Correct Solutions for pass@1 and pass@B generating up to 12k tokens/sequence. pass@B is defined as the expected pass@1 aggregated over budgets. The goal of mitigating early exploitation in PF is to maintain a distribution over hypotheses and do not over-optimize early on in the sampling process. Given that ePF is exploring more, we expect the algorithm to find more diverse solutions.

**Syntactic Diversity**    For syntactic diversity, we consider the Jaccard coefficient, which measures the similarity between two sets. Given two responses, $r_a$ and $r_b$, let $T_a$ and $T_b$ be their respective sets of unique tokens. The Jaccard coefficient is

defined as $J(T_a, T_b) = \frac{|T_a \cap T_b|}{|T_a \cup T_b|}$. A lower Jaccard coefficient indicates less overlap in vocabulary and thus greater syntactic diversity. The Jaccard distance, defined as $1 - J(T_a, T_b)$, provides a complementary view, where a higher value signifies greater dissimilarity. By averaging the pairwise Jaccard distance across all response pairs, we can obtain an overall measure of syntactic diversity for the entire set of responses; see Figure 31.

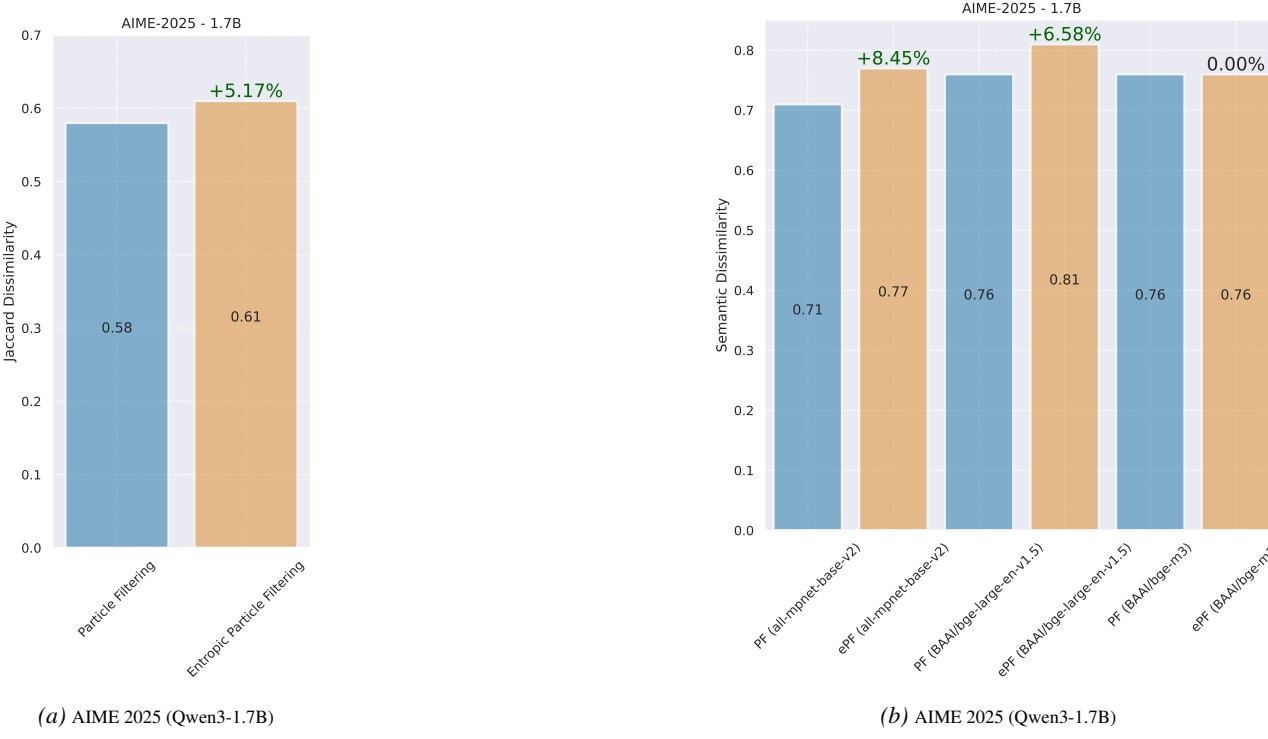

*(a)* AIME 2025 (Qwen3-1.7B)          *(b)* AIME 2025 (Qwen3-1.7B)

*Figure 31.* Syntactic (left) and Semantic (right) Dissimilarity between selected trajectories using Particle Filtering and Entropic Particle Filtering. Entropic Particle Filtering tends to generate responses that are more syntactically and semantically diverse.

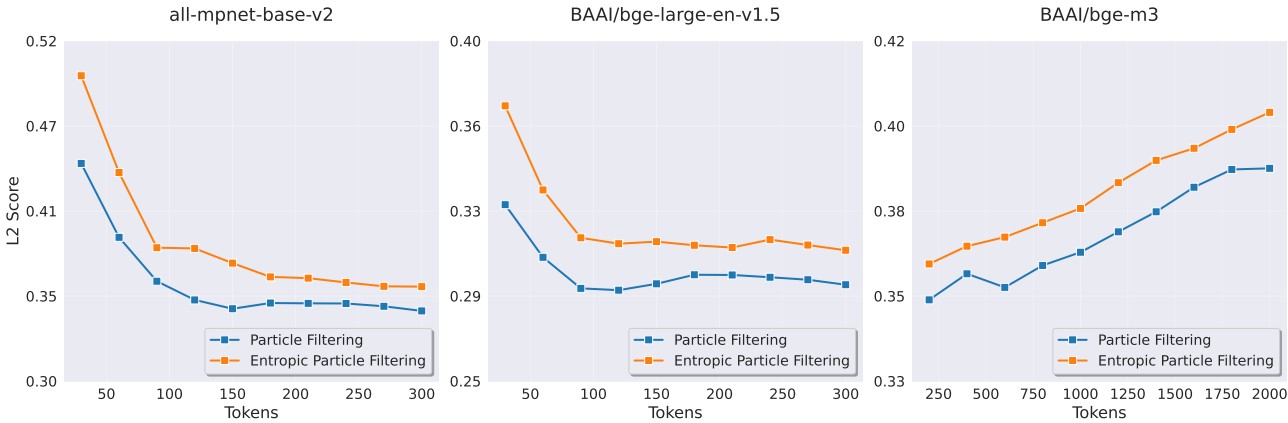

*Figure 32.* Semantic diversity measured using average pairwise $\ell_2$ distance across embeddings from three models. We compare Particle Filtering and Entropic Particle Filtering across all AIME 2025 questions using responses generated by the Qwen3-1.7B model. The analysis is based on the first $N$ tokens of each response. Entropic Particle Filtering tends to generate responses that are more semantically diverse according to the $\ell_2$ distance metric.

**Semantic Diversity**   To quantify the semantic diversity among a set of $N = 6$ generated responses, $\{r_1, r_2, \ldots, r_N\}$, we introduce a Semantic Entropy score. First, each response $r_i$ is encoded into a normalized embedding vector $\mathbf{e}_i = \text{model}(r_i)$, where $\|\mathbf{e}_i\|_2 = 1$, using one of three sentence-embedding models: `all-mpnet-base-v2`,

BAAI/bge-large-en-v1.5, or BAAI/bge-m3, which map text into a semantic vector space suitable for similarity comparisons. A pairwise cosine similarity matrix, $\mathbf{S}$, is then computed, where $S_{ij} = \mathbf{e}_i \cdot \mathbf{e}_j$. To ensure a response is not compared with itself, the diagonal elements of $\mathbf{S}$ are set to negative infinity, i.e., $S_{ii} = -\infty$. Next, a row-wise softmax function with a temperature parameter $\tau$ is applied to this matrix to obtain a probability distribution for each response over its neighbors: $p(j|i) = \frac{\exp(S_{ij}/\tau)}{\sum_{k \neq i} \exp(S_{ik}/\tau)}$. These individual distributions are then aggregated into a single mean probability distribution, $\bar{\mathbf{p}}$, where $\bar{p}_j = \frac{1}{N} \sum_{i=1}^{N} p(j|i)$. Finally, the Sem-Ent score is calculated as the Shannon entropy of this aggregated distribution, $H(\bar{\mathbf{p}}) = -\sum_{j=1}^{N} \bar{p}_j \log(\bar{p}_j)$, normalized by the maximum possible entropy for $N-1$ choices, which is $\log(N-1)$. The final score is clamped to the range $[0,1]$. As a complementary measure, we also compute the average pairwise $\ell_2$ distance between response embeddings, again using the same three models. For each set of $N = 6$ responses, we calculate the $\ell_2$ norm between every pair of embeddings and average across all $\binom{N}{2}$ pairs and all questions. A higher value indicates greater semantic diversity; see Figures 31 and 32.

# H. Entropic Particle Filtering Pipeline

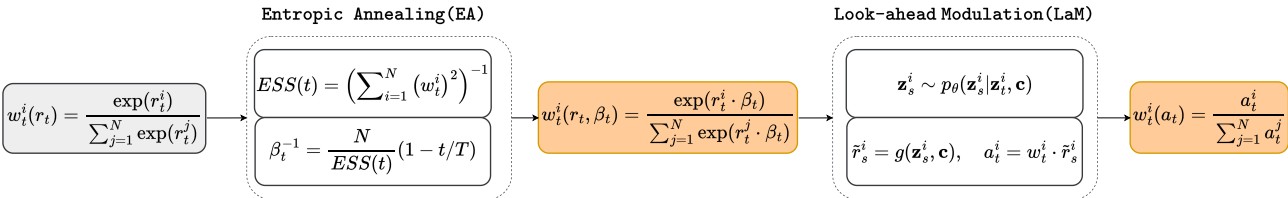

*(a)* Entropic Annealing (EA) and Look-ahead Modulation (LaM).

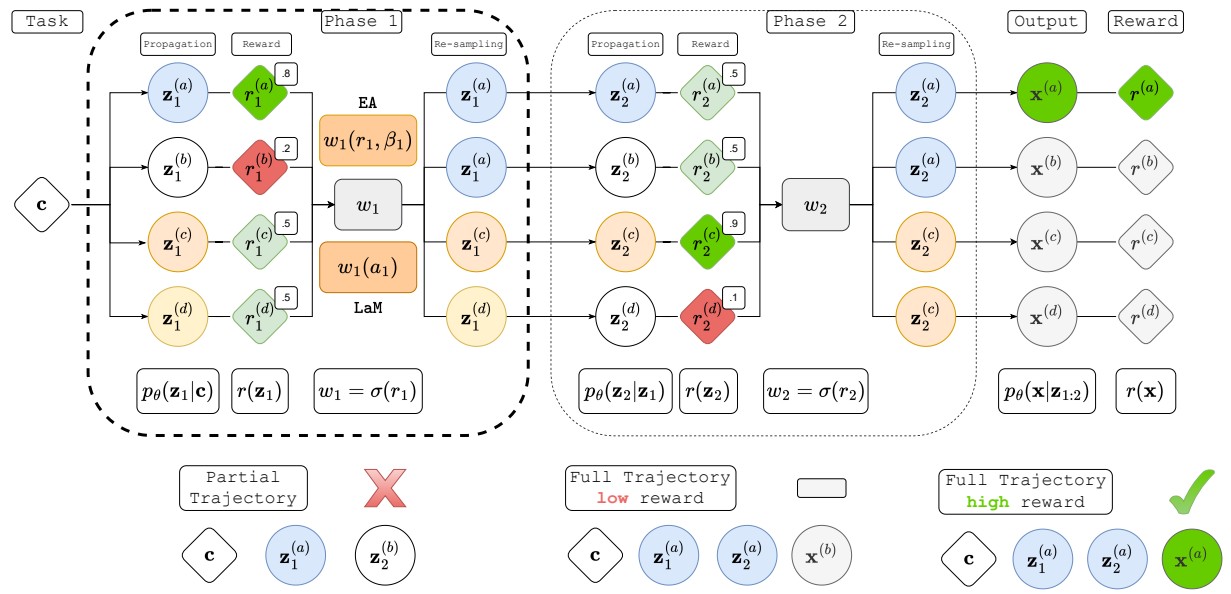

*(b)* Entropic Particle Filtering (ePF).

*Figure 33.* Entropic Particle Filtering Pipeline (ePF; Fig. 33b). Our method iteratively refines a set of $N$ particles (states $\mathbf{z}_t$) for a given prompt $\mathbf{c}$. In each step $t$, particles are propagated by sampling a language model, $p(\mathbf{z}_t|\mathbf{z}_{t-1}, \mathbf{c})$, and then weighted by a reward model, $r(\mathbf{z}_{1:t}, \mathbf{c})$. A resampling step, guided by these weights, approximates the posterior $p(\mathbf{z}_t|\mathbf{o}_t, \mathbf{c})$ to concentrate on promising candidates. After convergence, a final solution $\mathbf{x}$ is decoded via $p(\mathbf{x}|\mathbf{z}_{1:T}, \mathbf{c})$. To prevent premature convergence, we introduce two complementary mechanisms: *Entropic Annealing* (EA; Fig. 33a left), which preserves diversity by dynamically annealing the resampling distribution $w_t$ based on entropy, and *Look-ahead Modulation* (LaM; Fig. 33a right), which steers the search toward more promising future states. This dual strategy guides the algorithm toward regions of both high reward and high diversity, effectively mitigating convergence to suboptimal local solutions.

# I. Algorithms

---

**Algorithm 1** Particle Filtering (PF)

---

**Input:** Task Input $\mathbf{c}$, Particle Budget $N$, Steps $T$
**Dynamic Model:** $p(\mathbf{z}_t|\mathbf{z}_{t-1}, \mathbf{c})$
**Observation Model:** $p(\mathbf{o}_t|\mathbf{z}_t, \mathbf{c}) = \delta(s - g(\mathbf{z}_t, \mathbf{c}))$

// Initialization
Sample initial step: $\mathbf{z}_1 \sim p(\mathbf{z}_1|\mathbf{c})$
Initialize weights: $w_1 \leftarrow 1/N$
Initialize particles: $\mathcal{P} \leftarrow \{(\mathbf{z}_1^{(i)}, w_1^{(i)})\}_{i=1}^N$

// Sequential Importance Resampling
**for** step $t = 2$ to $T$ **do**
   **for** particle $i = 1$ to $N$ **do**
      // Propagation & Scoring
      Propagate state: $\mathbf{z}_t^{(i)} \sim p(\mathbf{z}_t|\mathbf{z}_{t-1}^{(i)}, \mathbf{c})$
      Compute score: $s_t^{(i)} \leftarrow g(\mathbf{z}_{1:t}^{(i)}, \mathbf{c})$
      Log-reward: $r_t^{(i)} \leftarrow \log s_t^{(i)}$
   **end for**
   // Resampling Step
   Compute weights: $w_t^{(i)} \leftarrow \exp(r_t^{(i)})/\sum_{j=1}^N \exp(r_t^{(j)})$
   Sample indices: $\{k_i\}_{i=1}^N \sim \text{Multinomial}(w_t)$
   Select particles: $\mathbf{z}_t \leftarrow \mathbf{z}_t[k_{1:N}]$
   Update particle set: $\mathcal{P} \leftarrow \mathcal{P} \cup \{(\mathbf{z}_t, w_t)\}$
**end for**

// Output
**Return** $\mathbf{x} \sim p(\mathbf{x}|\mathbf{z}_{1:T}, \mathbf{c})$

---

---

**Algorithm 2** Particle Filtering with Entropic Annealing (`ePF`)

---

**Input:** Task Input $\mathbf{c}$, Particle Budget $N$, Steps $T$
**Dynamic Model:** $p(\mathbf{z}_t|\mathbf{z}_{t-1}, \mathbf{c})$
**Observation Model:** $p(\mathbf{o}_t|\mathbf{z}_t, \mathbf{c}) = \delta(s - g(\mathbf{z}_t, \mathbf{c}))$

// Initialization
Sample initial state: $\mathbf{z}_1 \sim p(\mathbf{z}_1|\mathbf{c})$
Initialize weights: $w_1 \leftarrow 1/N$
Initialize particles: $\mathcal{P} \leftarrow \{(\mathbf{z}_1^{(i)}, w_1^{(i)})\}_{i=1}^N$

// Sequential Importance Resampling
**for** step $t = 2$ to $T$ **do**
    **for** particle $i = 1$ to $N$ **do**
        Propagate: $\mathbf{z}_t^{(i)} \sim p(\mathbf{z}_t|\mathbf{z}_{t-1}^{(i)}, \mathbf{c})$
        Score: $s_t^{(i)} \leftarrow g(\mathbf{z}_{1:t}^{(i)}, \mathbf{c})$
        Log-reward: $r_t^{(i)} \leftarrow \log s_t^{(i)}$
    **end for**
    // Entropic Annealing
    Compute weights: $w_t^{(i)} \leftarrow \exp(r_t^{(i)}) / \sum_{j=1}^N \exp(r_t^{(j)})$
    Compute ESS: $\mathrm{ESS}_n(t) \leftarrow \frac{1}{N} \left( \sum_{j=1}^N (w_t^{(j)})^2 \right)^{-1}$
    **if** $\mathrm{ESS}_n(t) < \tau$ **then**
        Anneal temperature: $\beta_t^{-1} \leftarrow \frac{(1-t/T)}{\mathrm{ESS}_n(t)}$
        Bound temperature: $\beta_t^{-1} \leftarrow \max(\beta_t^{-1}, 1)$
        Re-weight: $w_t^{(i)} \leftarrow \frac{\exp(r_t^{(i)} \cdot \beta_t)}{\sum_j \exp(r_t^{(j)} \cdot \beta_t)}$
    **end if**
    // Resampling (Systematic)
    Sample indices: $\{k_i\}_{i=1}^N \sim \mathrm{Systematic}(w_t)$
    Select particles: $\mathbf{z}_t \leftarrow \mathbf{z}_t[k_{1:N}]$
    Update set: $\mathcal{P} \leftarrow \mathcal{P} \cup \{(\mathbf{z}_t, w_t)\}$
**end for**

// Output
**Return** $\mathbf{x} \sim p(\mathbf{x}|\mathbf{z}_{1:T}, \mathbf{c})$

---

---

**Algorithm 3** Particle Filtering with Look-ahead Modulation (`PF w/ LaM`)

---

**Input:** Task Input $\mathbf{c}$, Particle Budget $N$, Steps $T$
**Dynamic Model:** $p(\mathbf{z}_t|\mathbf{z}_{t-1}, \mathbf{c})$
**Observation Model:** $p(\mathbf{o}_t|\mathbf{z}_t, \mathbf{c}) = \delta(s - g(\mathbf{z}_t, \mathbf{c}))$
**Parameters:** Threshold $\tau$

// Initialization
Sample initial step: $\mathbf{z}_1 \sim p(\mathbf{z}_1|\mathbf{c})$
Initialize weights: $w_1 \leftarrow 1/N$
Initialize particles: $\mathcal{P} \leftarrow \{(\mathbf{z}_1^{(i)}, w_1^{(i)})\}_{i=1}^N$

// Sequential Importance Resampling
**for** step $t = 2$ to $T$ **do**
  **for** particle $i = 1$ to $N$ **do**
    // Propagation & Base Scoring
    Propagate: $\mathbf{z}_t^{(i)} \sim p(\mathbf{z}_t|\mathbf{z}_{t-1}^{(i)}, \mathbf{c})$
    Score: $s_t^{(i)} \leftarrow g(\mathbf{z}_{t:T}^{(i)}, \mathbf{c})$
    Log-reward: $r_t^{(i)} \leftarrow \log s_t^{(i)}$
  **end for**
  Compute base weights: $w_t^{(i)} \leftarrow \frac{\exp(r_t^{(i)} \cdot \beta_t)}{\sum_j \exp(r_t^{(j)} \cdot \beta_t)}$
  // Look-ahead Modulation
  **for** particle $i = 1$ to $N$ **do**
    Auxiliary Propagation (1-step): $\mathbf{z}_s^{(i)} \sim p(\mathbf{z}_s|\mathbf{z}_t^{(i)}, \mathbf{c})$
    Auxiliary Scoring: $\tilde{r}_s^{(i)} \leftarrow g(\mathbf{z}_s^{(i)}, \mathbf{z}_{1:t}^{(i)}, \mathbf{c})$
    Modulate weight: $\tilde{w}_s^{(i)} \leftarrow w_t^{(i)} \cdot \tilde{r}_s^{(i)}$
  **end for**
  Normalize modulated weights: $w_t^{(i)} \leftarrow \tilde{w}_s^{(i)} / \sum_{j=1}^N \tilde{w}_s^{(j)}$
  // Resampling (Systematic)
  Sample indices: $\{k_i\}_{i=1}^N \sim \text{Systematic}(w_t)$
  Select particles: $\mathbf{z}_t \leftarrow \mathbf{z}_t[k_{1:N}]$
  Update particle set: $\mathcal{P} \leftarrow \mathcal{P} \cup \{(\mathbf{z}_t, w_t)\}$
**end for**

// Output
**Return** $\mathbf{x} \sim p(\mathbf{x}|\mathbf{z}_{1:T}, \mathbf{c})$

---

## J. Qualitative Results

We report a real example of early exploitation in Particle Filtering and exploration in Entropic Particle Filtering that guides the algorithms towards the correct solution for an AIME-2025 problem.

---

**Problem (AIME-2025)**

The parabola with equation $y = x^2 - 4$ is rotated $60°$ counterclockwise around the origin. The unique point in the fourth quadrant where the original parabola and its image intersect has $y$-coordinate $\frac{a - \sqrt{b}}{c}$, where $a$, $b$, and $c$ are positive integers, and $a$ and $c$ are relatively prime. Find $a + b + c$.

---

**Solution (Ground Truth)**

62

---

**Particle Filtering (3206 tokens)**

Okay, so I need to solve this problem where a parabola $y = x_2 - 4$ is rotated 60 degrees counterclockwise around the origin. Then, I have to find the unique point in the fourth quadrant where the original parabola and its image intersect. The y-coordinate of this point is given in the form $(a - \sqrt{b})/c$, and I need to find $a + b + c$.

Alright, let's start by understanding the problem step by step.

---

Step 1: Understand the original parabola

The original parabola is given by the equation:

$$y = x^2 - 4$$

This is a standard upward-opening parabola with vertex at $(0, -4)$. It opens upwards and is symmetric about the y-axis.

---

Step 2: Understand the rotation

We are told that the parabola is rotated $60°$ counterclockwise around the origin.

To rotate a point (x, y) counterclockwise by $\theta$, we use the rotation matrix:

$$\begin{bmatrix} \cos\theta & -\sin\theta \\ \sin\theta & \cos\theta \end{bmatrix}$$

So for $\theta = 60°$, we have:

$$\cos 60° = \frac{1}{2}, \quad \sin 60° = \frac{\sqrt{3}}{2}$$

Thus, the rotation matrix becomes:

$$\begin{bmatrix} \frac{1}{2} & -\frac{\sqrt{3}}{2} \\ \frac{\sqrt{3}}{2} & \frac{1}{2} \end{bmatrix}$$

So, if a point (x, y) is rotated $60°$ counterclockwise, the new coordinates (x', y') are:

$$x' = \frac{1}{2}x - \frac{\sqrt{3}}{2}y \quad y' = \frac{\sqrt{3}}{2}x + \frac{1}{2}y$$

—

Step 3: Find the image of the parabola after rotation

We want to find the equation of the image of the parabola $y = x_2 - 4$ after rotation.

Let's denote the original coordinates as (x, y), and the rotated coordinates as (x', y'). Then:

$$x' = \frac{1}{2}x - \frac{\sqrt{3}}{2}y \quad y' = \frac{\sqrt{3}}{2}x + \frac{1}{2}y$$

We can solve these equations for x and y in terms of x' and y':

Let's write them as a system:

$$\begin{cases} x' = \frac{1}{2}x - \frac{\sqrt{3}}{2}y \\ y' = \frac{\sqrt{3}}{2}x + \frac{1}{2}y \end{cases}$$

We can solve this system for x and y in terms of x' and y'.

Let me write the equations again:

$$\frac{1}{2}x - \frac{\sqrt{3}}{2}y = x' \quad (1) \qquad \frac{\sqrt{3}}{2}x + \frac{1}{2}y = y' \quad (2)$$

Multiply both equations by 2 to eliminate denominators:

$$x - \sqrt{3}y = 2x' \quad (1') \qquad \sqrt{3}x + y = 2y' \quad (2')$$

Now solve this system. Let's solve for x and y.

From (1'):

$$x = 2x' + \sqrt{3}y$$

Substitute into (2'):

$$\sqrt{3}(2x' + \sqrt{3}y) + y = 2y'$$

$$2\sqrt{3}x' + 3y + y = 2y'$$

$$2\sqrt{3}x' + 4y = 2y'$$

Divide both sides by 2:

$$\sqrt{3}x' + 2y = y'$$

Solve for y:

$$2y = y' - \sqrt{3}x' \Rightarrow y = \frac{y' - \sqrt{3}x'}{2}$$

Now substitute back into x:

$$x = 2x' + \sqrt{3} \cdot \frac{y' - \sqrt{3}x'}{2}$$

$$x = 2x' + \frac{\sqrt{3}y' - 3x'}{2}$$

$$x = \frac{4x' + \sqrt{3}y' - 3x'}{2} = \frac{x' + \sqrt{3}y'}{2}$$

So the coordinates (x, y) in terms of (x', y') are:

$$x = \frac{x' + \sqrt{3}y'}{2}, \quad y = \frac{y' - \sqrt{3}x'}{2}$$

—

Step 4: Find the image of the parabola

We want to find the image of the parabola $y = x_2 - 4$ after rotation. That is, for each point (x, y) on the original parabola, we find its image (x', y') and then find the equation of the new curve.

Let's write the original equation:

$$y = x^2 - 4$$

Now, we can express x and y in terms of x' and y' as above, and substitute into the original equation.

So:

$$y = x^2 - 4 \Rightarrow \frac{y' - \sqrt{3}x'}{2} = \left(\frac{x' + \sqrt{3}y'}{2}\right)^2 - 4$$

Let's compute the right-hand side:

$$\left(\frac{x' + \sqrt{3}y'}{2}\right)^2 = \frac{(x')^2 + 2\sqrt{3}x'y' + 3(y')^2}{4}$$

So:

$$\frac{y' - \sqrt{3}x'}{2} = \frac{(x')^2 + 2\sqrt{3}x'y' + 3(y')^2}{4} - 4$$

Multiply both sides by 4 to eliminate denominators:

$$2(y' - \sqrt{3}x') = (x')^2 + 2\sqrt{3}x'y' + 3(y')^2 - 16$$

Simplify left-hand side:

$$2y' - 2\sqrt{3}x' = (x')^2 + 2\sqrt{3}x'y' + 3(y')^2 - 16$$

Now bring all terms to one side:

$$(x')^2 + 2\sqrt{3}x'y' + 3(y')^2 - 16 - 2y' + 2\sqrt{3}x' = 0$$

This is the equation of the rotated parabola. So the image of the original parabola is:

$$(x')^2 + 2\sqrt{3}x'y' + 3(y')^2 - 16 - 2y' + 2\sqrt{3}x' = 0$$

—

Step 5: Find the intersection of the original and rotated parabola

We want to find the point where the original parabola and the rotated parabola intersect. That is, we solve the system:

$$y = x^2 - 4 \quad \text{(original)}$$
$$(x')^2 + 2\sqrt{3}x'y' + 3(y')^2 - 16 - 2y' + 2\sqrt{3}x' = 0 \quad \text{(rotated)}$$

But since we are looking for the **unique point in the fourth quadrant**, we can assume that (x', y') lies in the fourth quadrant. So $x' > 0, y' < 0$.

But maybe it's easier to work in terms of (x, y), and find where the original parabola and the rotated parabola intersect.

Alternatively, we can consider that the original parabola is $y = x_2 - 4$, and the rotated parabola is the image of that.

But maybe we can think in terms of coordinates: if we find the point (x, y) on the original parabola such that $(x', y')$ is also on the rotated parabola, and that $(x', y')$ is the image of (x, y).

So, for a point (x, y) on the original parabola, its image (x', y') is on the rotated parabola.

So, the point (x, y) lies on the original parabola, and the point (x', y') lies on the rotated parabola.

So, we can write:

1. $y = x^2 - 4$ 2. $(x')^2 + 2\sqrt{3}x'y' + 3(y')^2 - 16 - 2y' + 2\sqrt{3}x' = 0$

But since (x', y') is the image of (x, y), we can express x' and y' in terms of x and y.

From earlier, we had:

$$x' = \frac{1}{2}x - \frac{\sqrt{3}}{2}y y' = \frac{\sqrt{3}}{2}x + \frac{1}{2}y$$

So, substituting these into the rotated parabola equation:

$$\left(\frac{1}{2}x - \frac{\sqrt{3}}{2}y\right)^2 + 2\sqrt{3}\left(\frac{1}{2}x - \frac{\sqrt{3}}{2}y\right)\left(\frac{\sqrt{3}}{2}x + \frac{1}{2}y\right) +$$

$$3\left(\frac{\sqrt{3}}{2}x + \frac{1}{2}y\right)^2 - 16 - 2\left(\frac{\sqrt{3}}{2}x + \frac{1}{2}y\right) + 2\sqrt{3}\left(\frac{1}{2}x - \frac{\sqrt{3}}{2}y\right) = 0$$

This looks complicated, but we can try to simplify step by step.

Let me denote:

Let's compute each term:

1. First term: $\left(\frac{1}{2}x - \frac{\sqrt{3}}{2}y\right)^2 = \frac{1}{4}x^2 - \frac{\sqrt{3}}{2}xy + \frac{3}{4}y^2$

2. Second term: $2\sqrt{3} \cdot \left(\frac{1}{2}x - \frac{\sqrt{3}}{2}y\right)\left(\frac{\sqrt{3}}{2}x + \frac{1}{2}y\right)$

Let me compute the product inside:

$$\left(\frac{1}{2}x - \frac{\sqrt{3}}{2}y\right)\left(\frac{\sqrt{3}}{2}x + \frac{1}{2}y\right) = \frac{1}{2}x \cdot \frac{\sqrt{3}}{2}x + \frac{1}{2}x \cdot \frac{1}{2}y - \frac{\sqrt{3}}{2}y \cdot \frac{\sqrt{3}}{2}x - \frac{\sqrt{3}}{2}y \cdot \frac{1}{2}y$$

$$= \frac{\sqrt{3}}{4}x^2 + \frac{1}{4}xy - \frac{3}{4}xy - \frac{\sqrt{3}}{4}y^2$$

$$= \frac{\sqrt{3}}{4}x^2 - \frac{1}{2}xy - \frac{\sqrt{3}}{4}y^2$$

Multiply by $2\sqrt{3}$:

$$2\sqrt{3} \cdot \left(\frac{\sqrt{3}}{4}x^2 - \frac{1}{2}xy - \frac{\sqrt{3}}{4}y^2\right) = \frac{2\sqrt{3} \cdot \sqrt{3}}{4}x^2 - \frac{2\sqrt{3}}{2}xy - \frac{2\sqrt{3} \cdot \sqrt{3}}{4}y^2$$

$$= \frac{6}{4}x^2 - \sqrt{3}xy - \frac{6}{4}y^2 = \frac{3}{2}x^2 - \sqrt{3}xy - \frac{3}{2}y^2$$

3. Third term: $3\left(\frac{\sqrt{3}}{2}x + \frac{1}{2}y\right)^2 = 3\left(\frac{3}{4}x^2 + \frac{\sqrt{3}}{2}xy + \frac{1}{4}y^2\right) = \frac{9}{4}x^2 + \frac{3\sqrt{3}}{2}xy + \frac{3}{4}y^2$

4. Fourth term: $-16$

5. Fifth term: $-2\left(\frac{\sqrt{3}}{2}x + \frac{1}{2}y\right) = -\sqrt{3}x - y$

6. Sixth term: $2\sqrt{3}\left(\frac{1}{2}x - \frac{\sqrt{3}}{2}y\right) = \sqrt{3}x - 3y$

[the algorithm generates the stop token and does not find a solution]

---

**Entropic Particle Filtering (5472 tokens)**

Okay, so I need to solve this problem where a parabola $y = x_2 - 4$ is rotated 60 degrees counterclockwise around the origin. Then, I have to find the unique point in the fourth quadrant where the original parabola and its image intersect. The y-coordinate of this point is given in the form $(a - \sqrt{b})/c$, where a, b, c are positive integers, a and c are coprime,

and then find $a + b + c$.

First, let me recall what happens when a graph is rotated around the origin. Rotating a graph 60 degrees counterclockwise involves applying a rotation matrix to the coordinates of each point on the graph.

The rotation matrix for counterclockwise rotation by $\theta$ is:

$$\begin{pmatrix} \cos\theta & -\sin\theta \\ \sin\theta & \cos\theta \end{pmatrix}$$

For $\theta = 60°$, $cos(60°) = 0.5$, $sin(60°) = \sqrt{3}/2$.

So the rotation matrix becomes:

$$\begin{pmatrix} \frac{1}{2} & -\frac{\sqrt{3}}{2} \\ \frac{\sqrt{3}}{2} & \frac{1}{2} \end{pmatrix}$$

This matrix takes a point (x, y) and maps it to a new point (x', y'):

$$x' = \frac{1}{2}x - \frac{\sqrt{3}}{2}y \quad y' = \frac{\sqrt{3}}{2}x + \frac{1}{2}y$$

Now, the original parabola is $y = x_2 - 4$.

We need to find the image of this parabola under rotation. So, we can think of the image of the parabola as the set of points $(x', y')$ such that there exists some (x, y) on the original parabola, and $(x', y')$ is obtained by rotating (x, y) by $60°$ counterclockwise.

But to find the equation of the image of the parabola, we need to express x and y in terms of x' and y', and then substitute into the original equation.

Let me denote:

Let $(x, y)$ be a point on the original parabola, so $y = x^2 - 4$.

After rotation, $(x', y') = ((1/2)x - (\sqrt{3}/2)y, (\sqrt{3}/2)x + (1/2)y)$

We can solve this system for x and y in terms of x' and y'.

Let me write the equations:

1. $x' = \frac{1}{2}x - \frac{\sqrt{3}}{2}y$  2. $y' = \frac{\sqrt{3}}{2}x + \frac{1}{2}y$

Let me write this as a system of linear equations:

$$\begin{cases} \frac{1}{2}x - \frac{\sqrt{3}}{2}y = x' \\ \frac{\sqrt{3}}{2}x + \frac{1}{2}y = y' \end{cases}$$

To solve this system, we can use substitution or matrix methods. Let me solve for x and y in terms of x' and y'.

Let me multiply the first equation by 2 to eliminate denominators:

1. $x - \sqrt{3}y = 2x'$  2. $\sqrt{3}x + y = 2y'$

Now we have:

1. $x - \sqrt{3}y = 2x'$  2. $\sqrt{3}x + y = 2y'$

Let me solve this system using substitution or elimination.

Let me solve equation 1 for x:

From equation 1:

$x = 2x' + \sqrt{3}y$

Substitute into equation 2:

$\sqrt{3}(2x' + \sqrt{3}y) + y = 2y'$

Compute:

$2\sqrt{3}x' + 3y + y = 2y'$

Combine like terms:

$2\sqrt{3}x' + 4y = 2y'$

Divide both sides by 2:

$\sqrt{3}x' + 2y = y'$

Solve for y:

$2y = y' - \sqrt{3}x'$

$y = \frac{y' - \sqrt{3}x'}{2}$

Now substitute back into x:

From equation 1:

$x = 2x' + \sqrt{3}y$

Substitute y:

$x = 2x' + \sqrt{3} \cdot \frac{y' - \sqrt{3}x'}{2}$

Simplify:

$x = 2x' + \frac{\sqrt{3}y' - 3x'}{2}$

Combine terms:

$x = \frac{4x' + \sqrt{3}y' - 3x'}{2} = \frac{x' + \sqrt{3}y'}{2}$

So we have expressions for x and y in terms of x' and y':

$$x = \frac{x' + \sqrt{3}y'}{2}, \quad y = \frac{y' - \sqrt{3}x'}{2}$$

Now, since the original parabola is $y = x_2 - 4$, we can substitute these expressions into that equation.

Let me write that:

Original equation: $y = x^2 - 4$

Substitute:

$$\frac{y' - \sqrt{3}x'}{2} = \left(\frac{x' + \sqrt{3}y'}{2}\right)^2 - 4$$

Let me simplify this equation.

First, compute the right-hand side:

$$\left(\frac{x' + \sqrt{3}y'}{2}\right)^2 = \frac{(x')^2 + 2\sqrt{3}x'y' + 3(y')^2}{4}$$

So:

$$\frac{y' - \sqrt{3}x'}{2} = \frac{(x')^2 + 2\sqrt{3}x'y' + 3(y')^2}{4} - 4$$

Multiply both sides by 4 to eliminate denominators:

$$2(y' - \sqrt{3}x') = (x')^2 + 2\sqrt{3}x'y' + 3(y')^2 - 16$$

Expand the left side:

$$2y' - 2\sqrt{3}x' = (x')^2 + 2\sqrt{3}x'y' + 3(y')^2 - 16$$

Bring all terms to one side:

$$(x')^2 + 2\sqrt{3}x'y' + 3(y')^2 - 16 - 2y' + 2\sqrt{3}x' = 0$$

Now, group like terms:

- Terms with x'²: $(x')^2$ - Terms with y'²: $3(y')^2$ - Terms with x'y': $2\sqrt{3}x'y'$ - Terms with x': $2\sqrt{3}x'$ - Terms with y': $-2y'$ - Constant term: $-16$

So the equation becomes:

$$(x')^2 + 2\sqrt{3}x'y' + 3(y')^2 + 2\sqrt{3}x' - 2y' - 16 = 0$$

This is the equation of the image of the parabola. Now, we need to find the point of intersection between the original parabola and its image. So we need to solve this equation together with the original equation $y = x_2 - 4$. But we already used the original equation in the derivation. So the image of the parabola is this equation, and the original is $y = x_2 - 4$.

But since we are looking for the intersection of the original parabola and its image, we can set the two expressions for y equal?

Wait, but the image is a rotated version of the original. So the original parabola and its image intersect where the original parabola is mapped to itself.

Alternatively, perhaps it's easier to consider that the intersection points are points that are on both the original and the image parabola.

So, the original parabola is $y = x_2 - 4$.

The image of the parabola is given by the equation we derived:

$$(x')^2 + 2\sqrt{3}x'y' + 3(y')^2 + 2\sqrt{3}x' - 2y' - 16 = 0$$

But since $(x', y')$ are the coordinates of the image, and the original coordinates are $(x, y)$, and we have the expressions for x and y in terms of $(x', y')$, we can also write the original parabola equation as $y = x_2 - 4$. So, we can use the expressions for x and y in terms of x' and y' to substitute into the original equation.

Wait, but we already did that earlier and ended up with the above equation.

But actually, the image is a rotated version of the original, so the intersection points are points that are on both the original and the rotated parabola.

So the intersection points satisfy both:

1. $y = x^2 - 4$ 2. The rotated equation: $(x')^2 + 2\sqrt{3}x'y' + 3(y')^2 + 2\sqrt{3}x' - 2y' - 16 = 0$

But since we have expressions for x and y in terms of x' and y', we can write:

Original parabola: $y = x^2 - 4$

Image parabola: $(x')^2 + 2\sqrt{3}x'y' + 3(y')^2 + 2\sqrt{3}x' - 2y' - 16 = 0$

But since (x', y') are coordinates of the image, and (x, y) are coordinates of the original, so we can write:

From the rotation, we have:

$$x = \frac{x' + \sqrt{3}y'}{2}, \quad y = \frac{y' - \sqrt{3}x'}{2}$$

So substituting these into the original equation $y = x_2 - 4$ gives the equation we had earlier.

So we can now solve the equation:

$$\frac{y' - \sqrt{3}x'}{2} = \left(\frac{x' + \sqrt{3}y'}{2}\right)^2 - 4$$

Which we simplified earlier to:

$$(x')^2 + 2\sqrt{3}x'y' + 3(y')^2 + 2\sqrt{3}x' - 2y' - 16 = 0$$

But this is the equation of the image of the parabola. So the intersection points are the points that satisfy both the original parabola and this rotated equation.

But since we already used the original equation to derive this, the only way to find the intersection points is to solve the system:

1. $y = x^2 - 4$ 2. $(x')^2 + 2\sqrt{3}x'y' + 3(y')^2 + 2\sqrt{3}x' - 2y' - 16 = 0$

But since (x', y') are coordinates of the image, and (x, y) are coordinates of the original, and we have expressions for x and y in terms of x' and y', we can substitute those into the original equation.

But that's what we already did.

Alternatively, perhaps it's easier to think of this as a system of equations where we have the original parabola and the image of the parabola, and we need to find their intersection points.

Alternatively, since the image of the parabola is a rotated version, and the original parabola is $y = x_2 - 4$, the intersection points are points that lie on both.

But to find the intersection points, we can think of solving the system:

1. $y = x_2 - 4$ 2. The rotated parabola equation: $(x')^2 + 2\sqrt{3}x'y' + 3(y')^2 + 2\sqrt{3}x' - 2y' - 16 = 0$

But since (x', y') are coordinates of the image, and (x, y) are coordinates of the original, and we have expressions for x and y in terms of x' and y', we can substitute those expressions into the original equation.

Wait, but that's what we did earlier.

Alternatively, maybe I can think of this as a system of equations where we have the original parabola and the image parabola.

But since the image is a rotated version, the intersection points can be found by solving the system:

Original: $y = x^2 - 4$

Image: $y = x^2 - 4$ rotated by $60°$

But how do we find the image of the parabola? It's the set of points (x', y') such that there exists (x, y) on the original parabola, and (x', y') is the image of (x, y).

So, we can parametrize the original parabola.

Let me try a different approach. Since the original parabola is $y = x_2 - 4$, we can parametrize it as $(x, x_2 - 4)$, for all real x.

Then, the image of this parametrized point under rotation by $60°$ counterclockwise is:

Let's call the original point $(x, y) = (x, x_2 - 4)$. Then, the image is:

$x' = (1/2)x - (\sqrt{3}/2)y$

$y' = (\sqrt{3}/2)x + (1/2)y$

So, substituting $y = x_2 - 4$:

$x' = (1/2)x - (\sqrt{3}/2)(x_2 - 4)$

$y' = (\sqrt{3}/2)x + (1/2)(x_2 - 4)$

Then, we can write x' and y' in terms of x. Then, the image of the parabola is the set of points (x', y') where x = x, $y = x_2 - 4$, and the above expressions for x' and y'.

So, the image of the parabola is given by:

$x' = (1/2)x - (\sqrt{3}/2)(x_2 - 4)$

$y' = (\sqrt{3}/2)x + (1/2)(x_2 - 4)$

So, the image of the parabola is the set of points (x', y') where x' and y' are expressed in terms of x.

So, to find the intersection between the original and the image parabola, we can set the original parabola equation $y = x_2 - 4$ equal to the image parabola equation.

But since the image is parametrized by x, the image parabola is:

$y' = (\sqrt{3}/2)x + (1/2)(x_2 - 4) = (1/2)x_2 + (\sqrt{3}/2)x - 2$

$So, the image of the parabola is y' = (1/2)x_2 + (\sqrt{3}/2)x - 2$

So, to find the intersection points, we need to set the original $y = x_2 - 4$ equal to the image $y' = (1/2)x_2 + (\sqrt{3}/2)x - 2$.

So:

$x_2 - 4 = (1/2)x_2 + (\sqrt{3}/2)x - 2$

Bring all terms to one side:

$x_2 - 4 - (1/2)x_2 - (\sqrt{3}/2)x + 2 = 0$

Simplify:

$(1/2)x_2 - (\sqrt{3}/2)x - 2 = 0$

Multiply both sides by 2 to eliminate denominators:

$x_2 - \sqrt{3}x - 4 = 0$

So, we have a quadratic equation in x:

$x_2 - \sqrt{3}x - 4 = 0$

Solve this quadratic equation using the quadratic formula:

$x = [\sqrt{3} \pm \sqrt{((\sqrt{3})^2 + 16)}]/2$

Compute discriminant:

$(\sqrt{3})^2 + 16 = 3 + 16 = 19$

So,

$x = [\sqrt{3} \pm \sqrt{19}]/2$

So the x-coordinates of the intersection points are $([\sqrt{3}+\sqrt{19}]/2, [\sqrt{3}-\sqrt{19}]/2)$ and $([\sqrt{3}-\sqrt{19}]/2, [\sqrt{3}+\sqrt{19}]/2)$.

But we are interested in the point in the fourth quadrant, so we need to check which of these x and y coordinates are in the fourth quadrant.

Fourth quadrant is where x > 0 and y < 0.

Let me compute the two x-values:

1. $x_1 = [\sqrt{3} + \sqrt{19}]/2 \approx [1.732 + 4.358]/2 \approx 6.09/2 \approx 3.045 > 0$

2. $x_2 = [\sqrt{3} - \sqrt{19}]/2 \approx [1.732 - 4.358]/2 \approx (-2.626)/2 \approx -1.313 < 0$

So, only $x_1$ is positive, and $x_2$ is negative. So the x-coordinate of the point in the fourth quadrant is $x_1 = [\sqrt{3}+\sqrt{19}]/2$.

Now, let's compute the corresponding y-coordinate for this $x$. Since the original parabola is $y = x_2 - 4$.

So, $y = ([\sqrt{3} + \sqrt{19}]/2)^2 - 4$

Compute that:

First, compute $([\sqrt{3} + \sqrt{19}]/2)^2$:

$= [(\sqrt{3})^2 + 2\sqrt{3}\sqrt{19} + (\sqrt{19})^2]/4 = [3 + 2\sqrt{57} + 19]/4 = [22 + 2\sqrt{57}]/4 = [11 + \sqrt{57}]/2$

Then subtract 4:

$y = [11 + \sqrt{57}]/2 - 4 = [11 + \sqrt{57} - 8]/2 = [3 + \sqrt{57}]/2$

Wait, but this is positive, and we are looking for a point in the fourth quadrant, where y < 0.

But this result is positive, so this can't be the correct intersection point. So there must be a mistake here.

Wait, but we set the original $y = x_2 - 4$ equal to the image $y' = (1/2)x_2 + (\sqrt{3}/2)x - 2$, which is the image of the parabola.

But we got two solutions for x, one positive and one negative, but only the positive one gives a positive y. However, we need the point in the fourth quadrant, where y < 0.

Wait, so maybe we made a mistake in the sign when setting up the equations?

Let me double-check the image y':

Original $y = x_2 - 4$

Image $y' = (\sqrt{3}/2)x + (1/2)(x_2 - 4) = (1/2)x_2 + (\sqrt{3}/2)x - 2$

Yes, that's correct.

So setting original y = image y:

$x_2 - 4 = (1/2)x_2 + (\sqrt{3}/2)x - 2$

Bring everything to the left:

$x_2 - 4 - (1/2)x_2 - (\sqrt{3}/2)x + 2 = 0$

Simplify:

$(1/2)x_2 - (\sqrt{3}/2)x - 2 = 0$

Multiply by 2:

$x_2 - \sqrt{3}x - 4 = 0$

That's correct.

So solutions:

$$x = [\sqrt{3} \pm \sqrt{((\sqrt{3})^2 + 16)}]/2 = [\sqrt{3} \pm \sqrt{(3 + 16)}]/2 = [\sqrt{3} \pm \sqrt{19}]/2$$

So $x_1 = [\sqrt{3} + \sqrt{19}]/2 \approx$ positive, and $x_2 = [\sqrt{3} - \sqrt{19}]/2 \approx$ negative.

But then, when we compute $y = x_2 - 4$ for $x_1$, we get a positive y, and for $x_2$, we get:

$y = x_2^2 - 4$

But $x_2$ is negative, so $x_2^2$ is positive, so y = positive - 4.

But we need to check if this is negative.

Let me compute y for $x_2 = [\sqrt{3} - \sqrt{19}]/2$:

First, compute $x_2^2$:

$x_2 = [\sqrt{3} - \sqrt{19}]/2$

$x_2^2 = [(\sqrt{3} - \sqrt{19})^2]/4 = [3 - 2\sqrt{57} + 19]/4 = [22 - 2\sqrt{57}]/4 = [11 - \sqrt{57}]/2$

Then y = $x_2^2 - 4 = [11 - \sqrt{57}]/2 - 4 = [11 - \sqrt{57} - 8]/2 = [3 - \sqrt{57}]/2$

Now, $\sqrt{57} \approx 7.55$, so $3 - \sqrt{57} \approx 3 - 7.55 = -4.55, so y \approx -4.55/2 \approx -2.275$, which is negative.

So, the point with $x = [\sqrt{3} - \sqrt{19}]/2 and y = [3 - \sqrt{57}]/2$ is in the fourth quadrant.

Therefore, the unique point in the fourth quadrant is $(x, y) = ([\sqrt{3} - \sqrt{19}]/2, [3 - \sqrt{57}]/2)$

But the question says the y-coordinate is of the form $(a - \sqrt{b})/c$, where a, b, c are positive integers, a and c are coprime.

So let's write the y-coordinate:

$y = [3 - \sqrt{57}]/2$

So here, $a = 3$, $b = 57$, $c = 2$.

But 3 and 2 are coprime. So $a + b + c = 3 + 57 + 2 = 62$.

But let me check if that's correct. Wait, the question says "the unique point in the fourth quadrant where the original parabola and its image intersect has y-coordinate $(a - \sqrt{b})/c$".

So in our case, the y-coordinate is $[3 - \sqrt{57}]/2$, so a = 3, b = 57, c = 2.

But 3 and 2 are coprime, so that's good.

Therefore, the answer is $3 + 57 + 2 = 62$.

But let me check once again to make sure I didn't make any mistakes.

Original parabola: $y = x_2 - 4$

Image parabola: $y' = (1/2)x_2 + (\sqrt{3}/2)x - 2$

Set equal:

$x_2 - 4 = (1/2)x_2 + (\sqrt{3}/2)x - 2$

Multiply by 2:

$2x_2 - 8 = x_2 + \sqrt{3}x - 4$

Bring all terms to left:

$2x_2 - 8 - x_2 - \sqrt{3}x + 4 = 0 \rightarrow x_2 - \sqrt{3}x - 4 = 0$

Solutions: $x = [\sqrt{3} \pm \sqrt{(3+16)}]/2 = [\sqrt{3} \pm \sqrt{19}]/2$

For $x = [\sqrt{3} - \sqrt{19}]/2$, compute $y = x_2 - 4$:

$x = [\sqrt{3} - \sqrt{19}]/2$

$x_2 = [(\sqrt{3} - \sqrt{19})^2]/4 = [3 - 2\sqrt{57} + 19]/4 = [22 - 2\sqrt{57}]/4 = [11 - \sqrt{57}]/2$

Then $y = [11 - \sqrt{57}]/2 - 4 = [11 - \sqrt{57} - 8]/2 = [3 - \sqrt{57}]/2$

Yes, so that is correct.

So the y-coordinate is $[3 - \sqrt{57}]/2$, so a = 3, b = 57, c = 2, and they are coprime.

Thus, $a + b + c = 3 + 57 + 2 = 62$.

Final Answer $\boxed{62}$

# K. Details

*Table 17.* Models.

| Models | Size | Mode | Hugging Face Path |
|--------|------|------|-------------------|
| Qwen2.5 | 1.5B | Instruct | `Qwen/Qwen2.5-1.5B-Instruct` |
| Qwen2.5 | 7B | Instruct | `Qwen/Qwen2.5-7B-Instruct` |
| Qwen3 | 0.6B | w/o Thinking | `Qwen/Qwen3-0.6B` |
| Qwen3 | 1.7B | w/o Thinking | `Qwen/Qwen3-1.7B` |
| Llama3.2 | 1B | Instruct | `meta-llama/Llama-3.2-1B-Instruct` |
| Llama3.1 | 8B | Instruct | `meta-llama/Llama-3.1-8B-Instruct` |

**Datasets** We focus our attention on six math benchmarks.

GSM8K: A dataset of 8,500 high-quality, linguistically diverse grade school math word problems created to test multi-step reasoning.

MATH500: A collection of 500 challenging high-school competition math problems, each with detailed step-by-step solutions.

DEEPMATH: A dataset of advanced mathematical problems and their corresponding solutions.

OMNIMATH: A large-scale, multilingual benchmark featuring math problems covering a vast spectrum of mathematical topics and complexities.

AIME 2024: A dataset containing the 30 problems from the two AIME I and AIME II competitions held in 2024. It is not formally split into training and testing sets. Instead, it serves as a direct evaluation set to test the mathematical reasoning capabilities of advanced AI models.

AIME 2025: Similar to its predecessor, the AIME25 dataset is comprised of the problems from the 2025 AIME competitions and is also a smaller, focused dataset used for benchmarking. Each sample in these datasets includes the problem text, a detailed solution, and the final numerical answer.

*Table 18.* Datasets.

| Dataset | Split | Sample | Hugging Face Path |
|---------|-------|--------|-------------------|
| GSM8K | test (subset) | 500 | `openai/gsm8k-main` |
| MATH500 | test | 500 | `HuggingFaceH4/MATH-500` |
| DEEPMATH | subset | 500 | `zwhe99/DeepMath-103K` |
| OMNIMATH | test (subset) | 500 | `KbsdJames/Omni-MATH` |
| AIME 2024 | train | 30 | `Maxwell-Jia/AIME_2024` |
| AIME 2025 | train | 30 | `MathArena/aime_2025` |
| AIME-24-25 | train | 60 | `Maxwell-Jia/AIME_2024,MathArena/aime_2025` |

*Table 19.* Hyper-parameters for the main experiments.

|  | Qwen2.5-1.5B-Instruct | Qwen2.5-7B-Instruct | Qwen3-0.6B | Qwen3-1.7B |
|---|---|---|---|---|
| Budgets $N$ | 2,4,8,16,32 | 2,4,8,16,32 | 2,4,8,16,32 | 2,4,8,16,32 |
| PRM | Qwen2.5-Math-PRM-7B | Qwen2.5-Math-PRM-7B | Qwen2.5-Math-PRM-7B | Qwen2.5-Math-PRM-7B |
| Threshold $\tau$ | 0.5 | 0.5 | 0.5 | 0.5 |
| Repetition Penalty Temperature | 0.7 | 0.7 | 0.7 | 0.7 |
| TopK | 20 | 20 | 20 | 20 |
| TopP | 0.8 | 0.8 | 0.8 | 0.8 |
| Max Steps | 300 | 300 | 300 | 300 |
| Max Tokens/Step | 512 | 512 | 512 | 512 |
| Max Seq Tokens | 4k/12k | 4k/12k | 4k/12k | 4k/12k |

