# OpenReview forum: "Mitigating Premature Exploitation in Particle-based Monte Carlo for Inference-Time Scaling"
_ICML.cc/2026/Conference — ICML 2026 regular_

### Official Review · Reviewer_6KaF · 2026-03-12

**Soundness:** 3
**Presentation:** 3
**Significance:** 2
**Originality:** 2
**Overall Recommendation:** 4
**Confidence:** 4

**Summary:**

The paper studies premature exploitation in particle filtering (PF) for inference-time scaling of large language models (LLMs) guided by process reward models (PRMs). It identifies PRM overconfidence at early reasoning steps as a key cause of particle impoverishment. To address this issue, the authors propose Entropic Particle Filtering (ePF), which combines two mechanisms: (i) Entropic Annealing (EA), an ESS/entropy-driven temperature adaptation that flattens overconfident resampling distributions to preserve particle diversity, and (ii) Look-ahead Modulation (LaM), a one-step auxiliary evaluation of successor states inspired by auxiliary particle filtering (APF) to mitigate myopic resampling. Experiments on several mathematical reasoning benchmarks (GSM8K, MATH500, DeepMath, OmniMath, and AIME-24/25) using multiple Qwen models show that ePF consistently outperforms standard PF and common baselines, with particularly strong improvements under limited compute budgets.

**Compliance With Llm Reviewing Policy:**

Affirmed.

**Final Justification:**

Thank you to the authors for their response and experiments, I will keep my score.

**Key Questions For Authors:**

1. Please clarify the exact definitions used in code for ESS, normalized ESS, entropy, and the β_t schedule. The equations in the text suggest inconsistencies.
2. How sensitive is performance to the specific β_t functional form and to the ESS threshold (e.g., 0.5) at which EA/LaM are activated? Could you provide a sensitivity or ablation plot in the main paper?
3. Final selection uses the same PRM used for stepwise guidance. Can you report results with an outcome-level verifier or an ORM-based selection to quantify any selection bias due to PRM preferences?
4. Do EA and LaM retain their benefits when pairing with better-calibrated PRMs (e.g., uncertainty-aware)? An ablation with a more calibrated PRM would help assess complementarity vs redundancy.

**Limitations:**

yes

**Strengths And Weaknesses:**

# Strengths
- The paper clearly motivates the problem (PRM miscalibration leading to high-variance resampling and particle collapse) and articulates the design goals of the proposed approach.
- Introduces a simple yet principled diversity-preserving mechanism that adapts the resampling temperature using ESS/entropy metrics, directly addressing particle impoverishment caused by PRM overconfidence.
- Proposes Look-ahead Modulation (LaM), which adapts ideas from auxiliary particle filtering to LLM inference by incorporating a one-step forward-looking evaluation to mitigate myopic reward signals.
- The combination of EA + LaM provides a clear exploration–exploitation balance while remaining lightweight and easy to integrate into existing particle filtering pipelines.
- Evaluated across multiple mathematical reasoning benchmarks and model sizes, including challenging AIME-24/25 settings where premature exploitation is particularly severe.
- Demonstrates consistent performance improvements over PF and common baselines, especially under small compute budgets.
- Includes informative intrinsic analyses (ESS, entropy, weight variance, response-length distributions) that connect algorithmic behavior with downstream performance and help explain observed improvements.

# Weaknesses
- The Effective Sample Size (ESS) is incorrectly defined as $\sum (w_i)^{-2}$ in several places, rather than the standard $(\sum w_i)^2/(\sum w_i^2)$. As the temperature schedule $\beta_t$ relies on this value, these inconsistencies raise significant concerns regarding implementation correctness and reproducibility.
- Both EA and LaM are conceptually very similar to established Sequential Monte Carlo (SMC) techniques, specifically adaptive tempering and Auxiliary Particle Filters (APF). The paper lacks a rigorous technical comparison to these frameworks, making the contribution appear more as an application adaptation than a new theoretical advancement in SMC.
- Several evaluations rely on small subsets (e.g., $n=128$). Larger-scale evaluations are necessary to ensure the observed gains are statistically robust and generalizable across the benchmarks.

- If the ESS is unnormalized or incorrectly inverted, the scaling $\beta_t^{-1} = \frac{N}{ESS(t)}(1 - \frac{t}{T})$ will not behave as intended. Please provide the exact implementation formula.
- LaM Derivation: While $a_t^i \propto w_t^i \cdot \tilde{r}_s^i$ is a plausible surrogate for APF auxiliary weights, a formal derivation connecting this to APF theory (including a discussion on adjustment weights and variance properties) would significantly strengthen the work.
- In line 216:  "As $t \to T$, $\beta_t$ gradually anneals to 1...". With your formula, $1 / \beta_t \to 0$, so $\beta_t \to \infty$?

---

> ### Author Rebuttal · Authors · 2026-03-27
>
> **Reviewer**:
>
> > The Effective Sample Size (ESS)
>
> **Authors**: We apologize for this typographical error, which was inadvertently introduced in the main text while condensing the manuscript. Our actual theoretical foundation and implementation use the standard, correct formulation: $ESS(t)=\frac{1}{\sum_{i=1}^{N}(w_{t}^{i})^{2}}$ for normalized $w_{t}$. The extended derivation in `Appendix B` accurately reflects this. We have corrected all instances of this typo in the revised manuscript to ensure reproducibility. Thank you for catching this.
>
> --------
>
> **Reviewer**:
>
> > Both EA and LaM are conceptually very similar to established SMC techniques
>
> **Authors**: We agree that our methods are deeply rooted in classic SMC literature, which we discuss extensively in Appendix A. Our primary contribution is indeed adapting these principles and algorithms to the unique challenges of LLM reasoning and inference-time scaling - specifically addressing PRM overconfidence. By translating these frameworks into a highly effective, scalable application for LLMs, and providing empirical evidence of variance reduction (Appendix B) and diversity preservation (Appendix G.12), we aim to provide the inference community with a robust, practical tool.
>
> -------
>
> **Reviewer**:
>
> > Several evaluations rely on small subsets
>
> **Authors**: To demonstrate statistical robustness, we provide numerical confidence intervals for our evaluations on the math datasets (GSM8K, MATH500, DEEPMATH, and OMNIMATH) in Appendix G.3, Table 9. For the evaluations on the AIME 2024 and AIME 2025 datasets, we visualize the variance directly as shaded confidence regions across all budgets in Fig 6/7/9.
>
> --------
>
> **Reviewer**:
> > LaM Derivation
>
> **Authors**: While we currently provide a formal derivation for the connection between ESS and the variance of the resampling distribution in Appendix B, we agree that formally mapping LaM to APF theory would strengthen the theoretical underpinnings. We are actively working on extending this specific variance analysis for the single and multi-step look-ahead variants for future work.
>
> --------
>
>
> **Reviewer**:
> > In line 216: "As $t\rightarrow T$, $\beta_{t}$ gradually anneals to 1...".
>
> **Authors**: The temperature $\beta_{t}^{-1}$ is lower-bounded by $1$: $\beta_{t}^{-1}\leftarrow \max(\beta_{t}^{-1}, 1)$. Furthermore, by default, ePF is only activated for the first 50% of the generation steps to focus on early exploration. The precise temperature schedules and their behaviors over sampling steps are visualized and ablated in Appendix G.10 and Figure 26.
>
> ---------
>
> **Reviewer**:
> > Please clarify the exact definitions used in code for ESS, normalized ESS, entropy, and the $\beta_t$ schedule.
>
> **Authors**:
> The exact definitions used in our codebase match the derivations provided in `Appendix B`.
>
> ----------
>
> **Reviewer**:
> > How sensitive is performance to the specific $\beta_t$ functional form
>
> **Authors**:
> We explore this sensitivity in Appendix G.10, which compares three different functional forms for the temperature schedule: Linear, ESS-based, and Entropy-based. Figure 27 displays the ablation results across multiple datasets (AIME 2024, AIME 2025) and model sizes. Across these diverse configurations, the ESS-based schedule consistently performs the best, demonstrating robustness without hyper-specific tuning.
>
> ---------
>
>
> **Reviewer:**
> > Can you report results with an outcome-level verifier or an ORM-based selection to quantify any selection bias due to PRM preferences?
>
> **Authors:**
> This is a great point. To completely isolate the quality of the generated trajectories from any potential PRM or ORM selection bias, we utilize the pass@B metric, detailed in Appendix G.12. This metric evaluates the diversity of correct solutions found across the entire budget range, effectively removing the reward model from the final selection step. Using Qwen3-1.7B, the results are as follows:
>
> | Method | AIME 2024 | AIME 2025 |
> | :--- | :--- | :--- |
> | PF | 0.36 | 0.36 |
> | ePF | 0.56 | 0.46 |
> | ePF-LaM | 0.50 | 0.46 |
>
> This intrinsic analysis proves that ePF inherently discovers a wider, more diverse set of correct solutions compared to standard PF, confirming the behavioral advantages shown via intrinsic metrics in Figures 1 and 9.
>
> -----------
>
>
> **Reviewer**:
>
> > Do EA and LaM retain their benefits when pairing with better-calibrated PRMs
>
> **Authors**: While a perfectly calibrated PRM would naturally improve baseline PF, building such models remains exceptionally difficult. To address this, we compared ePF against IAS-C in Appendix G.8, Table 16(c). IAS-C is an inference scaling method that specifically relies on an uncertainty-calibrated PRM. On the combined AIME-24-25 dataset, ePF still substantially outperformed IAS-C (Score: 26.4 vs 18.3). This demonstrates that ePF is not redundant when calibration improves; it acts as a robust, complementary mechanism for online rebalancing.

---

> > ### Author Rebuttal · Reviewer_6KaF · 2026-04-02
> >
> > Thank you to the authors for their response and experiments, I will keep my score.

---

> > > ### Author Response · Authors · 2026-04-02
> > >
> > > Thank you for the constructive feedback and discussion. Please see also the responses to Reviewer `Mzs2` and `BE8T`, where we provide additional analysis and experiments on:
> > >
> > > - new datasets (AIME 2026, HealthBench)
> > > - new model (Qwen3-4B-Instruct-2507)
> > > - new domain without explicit verification (healthcare)
> > > - different PRM (Llama3.1-8B-PRM)

---

### Official Review · Reviewer_Mzs2 · 2026-03-12

**Soundness:** 2
**Presentation:** 3
**Significance:** 2
**Originality:** 2
**Overall Recommendation:** 5
**Confidence:** 3

**Summary:**

The authors propose an improved particle-filtering based inference time scaling strategy by adaptively varying the temperature used for sampling based on the effective sample size at any given intermediate stage. This mitigates the problem of degenerate solutions in sequential monte-carlo styled methods especially when combined with partial reward models. This work applies the framework in the domain of language where they highlight that this adaptive temperature framework leads to improved performance across the board on standard particle filtering for math-based tasks. They further also propose a look-ahead strategy that allows steering of the generations and resampling based on look-ahead scores, alleviating the PRMs nature of greedily pruning multiple potentially useful trajectories at earlier steps.

**Compliance With Llm Reviewing Policy:**

Affirmed.

**Final Justification:**

All my concerns have been satisfactorily addressed and I am happy to recommend accept.

**Key Questions For Authors:**

- Is there a reason why the benchmark numbers for PF, and others, are different from Puri et al 2025?

**Limitations:**

yes

**Strengths And Weaknesses:**

**Soundness**: The work is technically sound and the authors provide extensive experiments to highlight that the proposed method, entropic particle filtering, along with the lookahead framework leads to improvement in performance.

**Presentation**: The submission is clearly written and well structured. It highlights the limitations of existing methods, illustrates how the proposed method aims to alleviate this and also openly describes how different parts of the proposed method can introduce bias; after which they rigorously evaluate to showcase that even then they see improved performance.

**Significance**: While the work does demonstrate improvements over the basic particle filtering methods, it does not show this at scale. From my understanding, many such particle based methods, and even beam search, are not generally used in large-scale systems but at the same time, this work seems to suggest that these methods tend to be quite competitive with BoN sampling. Please refer to weaknesses for additional points.

**Originality**: Alleviating degenerate solutions in particle filtering methods has been a long-standing topic of interest, and a number of existing works devise metrics based on ESS for sampling or other interventions. However, I don't think that this should in general be a negative of this work, since it does show that the proposed method is simple but can show consistent improvements over baselines.

**Strengths**

- The paper proposes a novel method based on look-ahead and adaptive temperature resampling in SMC, thus improving on the particle filtering baseline.
- Experiments are conducted on both easy and hard tasks, going from GSM8k to AIME.
- A number of different baselines are considered, signifying extensive evaluation.

**Weaknesses**

- The experiments are done predominantly in math domain which reduces the applicability of the method. From the draft, it is unclear if the gains extend to other domains: e.g. code, general reasoning, alignment or factual knowledge based tasks. Having a diverse test of tasks would be a testament for ePF to be a good candidate for inference time scaling; especially since it is a solely inference only method making experiments cheaper in comparison to training.
- The scale of models considered is quite limiting and it would have been useful to see experiments on larger and more recent models (e.g. Qwen3-4B-Instruct-2507 or GPT-OSS-20B) as a testament to whether the proposed method scales well or not.
- The authors do not provide estimates of variance in their evaluation. Given some of the test datasets like AIME are quite small, it would be good to have an estimate of variance to see the significance of the results (in the tables).

---

> ### Author Rebuttal · Authors · 2026-03-27
>
> **Reviewer**:
>
> > many such particle based methods, and even beam search, are not generally used in large-scale systems
>
> **Authors**: Thank you for highlighting this point. The bottleneck is engineering, not algorithmic: frameworks like vLLM prioritize continuous batching over native step-level interventions. However, deployment is feasible with targeted adjustments—we successfully ran all experiments on vLLM servers. We hope ePF's strong empirical results accelerate native framework support for step-level ITS.
>
> --------
>
> **Reviewer**:
>
> > Alleviating degenerate solutions in particle filtering methods has been a long-standing topic of interest
>
> **Authors**: We agree ESS-based interventions are established in SMC literature, which we discuss in Appendix A and compare against in Appendix F.1 and F.2. Despite this, the LLM inference community lacks a generalized, high-performing method for step-level generation at scale. ePF practically bridges this gap, backed by strong empirical results.
>
> --------
>
> **Reviewer**:
>
> > Having a diverse test of tasks would be a testament for ePF to be a good candidate for inference time scaling;
>
> **Authors**: We agree that validating ePF in open-ended, non-math domains strengthens its broad applicability, so we conducted new experiments on realistic medical reasoning. See also Finance and Chemistry tasks (Appx G.2).
>
> We selected a stratified subset of 100 multi-turn QA pairs from the [HealthBench](https://openai.com/index/healthbench/) hard set (OpenAI, May 2025). Lacking a domain-specific PRM or ORM, we used MedGemma-4B as both the generator and scorer, and MedGemma-27B as the final evaluator. For intermediate scoring, we conditioned the generator on physician-provided rubrics, leveraging in-context learning. This setup tests ePF in a novel domain without explicit verification or a parametric PRM.
> The results below demonstrate that ePF significantly outperforms standard PF and BoN, reinforcing its effectiveness as a general inference mechanism:
>
> **HealthBench (Avg@3, N@8, 100 samples)**
> | Name | HealthBench Score x100 | Relative Gain |
> | :--- | :--- | :--- |
> | Base Sampling | 27.6 |
> | BoN | 30.3 | +9.7% |
> | PF | 29.8 | +8.0% |
> | ePF | 31.5 | +14.2% |
>
> We also break down results by specific HealthBench themes—critical sub-categories for medical reasoning. As shown below, ePF outperforms both BoN and PF by a large margin across most categories.
>
> **Relative Improvement by Theme (vs Base Sampling)**
> | Name | Emergency Referral | Complex Responses | Context Seeking | Health Data Tasks | Hedging |
> | :--- | :--- | :--- | :--- | :--- | :--- |
> | BoN | 28% | -1% | 25% | 18% | 36% |
> | PF | 26% | -16% | 24% | 8% | 29% |
> | ePF | 81% | 4% | 16% | 21% | 41% |
>
> -----------------
>
>
> **Reviewer**:
> > it would have been useful to see experiments on larger and more recent models (e.g. Qwen3-4B-Instruct-2507 or GPT-OSS-20B)
>
> **Authors**: While we believe that the results in our manuscript consistently demonstrate ePF's strengths, we agree that testing newer models adds value. To strictly prevent test-set contamination, we ran additional experiments on the new AIME 2026 dataset using Qwen3-4B-Instruct-2507 and our existing Qwen2.5-7B-Instruct model. We evaluated budgets $N\in\{4,8,16\}$ across 3 seeds, with the results provided below:
>
> **AIME 2026 Evaluation (3 Seeds)**
>
> | Method | N=1 (Base) | N=4 | N=8 | N=16 |
> | :--- | :--- | :--- | :--- | :--- |
> | Qwen3-4B-In  | 52.1% ± 1.1% | - | - | - |
> | Qwen3-4B-In PF | - | 65.0% ± 5.3% | 70.0% ± 1.3% | 73.3% ± 1.3% |
> | Qwen3-4B-In ePF | - | 68.3% ± 1.3% | 78.0% ± 3.8% | 80.0% ± 2.5% |
> |  | | | | |
> | Qwen2.5-7B-In | 6.2% ± 0.9% | - | - | - |
> | Qwen2.5-7B-In PF | - | 13.3% | 16.7% | 16.7% |
> | Qwen2.5-7B-In ePF | - | 11.7% ± 1.3% | 23.3% ± 1.3% | 24.5% ± 1.3% |
>
> Even using a more modern model and recent math dataset, results are consistent, and ePF tends to outperform standard PF for most budgets.
>
> ---------
>
> **Reviewer**:
> > The authors do not provide estimates of variance in their evaluation.
>
> **Authors**: We do provide confidence intervals: numerical ranges for GSM8K, MATH500, DEEPMATH, and OMNIMATH are in Appendix G.3, Table 9 , and shaded regions for AIME 2024 and AIME 2025 are in Figures 6, 7, and 9. We will reference these more explicitly in the main text.
>
> -------
>
> **Reviewer**:
>
> > Is there a reason why the benchmark numbers for PF, and others, are different from Puri et al 2025?
>
> **Authors**: The differences stem from two main factors:
>
> - Model Choice: Puri et al. evaluated Math-specialist models, whereas we used standard, generalist Instruct models for broader applicability.
>
> - Evaluation Protocol: Answer parsing is notoriously brittle, and different heuristic verifiers or library versions yield varying scores. To guarantee a rigorous comparison, we completely re-ran all baselines in our pipeline using the strict math_verify library, which provides a more conservative performance estimate.

---

> > ### Author Rebuttal · Reviewer_Mzs2 · 2026-04-02
> >
> > Thanks a lot for the response. Most of my concerns have been addressed; I just had a couple more that I would like to ask. Could the authors could also provide BoN numbers for the new experiments. In addition, is there a time-matched baseline? For example in Table 6, we see that the proposed method is roughly $3\times$ slower so does it perform better than an ORM on $3\times$ more generations from base sampling?

---

> > > ### Author Response · Authors · 2026-04-03
> > >
> > > Thank you for the constructive discussion.
> > >
> > > **Reviewer**:
> > >
> > > > Could the authors could also provide BoN numbers for the new experiments.
> > >
> > > **Authors**:
> > >
> > > We have already provided Best-of-N results for the new domain and dataset (HealthBench and its corresponding themes). Below is the updated table for AIME 2026 and Qwen3-4B-Instruct-2507, which now includes the requested Best-of-N results:
> > >
> > > | Method |  N=1 | N=4 | N=8 | N=16 |
> > > |---|---|---|---|---|
> > > | Single Sample (qwen3-4b) |  52.1\% ± 1.1\% | - | - | - |
> > > | BoN (qwen3-4b) | - |  56.7\% ± 2.6\% | 63.3\% ± 2.4\% | 68.3\% ± 1.7\% |
> > > | PF (qwen3-4b) | - | 65.0\% ± 5.3\% | 70.0\% ± 1.3\% |  73.3\% ± 1.3\% |
> > > | ePF (qwen3-4b) | - | 68.3\% ± 1.3\% | 78.0\% ± 3.8\% |  80.0\% ± 2.5\% |
> > > |     |      |    |      |
> > > | Single Sample (qwen2.5-7b) | 6.2\% | - | - | - |
> > > | BoN (qwen2.5-7b) | - | 10.0\% | 16.7\% | 20.1\% |
> > > | PF (qwen2.5-7b) | - | 13.3\% | 16.7\% | 16.7\% |
> > > | ePF (qwen2.5-7b) | - | 11.7\% | 23.3\% | 24.5\% |
> > >
> > > Please also refer to our response to Reviewer `BE8T` for additional results using weaker PRM models.
> > >
> > > ---------
> > >
> > > **Reviewer**:
> > >
> > > > does it perform better than an ORM on 3x more generations from base sampling?
> > >
> > > **Authors**:
> > >
> > >
> > > Thank you for raising this point. The tables below compare the performance of ePF (using 4 or 8 particles) against BoN (using 16 or 32 particles - **4x more generations**) across different datasets and models:
> > >
> > > | Method |  MATH500 (128 samples) | DEEPMATH (128 samples) |
> > > |---|---|---|
> > > | BoN@32 (qwen2.5-1.5b) |  0.56      | 0.19  |
> > > | ePF@8 (qwen2.5-1.5b)    |  **0.62** | 0.18  |
> > > | ePF@32 (qwen2.5-1.5b)    |  0.66 | 0.25 |
> > >
> > > | Method |  DEEPMATH (128 samples) |
> > > |---|---|
> > > | BoN@16 (qwen2.5-7b) |  0.276   |
> > > | ePF@4 (qwen2.5-7b)    |  **0.290**   |
> > > | ePF@16 (qwen2.5-7b)    |  0.342 |
> > >
> > >
> > > | Method |  AIME 2025 |
> > > |---|---|
> > > | BoN@32 (qwen2.5-1.5b) | 0.06    |
> > > | ePF@8 (qwen2.5-1.5b)    | **0.10**    |
> > > | ePF@32 (qwen2.5-1.5b)    | 0.13  |
> > >
> > > | Method |  AIME 2025 |
> > > |---|---|
> > > | BoN@16 (qwen2.5-7b) | 0.13    |
> > > | ePF@4 (qwen2.5-7b)    | **0.23**    |
> > > | ePF@16 (qwen2.5-7b)    | 0.26  |
> > >
> > > | Method |  AIME 2024 |
> > > |---|---|
> > > | BoN@32 (qwen3-0.6b) | 0.10    |
> > > | ePF@8 (qwen3-0.6b)    | 0.10   |
> > > | ePF@32 (qwen3-0.6b)  | 0.20  |
> > >
> > > | Method |  AIME 2024 |
> > > |---|---|
> > > | BoN@16 (qwen3-1.7b) | 0.16    |
> > > | ePF@4 (qwen3-1.7b)    | **0.20**    |
> > > | ePF@16 (qwen3-1.7b)  | 0.26  |
> > >
> > > Across multiple datasets (MATH500, DEEPMATH, AIME2024, AIME2025), models (qwen2.5-1.5b, qwen2.5-7b, qwen3-0.6b, qwen3-1.7b), and scales (4/16 and 8/32, maintaining a 4x ratio), ePF significantly outperforms BoN with ORM in five out of seven experiments and performs comparably in the remaining two. These results reinforce the ablation study in Appendix E, demonstrating that **ePF is an effective method for inference-time scaling, even with a small number of particles**.

---

### Official Review · Reviewer_BE8T · 2026-03-13

**Soundness:** 3
**Presentation:** 4
**Significance:** 3
**Originality:** 2
**Overall Recommendation:** 4
**Confidence:** 4

**Summary:**

This paper mitigates a drawback of Particle Filtering (PF) when applied to Inference-Time Scaling (ITS) for Large Language Models on complex mathematical reasoning tasks. The authors identify that standard PF suffers from "particle impoverishment"—a premature collapse of diversity—because Process Reward Models (PRMs) often assign overly confident and noisy scores to early, unverified reasoning steps. To solve this, the authors propose Entropic Particle Filtering (ePF), which introduces two interventions: (1) Entropic Annealing (EA), which dynamically raises the resampling temperature when particle diversity (measured by Effective Sample Size, ESS) drops, and (2) Look-ahead Modulation (LaM), which re-weights particles based on a one-step look-ahead. Evaluated on rigorous math benchmarks (MATH500, AIME) using Qwen models, ePF outperforms standard PF and Best-of-N sampling.

**Compliance With Llm Reviewing Policy:**

Affirmed.

**Final Justification:**

I am maintaining my score of 4. Although the authors' rebuttal effectively addressed my concerns regarding PRM generalizability, the restricted evaluation scale ($N \le 32$) remains unresolved. The lack of large-scale experiments prevents a stronger endorsement for a paper broadly framed around inference-time scaling.

**Key Questions For Authors:**

1. Can the authors scale the experiments to much larger particle budgets (e.g., $N=64$, $N=128$, or higher)? Given the diminishing returns, does ePF eventually converge to the baseline performance of standard PF at scale?


2. The overconfidence analysis in Figure 2 and all subsequent experiments are based on a single reward model (Qwen2.5-Math-PRM-7B). Can you evaluate ePF with a completely different PRM to confirm whether this overconfidence is a universal phenomenon and if ePF remains effective?

**Limitations:**

Yes.

**Strengths And Weaknesses:**

Strengths:

- The authors succeed in isolating the reason why standard PF fails, which proves this work is highly motivated for the current LLM inference-time alignment research.

- The empirical results are strong, as they show the performance of ePF compared to PF/BoN across different models and different benchmarks.

- The paper is well-written and easy to understand. The authors clearly present their ideas and provide highly informative figures that make the architecture easy to digest.


Weaknesses:

-  The experiments only scale the particle budget up to $N=32$. This is insufficient for an "inference-time scaling" paper, especially since the authors themselves note that the relative improvements diminish as $N$ grows.

- Training a PRM is highly expensive, and ePF's success relies entirely on it. Furthermore, forcing diversity when ESS drops creates a concern: if the PRM is actually correct about a trajectory, forcing diversity actively harms correctness. It will be better if the paper includes some discussions about this trade-off.

- The paper offers no theoretical guarantees nor guidance on the choice of $N$. It's a minor point since the empirical results of this paper are strong.

---

> ### Author Rebuttal · Authors · 2026-03-27
>
> **Reviewer**:
> > The authors succeed in isolating the reason why standard PF fails, which proves this work is highly motivated for the current LLM inference-time alignment research.
>
> **Authors**:
> We thank the reviewer for highlighting the strong motivation of our work. Identifying standard Particle Filtering's vulnerability to uncalibrated, overconfident process reward models early in the reasoning process was a primary driver for this research. We are glad the reviewer found this analysis valuable for the broader LLM inference-time alignment landscape.
>
> --------
>
> **Reviewer**:
> > The empirical results are strong, as they show the performance of ePF compared to PF/BoN across different models and benchmarks.
>
> **Authors**: We appreciate the positive feedback on our empirical evaluation. Demonstrating ePF's consistent performance gains across a diverse suite of challenging mathematical reasoning benchmarks and multiple model sizes was a core objective of our study.
>
> ---------
>
> **Reviewer**:
> > The paper is well-written and easy to understand. The authors clearly present their ideas and provide highly informative figures that make the architecture easy to digest.
>
> **Authors**: We are grateful for the encouraging remarks regarding the clarity and presentation of our paper. We dedicated significant effort to ensuring the mechanics of both Entropic Annealing and Look-ahead Modulation were visually accessible and clearly communicated.
>
> ---------
>
> **Reviewer**:
> > The experiments only scale the particle budget up to 32. This is insufficient for an "inference-time scaling" paper, especially since the authors themselves note that the relative improvements diminish as N grows.
>
> **Authors**: We respectfully disagree on this point. We argue that being able to **leverage relatively small budgets** (a budget of 32 is not a small amount of additional compute) is a **significant feature of our algorithm**. As shown in Appendix E, Figure 14, ePF reaches the performance of standard PF with only 1/4 of the budget using a 1.5B model, and 1/2 of the budget using a 7B model on AIME 2024. ePF can better explore and exploit using the provided compute, making it highly efficient.
>
> ------------
>
>
> **Reviewer**:
> > if the PRM is actually correct about a trajectory, forcing diversity actively harms correctness. It will be better if the paper includes some discussions about this trade-off.
>
>
> **Authors**: Thank you for the feedback. We agree that Effective Sample Size (ESS) is a relative measure of internal diversity, not a guarantee of ground truth correctness. However, in the absence of access to the true posterior during inference, maintaining high ESS (or Entropy) early on in the sampling process is an effective and grounded mechanism to prevent the algorithm from collapsing into a single, potentially wrong mode early on. We discuss these concepts in Appendix B and we have expanded on this trade-off in the main manuscript.
>
>
> ---------------
>
>
> **Reviewer**:
> > The paper offers no theoretical guarantees nor guidance on the choice of N. It's a minor point since the empirical results of this paper are strong.
>
> **Authors**: We agree this is a valid observation. Providing theoretical bounds and formal guidance on the choice of N is an interesting direction for future work.
>
>
> --------------
>
> **Reviewer**:
> > Can the authors scale the experiments to much larger particle budgets (e.g., 64, 128, or higher)
>
> **Authors**: Thank you for the question. **Please see the response to Reviewer Mzs2 for additional experiments**.
> We can try to scale the budget further, but it becomes computationally more and more expensive for all algorithms. In the limit (e.g., N=512 or 1024), assuming a strong selection mechanism, most inference-time scaling algorithms converge to behave similarly to Best-of-N.
> A core strength of our approach is its superior sample efficiency at smaller, more practical budgets. For instance, Appendix E (Figure 14) demonstrates that ePF with Look-ahead Modulation (LaM) matches the performance of standard ePF with only 1/4 of the budget for a 1.5B model, and 1/2 the budget for a 7B model on AIME 2024. Ultimately, ePF consistently provides a better balance of exploration and exploitation under constrained compute.
>
> --------------
>
> **Reviewer:**
> > Can you evaluate ePF with a completely different PRM to confirm whether this overconfidence is a universal phenomenon and if ePF remains effective?
>
> **Authors:** Thank you for your question. We have reported additional experiments with a different PRM (Llama3.1-8B-PRM) in Appendix G.1. Here we report the results at budgets 8 and 16 using Qwen2.5-1.5B as the sampler and Llama3.1-8B-PRM as the reward model on AIME 2024:
>
> | Method | \(N=8\) | \(N=16\) |
> | :--- | :--- | :--- |
> | PF | 0.021 | 0.042 |
> | ePF | 0.034 | 0.10 |
>
> Even when paired with a weaker and more uncalibrated PRM, ePF demonstrates a clear advantage over standard PF, and this performance gain increases alongside the budget.

---

> > ### Author Rebuttal · Reviewer_BE8T · 2026-04-03
> >
> > I thank the authors for their detailed rebuttal.
> >
> > Regarding the scaling limits, I respectfully disagree with the claim that most inference-time scaling algorithms inevitably converge to behave similarly to Best-of-N at large limits like $N=512$ or $1024$. In fact, recent literature has empirically and theoretically demonstrated that structured search algorithms (such as MCTS or PRM-guided tree search) break the performance ceiling of standard parallel sampling (BoN), especially on highly complex reasoning tasks ([1, 2, 3]). Breaking the BoN ceiling is currently a popular research direction in inference-time alignment.
> >
> > That being said, I fully acknowledge and agree with your argument that the core strength of ePF is its superior sample efficiency at smaller, more practical budgets ($N \le 32$). This is a highly valuable and practical contribution to the community. However, because the method and its evaluation are bounded within this small-budget regime, framing the paper broadly as "Inference-Time Scaling" feels slightly mismatched.
> >
> > Overall, this remains a solid, practical, and well-executed paper. I will maintain my score.
> >
> >
> > **References:**
> >
> > [1]. Snell, Charlie, et al. "Scaling llm test-time compute optimally can be more effective than scaling model parameters." arXiv preprint arXiv:2408.03314 (2024).
> >
> > [2]. Lightman, Hunter, et al. "Let's verify step by step." The twelfth international conference on learning representations. 2023.
> >
> > [3]. Brown, Bradley, et al. "Large language monkeys: Scaling inference compute with repeated sampling." arXiv preprint arXiv:2407.21787 (2024).

---

> > > ### Author Response · Authors · 2026-04-03
> > >
> > > Thank you for the constructive feedback and discussion.
> > >
> > > **Reviewer**:
> > >
> > > > Breaking the BoN ceiling is currently a popular research direction in inference-time alignment.
> > >
> > > **Authors**:
> > >
> > > We agree with you and would like to clarify our statement. Our comment referred specifically to standard mathematical benchmarks, where coverage (pass@N) scales consistently with a large N (as demonstrated in Figures 2 and 3 of [3] for N > 1000). Our main point is that when pass@N approaches 1 for a sufficiently large budget, even simple inference-time scaling methods like Best-of-N become effective, provided the selection method is robust. While this assumption of strong selection is realistic for standard mathematics - since verification is feasible to a certain extent - it does not hold for complex reasoning, coding benchmarks, or tool calling.
> > >
> > > ------
> > >
> > > **Reviewer**:
> > >
> > > > This is a highly valuable and practical contribution to the community.
> > >
> > > **Authors**:
> > >
> > > Thank you for recognizing the practical value and impact of our contribution.
> > >
> > > -------
> > >
> > >
> > > Please see also the responses to Reviewer `Mzs2` and `6KaF`, where we provide additional analysis and experiments on:
> > >
> > > - new datasets (AIME 2026, HealthBench)
> > > - new model (Qwen3-4B-Instruct-2507)
> > > - new domain without explicit verification (healthcare)
> > > - different PRM (Llama3.1-8B-PRM)

---

### Decision · Program_Chairs · 2026-04-30

**Decision:**

Accept (regular)

**Comment:**

The reviewers agree to accept the article.
It is not easy to assess the significance of the article, but looks like a good contribution.